# Concentration of Multilinear Functions of the Ising Model with Applications to Network Data

**Constantinos Daskalakis** *
EECS & CSAIL, MIT
costis@csail.mit.edu

**Nishanth Dikkala***
EECS & CSAIL, MIT
nishanthd@csail.mit.edu

**Gautam Kamath***
EECS & CSAIL, MIT
g@csail.mit.edu

## Abstract

We prove near-tight concentration of measure for polynomial functions of the Ising model under high temperature. For any degree $d$, we show that a degree-$d$ polynomial of a $n$-spin Ising model exhibits exponential tails that scale as $\exp(-r^{2/d})$ at radius $r = \tilde{\Omega}_d(n^{d/2})$. Our concentration radius is optimal up to logarithmic factors for constant $d$, improving known results by polynomial factors in the number of spins. We demonstrate the efficacy of polynomial functions as statistics for testing the strength of interactions in social networks in both synthetic and real world data.

## 1 Introduction

The *Ising model* is a fundamental probability distribution defined in terms of a graph $G = (V, E)$ whose nodes and edges are associated with scalar parameters $(\theta_v)_{v \in V}$ and $(\theta_{u,v})_{\{u,v\} \in E}$ respectively. The distribution samples a vector $x \in \{\pm 1\}^V$ with probability:

$$p(x) = \exp\left(\sum_{v \in V} \theta_v x_v + \sum_{(u,v) \in E} \theta_{u,v} x_u x_v - \Phi\left(\vec{\theta}\right)\right), \tag{1}$$

where $\Phi\left(\vec{\theta}\right)$ serves to provide normalization. Roughly speaking, there is a random variable $X_v$ at every node of $G$, and this variable may be in one of two states, or spins: up ($+1$) or down ($-1$). The scalar parameter $\theta_v$ models a local field at node $v$. The sign of $\theta_v$ represents whether this local field favors $X_v$ taking the value $+1$, i.e. the up spin, when $\theta_v > 0$, or the value $-1$, i.e. the down spin, when $\theta_v < 0$, and its magnitude represents the strength of the local field. Similarly, $\theta_{u,v}$ represents the direct interaction between nodes $u$ and $v$. Its sign represents whether it favors equal spins, when $\theta_{u,v} > 0$, or opposite spins, when $\theta_{u,v} < 0$, and its magnitude corresponds to the strength of the direct interaction. Of course, depending on the structure of $G$ and the node and edge parameters, there may be indirect interactions between nodes, which may overwhelm local fields or direct interactions.

Many popular models, for example, the usual ferromagnetic Ising model [Isi25, Ons44], the Sherrington-Kirkpatrick mean field model [SK75] of spin glasses, and the Hopfield model [Hop82] of neural networks, the Curie-Weiss model [DCG68] all belong to the above family of distributions, with various special structures on $G$, the $\theta_{u,v}$'s and the $\theta_v$'s. Since its introduction in

Statistical Physics, the Ising model has found a myriad of applications in diverse research disciplines, including probability theory, Markov chain Monte Carlo, computer vision, theoretical computer science, social network analysis, game theory, computational biology, and neuroscience; see e.g. [LPW09, Cha05, Fel04, DMR11, GG86, Ell93, MS10] and their references. The ubiquity of these applications motivate the problem of inferring Ising models from samples, or inferring statistical properties of Ising models from samples. This type of problem has enjoyed much study in statistics, machine learning, and information theory; see, e.g., [CL68, AKN06, CT06, Cha07, RWL10, JJR11, SW12, BGS14, Bre15, VMLC16, BK16, Bha16, BM16, MdCCU16, KM17, HKM17, DDK18].

Despite the wealth of theoretical study and practical applications of this model, outlined above, there are still aspects of it that are poorly understood. In this work, we focus on the important topic of concentration of measure. We are interested in studying the concentration properties of polynomial functions $f(X)$ of the Ising model. That is, for a random vector $X$ sampled from $p$ as above and a polynomial $f$, we are interested in the concentration of $f(X)$ around its expectation $\mathbf{E}[f(X)]$. Since the coordinates of $X$ take values in $\{\pm 1\}$, we can without loss of generality focus our attention to multi-linear functions $f$.

While the theory of concentration inequalities for functions of independent random variables has reached a high level of sophistication, proving concentration of measure for functions of dependent random variables is significantly harder, the main tools being martingale methods, logarithmic Sobolev inequalities and transportation cost inequalities. One shortcoming of the latter methods is that explicit constants are very hard or almost impossible to get. For the Ising model, in particular, the log-Sobolev inequalities of Stroock and Zegarlinski [SZ92], known under high temperature,[2] do not give explicit constants, and it is also not clear whether they extend to systems beyond the lattice. The high temperature regime is an interesting regime of 'weak' dependence where many desirable properties related to Ising models hold. Perhaps the most important of them is that the canonical Markov chain used to sample from these models, namely the Glauber dynamics, is fast mixing. Although the high-temperature regime allows only 'weak' pairwise correlations, it is still rich enough to encode interesting dependencies. For instance, in neuroscience, it has been seen that weak pairwise correlations can coexist with strong correlations in the state of the population as a whole [SBSB06].

An alternative approach, proposed recently by Chatterjee [Cha05], is an adaptation to the Ising model of Stein's method of exchangeable pairs. This powerful method is well-known in probability theory, and has been used to derive concentration inequalities with explicit constants for functions of dependent random variables (see [MJC+14] for a recent work). Chatterjee uses this technique to establish concentration inequalities for Lipschitz functions of the Ising model under high temperature. While these inequalities are tight (and provide Gaussian tails) for linear functions of the Ising model, they are unfortunately not tight for higher degree polynomials, in that the concentration radius is off by factors that depend on the dimension $n = |V|$. For example, consider the function $f_c(X) = \sum_{i \neq j} c_{ij} X_i X_j$ of an Ising model without external fields, where the $c_{ij}$'s are signs. Chatterjee's results imply that this function concentrates at radius $\pm O(n^{1.5})$, but as we show this is suboptimal by a factor of $\tilde{\Omega}(\sqrt{n})$.

In particular, our main technical contribution is to obtain near-tight concentration inequalities for polynomial functions of the Ising model, whose concentration radii are tight up to logarithmic factors. A corollary of our main result (Theorem 4) is as follows:

**Theorem 1.** *Consider any degree-$d$ multilinear function $f$ with coefficients in $[-1, 1]$, defined on an Ising model $p$ without external field in the high-temperature regime. Then there exists a constant $C = C(d) > 0$ (depending only on $d$) such that for any $r = \tilde{\Omega}_d(n^{d/2})$, we have*

$$\Pr_{X \sim p}[|f(X) - \mathbf{E}[f(X)]| > r] \leq \exp\left(-C \cdot \frac{r^{2/d}}{n \log n}\right).$$

*The concentration radius is tight up to logarithmic factors, and the tail bound is tight up to a $O_d(1/\log n)$ factor in the exponent of the tail bound.*

Our formal theorem statements for bilinear and higher degree multilinear functions appear as Theorems 2 and 4 of Sections 3 and 4, respectively. Some further discussion of our results is in order:

- Under existence of external fields, it is easy to see that the above concentration does not hold, even for bilinear functions. Motivated by our applications in Section 5 we extend the above concentration of measure result to centered bilinear functions (where each variable $X_i$ appears as $X_i - \mathbf{E}[X_i]$ in the function) that also holds under arbitrary external fields; see Theorem 3. We leave extensions of this result to higher degree multinear functions to the next version of this paper.

- Moreover, notice that the tails for degree-2 functions are exponential and not Gaussian, and this is unavoidable, and that as the degree grows the tails become heavier exponentials, and this is also unavoidable. In particular, the tightness of our bound is justified in the supplementary material.

- Lastly, like Chatterjee and Stroock and Zegarlinski, we prove our results under high temperature. On the other hand, it is easy to construct low temperature Ising models where no non-trivial concentration holds.[3]

With our theoretical understanding in hand, we proceed with an experimental evaluation of the efficacy of multilinear functions applied to hypothesis testing. Specifically, given a binary vector, we attempt to determine whether or not it was generated by an Ising model. Our focus is on testing whether choices in social networks can be approximated as an Ising model, a common and classical assumption in the social sciences [Ell93, MS10]. We apply our method to both synthetic and real-world data. On synthetic data, we investigate when our statistics are successful in detecting departures from the Ising model. For our real-world data study, we analyze the Last.fm dataset from HetRec'11 [CBK11]. Interestingly, when considering musical preferences on a social network, we find that the Ising model may be more or less appropriate depending on the genre of music.

## 1.1 Related Work

As mentioned before, Chatterjee previously used the method of exchangeable pairs to prove variance and concentration bounds for linear statistics of the Ising model [Cha05]. In [DDK18], the authors prove variance bounds for bilinear statistics. The present work improves upon this by proving concentration rather than bounding the variance, as well as considering general degrees $d$ rather than just $d = 2$. In simultaneous work, Gheissari, Lubetzky, and Peres proved concentration bounds which are qualitatively similar to ours, though the techniques are somewhat different [GLP17].

## 2 Preliminaries

We will state some preliminaries here, see the supplementary material for further preliminaries.

We define the high-temperature regime, also known as Dobrushin's uniqueness condition – in this paper, we will use the terms interchangeably.

**Definition 1.** *Consider an Ising model $p$ defined on a graph $G = (V, E)$ with $|V| = n$ and parameter vector $\vec{\theta}$. Suppose $\max_{v \in V} \sum_{u \neq v} \tanh(|\theta_{uv}|) \leq 1 - \eta$ for some $\eta > 0$. Then $p$ is said to satisfy Dobrushin's uniqueness condition, or be in the $\eta$-high temperature regime.*

In some situations, we may use the parameter $\eta$ implicitly and simply say the Ising model is in the high temperature regime.

Glauber dynamics refers to the canonical Markov chain for sampling from an Ising model, see the supplementary material for a formal definition. Glauber dynamics define a reversible, ergodic Markov chain whose stationary distribution is identical to the corresponding Ising model. In many relevant settings, including the high-temperature regime, the dynamics are rapidly mixing and hence offer an

efficient way to sample from Ising models. In particular, the mixing time in $\eta$-high-temperature is $t_{\text{mix}} = \frac{n \log n}{\eta}$.

We may couple two executions of the Glauber dynamics using a *greedy coupling* (also known as a *monotone coupling*). Roughly, this couples the choices made by the runs to maximize the probability of agreement; see the supplementary material for a formal definition. One of the key properties of this coupling is that it satisfies the following *contraction* property:

**Lemma 1.** *If $p$ is an Ising model in $\eta$-high temperature, then the greedy coupling between two executions satisfies the following contraction in Hamming distance:*

$$\mathbf{E}\left[d_H(X_t^{(1)}, X_t^{(2)})\Big|(X_0^{(1)}, X_0^{(2)})\right] \leq \left(1 - \frac{\eta}{n}\right)^t d_H(X_0^{(1)}, X_0^{(2)}).$$

The key technical tool we use is the following concentration inequality for martingales:

**Lemma 2** (Freedman's Inequality (Proposition 2.1 in [Fre75])). *Let $X_0, X_1, \ldots, X_t$ be a sequence of martingale increments, such that $S_i = \sum_{j=0}^{i} X_j$ forms a martingale sequence. Let $\tau$ be a stopping time and $K \geq 0$ be such that $\Pr[|X_i| \leq K \ \forall \ i \leq \tau] = 1$. Let $v_i = \mathbf{Var}[X_i|X_{i-1}]$ and $V_t = \sum_{i=0}^{t} v_i$. Then $\Pr[|S_t| \geq r \text{ and } V_t \leq b \text{ for some } t \leq \tau] \leq 2 \exp\left(-\frac{r^2}{2(rK+b)}\right)$.*

## 3 Concentration of Measure for Bilinear Functions

In this section, we describe our main concentration result for bilinear functions of the Ising model. This is not as technically involved as the result for general-degree multilinear functions, but exposes many of the main conceptual ideas. The theorem statement is as follows:

**Theorem 2.** *Consider any bilinear function $f_a(x) = \sum_{u,v} a_{uv} x_u x_v$ on an Ising model $p$ (defined on a graph $G = (V, E)$ such that $|V| = n$) in $\eta$-high-temperature regime with no external field. Let $\|a\|_\infty = \max_{u,v} a_{uv}$. If $X \sim p$, then for any $r \geq 300\|a\|_\infty n \log^2 n/\eta + 2$, we have*

$$\Pr\left[|f_a(X) - \mathbf{E}[f_a(X)]| \geq r\right] \leq 5 \exp\left(-\frac{\eta r}{1735\|a\|_\infty n \log n}\right).$$

**Remark 1.** *We note that $\eta$-high-temperature is not strictly needed for our results to hold – we only need Hamming contraction of the "greedy coupling" (see Lemma 1). This condition implies rapid mixing of the Glauber dynamics (in $O(n \log n)$ steps) via path coupling (Theorem 15.1 of [LPW09]).*

### 3.1 Overview of the Technique

A well known approach to proving concentration inequalities for functions of dependent random variables is via martingale tail bounds. For instance, Azuma's inequality gives useful tail bounds whenever one can bound the martingale increments (i.e., the differences between consecutive terms of the martingale sequence) of the underlying martingale in absolute value, without requiring any form of independence. Such an approach is fruitful in showing concentration of linear functions on the Ising model in high temperature. The Glauber dynamics associated with Ising models in high temperature are fast mixing and offer a natural way to define a martingale sequence. In particular, consider the Doob martingale corresponding to any linear function $f$ for which we wish to show concentration, defined on the state of the dynamics at some time step $t^*$, i.e. $f(X_{t^*})$. If we choose $t^*$ larger than $O(n \log n)$ then $f(X_{t^*})$ would be very close to a sample from $p$ irrespective of the starting state. We set the first term of the martingale sequence as $\mathbf{E}[f(X_{t^*})|X_0]$ and the last term is simply $f(X_{t^*})$. By bounding the martingale increments we can show that $|f(X_{t^*}) - \mathbf{E}[f(X_{t^*})|X_0]|$ concentrates at the right radius with high probability. By making $t^*$ large enough we can argue that $\mathbf{E}[f(X_{t^*})|X_0] \approx \mathbf{E}[f(X)]$. Also, crucially, $t^*$ need not be too large since the dynamics are fast mixing. Hence we don't incur too big a hit when applying Azuma's inequality, and one can argue that linear functions are concentrated with a radius of $\tilde{O}(\sqrt{n})$. Crucial to this argument is the fact that linear functions are $O(1)$-Lipschitz (when the entries of $a$ are constant), bounding the Doob martingale differences to be $O(1)$.

The challenge with bilinear functions is that they are $O(n)$-Lipschitz – a naive application of the same approach gives a radius of concentration of $\tilde{O}(n^{3/2})$, which albeit better than the trivial radius

of $O(n^2)$ is not optimal. To show stronger concentration for bilinear functions, at a high level, the idea is to bootstrap the known fact that linear functions of the Ising model concentrate well at high temperature.

The key insight is that, when we have a $d$-linear function, its Lipschitz constants are bounds on the absolute values of certain $d-1$-linear functions. In particular, this implies that the Lipschitz constants of a bilinear function are bounds on the absolute values of certain associated linear functions. And although a worst case bound on the absolute value of linear functions with bounded coefficients would be $O(n)$, the fact that linear functions are concentrated within a radius of $\tilde{O}(\sqrt{n})$, means that bilinear functions are $\tilde{O}(\sqrt{n})$-Lipschitz in spirit. In order to exploit this intuition, we turn to more sophisticated concentration inequalities, namely Freedman's inequality (Lemma 2). This is a generalization of Azuma's inequality, which handles the case when the martingale differences are only bounded until some stopping time (very roughly, the first time we reach a state where the expectation of the linear function after mixing is large). To apply Freedman's inequality, we would need to define a stopping time which has two properties:

1. The stopping time is larger than $t^*$ with high probability. Hence, with a good probability the process doesn't stop too early. The harm if the process stops too early (at $t < t^*$) is that we will not be able to effectively decouple $\mathbf{E}\left[f_a(X_t)|X_0\right]$ from the choice of $X_0$. $t^*$ is chosen to be larger than the mixing time of the Glauber dynamics precisely because it allows us to argue that $\mathbf{E}\left[f_a(X_{t^*})|X_0\right] \approx \mathbf{E}\left[f_a(X_{t^*})\right] = \mathbf{E}[f_a(X)]$.

2. For all times $i+1$ less than the stopping time, the martingale increments are bounded, i.e. $|B_{i+1} - B_i| = O(\sqrt{n})$ where $\{B_i\}_{i \geq 0}$ is the martingale sequence.

We observe that the martingale increments corresponding to a martingale defined on a bilinear function have the flavor of the conditional expectations of certain linear functions which can be shown to concentrate at a radius $\tilde{O}(\sqrt{n})$ when the process starts at its stationary distribution. This provides us with a nice way of defining the stopping time to be the first time when one of these conditional expectations deviates by more than $\Omega(\sqrt{n}\,\text{poly}\log n)$ from the origin. More precisely, we define a set $G_K^a(t)$ of configurations $x_t$, which is parameterized by a function $f_a(X)$ and parameter $K$ (which we will take to be $\tilde{\Omega}(\sqrt{n})$). The objects of interest are linear functions $f_a^v(X_{t^*})$ conditioned on $X_t = x_t$, where $f_a^v$ are *linear* functions which arise when examining the evolution of $f_a$ over steps of the Glauber dynamics. $G_K^a(t)$ are the set of configurations for which all such linear functions satisfy certain conditions, including bounded expectation and concentration around their mean. The stopping time for our process $T_K$ is defined as the first time we have a configuration which leaves this set $G_K^a(t)$. We can show that the stopping time is large via the following lemma:

**Lemma 3.** *For any $t \geq 0$, for $t^* = 3t_{mix}$,*

$$\Pr\left[X_t \notin G_K^a(t)\right] \leq 8n \exp\left(-\frac{K^2}{8t^*}\right).$$

Next, we require a bound on the conditional variance of the martingale increments. This can be shown using the property that the martingale increments are bounded up until the stopping time:

**Lemma 4.** *Consider the Doob martingale where $B_i = \mathbf{E}[f_a(X_{t^*})|X_i]$. Suppose $X_i \in G_K^a(i)$ and $X_{i+1} \in G_K^a(i+1)$. Then*

$$|B_{i+1} - B_i| \leq 16K + 16n^2 \exp\left(-\frac{K^2}{16t^*}\right).$$

With these two pieces in hand, we can apply Freedman's inequality to bound the desired quantity.

It is worth noting that the martingale approach described above closely relates to the technique of exchangeable pairs exposited by Chatterjee [Cha05]. When we look at differences for the martingale sequence defined using the Glauber dynamics, we end up analyzing an exchangeable pair of the following form: sample $X \sim p$ from the Ising model. Take a step along the Glauber dynamics starting from $X$ to reach $X'$. $(X, X')$ forms an exchangeable pair. This is precisely how Chatterjee's application of exchangeable pairs is set up. Chatterjee then goes on to study a function of $X$ and $X'$ which serves as a proxy for the variance of $f(X)$ and obtains concentration results by bounding the absolute value of this function. The definition of the function involves considering two greedily coupled runs of the Glauber dynamics just as we do in our martingale based approach.

To summarize, our proof of bilinear concentration involves showing various concentration properties for linear functions via Azuma's inequality, showing that the martingale has $\tilde{O}(\sqrt{n})$-bounded differences before our stopping time, proving that the stopping time is larger than the mixing time with high probability, and combining these ingredients using Freedman's inequality. Full details are provided in the supplementary material.

## 3.2 Concentration Under an External Field

Under an external field, not all bilinear functions concentrate nicely even in the high temperature regime – in particular, they may concentrate with a radius of $\Theta(n^{1.5})$, instead of $O(n)$. As such, we must instead consider "recentered" statistics to obtain the same radius of concentration. The following theorem is proved in the supplementary material:

**Theorem 3.** *1. Bilinear functions on the Ising model of the form $f_a(X) = \sum_{u,v} a_{uv}(X_u - \mathbf{E}[X_u])(X_v - \mathbf{E}[X_v])$ satisfy the following inequality at high temperature. There exist absolute constants $c$ and $c'$ such that, for $r \geq cn \log^2 n/\eta$,*

$$\Pr\left[|f_a(X) - \mathbf{E}[f_a(X)]| \geq r\right] \leq 4\exp\left(-\frac{r}{c'n \log n}\right).$$

*2. Bilinear functions on the Ising model of the form $f_a(X^{(1)}, X^{(2)}) = \sum_{u,v} a_{uv}(X_u^{(1)} - X_u^{(2)})(X_v^{(1)} - X_v^{(2)})$, where $X^{(1)}, X^{(2)}$ are two i.i.d samples from the Ising model, satisfy the following inequality at high temperature. There exist absolute constants $c$ and $c'$ such that, for $r \geq cn \log^2 n/\eta$,*

$$\Pr\left[\left|f_a(X^{(1)}, X^{(2)}) - \mathbf{E}[f_a(X^{(1)}, X^{(2)})]\right| \geq r\right] \leq 4\exp\left(-\frac{r}{c'n \log n}\right).$$

# 4 Concentration of Measure for $d$-linear Functions

More generally, we can show concentration of measure for $d$-linear functions on an Ising model in high temperature, when $d \geq 3$. Again, we will focus on the setting with *no external field*. Although we will follow a recipe similar to that used for bilinear functions, the proof is more involved and requires some new definitions and tools. The proof will proceed by induction on the degree $d$. Due to the proof being more involved, for ease of exposition, we present the proof of Theorem 4 without explicit values for constants.

Our main theorem statement is the following:

**Theorem 4.** *Consider any degree-$d$ multilinear function $f_a(x) = \sum_{U \subseteq V:|U|=d} a_U \prod_{u \in U} x_u$ on an Ising model $p$ (defined on a graph $G = (V, E)$ such that $|V| = n$) in $\eta$-high-temperature regime with no external field. Let $\|a\|_\infty = \max_{U \subseteq V:|U|=d} |a_U|$. There exist constants $C_1 = C_1(d) > 0$ and $C_2 = C_2(d) > 0$ depending only on $d$, such that if $X \sim p$, then for any $r \geq C_1 \|a\|_\infty (n \log^2 n/\eta)^{d/2}$, we have*

$$\Pr\left[|f_a(X) - \mathbf{E}[f_a(X)]| > r\right] \leq 2\exp\left(-\frac{\eta r^{2/d}}{C_2 \|a\|_\infty^{2/d} n \log n}\right).$$

Similar to Remark 1, our theorem statement still holds under the weaker assumption of Hamming contraction. This bound is also tight up to polylogarithmic factors in the radius of concentration and the exponent of the tail bound, see Remark 1 in the supplementary material.

## 4.1 Overview of the Technique

Our approach uses induction and is similar to the one used for bilinear functions. To show concentration for $d$-linear functions we will use the concentration of $(d-1)$-linear functions together with Freedman's martingale inequality.

Consider the following process: Sample $X_0 \sim p$ from the Ising model of interest. Starting at $X_0$, run the Glauber dynamics associated with $p$ for $t^* = (d+1)t_{\text{mix}}$ steps. We will study the target

quantity, $\Pr\left[\left|f_a(X_{t^*}) - \mathbf{E}[f_a(X_{t^*})|X_0]\right| > K\right]$, by defining a martingale sequence similar to the one in the bilinear proof. However, to bound the increments of the martingale for $d$-linear functions we will require an induction hypothesis which is more involved. The reason is that with higher degree multilinear functions ($d > 2$), the argument for bounding increments of the martingale sequence runs into multilinear terms which are a function of not just a single instance of the dynamics $X_t$, but also of the configuration obtained from the coupled run, $X_t'$. We call such multilinear terms *hybrid* terms and multilinear functions involving *hybrid* terms as *hybrid* multilinear functions henceforth. Since the two runs (of the Glauber dynamics) are coupled greedily to maximize the probability of agreement and they start with a small Hamming distance from each other ($\leq 1$), these hybrid terms behave very similar to the non-hybrid multilinear terms. Showing that their behavior is similar, however, requires some supplementary statements about them which are presented in the supplementary material. In addition to the martingale technique of Section 3, an ingredient that is crucial to the proving concentration for $d \geq 3$ is a bound on the magnitude of the $(d-1)$-order marginals of the Ising model:

**Lemma 5.** *Consider any Ising model $p$ at high temperature. Let $d$ be a positive integer. We have*

$$\left| \sum_{u_1,\ldots,u_d} \mathbf{E}_p[X_{u_1}X_{u_2}\ldots,X_{u_d}] \right| \leq 2\left(\frac{4nd\log n}{\eta}\right)^{d/2}.$$

This is because when studying degree $d \geq 3$ functions we find ourselves having to bound expected values of degree $d-1$ multilinear functions on the Ising model. A naive bound of $O_d(n^{d-1})$ can be argued for these functions but by exploiting the fact that we are in high temperature, we can show a bound of $O_d(n^{(d-1)/2})$ via a coupling with the Fortuin-Kastelyn model. When $d = 2$, $(d-1)$-linear functions are just linear functions which are zero mean. However, for $d \geq 3$, this is not the case. Hence, we first need to prove this desired bound on the marginals of an Ising model in high temperature.

Further details are provided in the supplementary material.

## 5 Experiments

In this section, we apply our family of bilinear statistics on the Ising model to a problem of statistical hypothesis testing. Given a single sample from a multivariate distribution, we attempt to determine whether or not this sample was generated from an Ising model in the high-temperature regime. More specifically, the null hypothesis is that the sample is drawn from an Ising model with a known graph structure with a common edge parameter and a uniform node parameter (which may potentially be known to be 0). In Section 5.1, we apply our statistics to synthetic data. In Section 5.2, we turn our attention to the Last.fm dataset from HetRec 2011 [CBK11].

The running theme of our experimental investigation is testing the classical and common assumption which models choices in social networks as an Ising model [Ell93, MS10]. To be more concrete, choices in a network could include whether to buy an iPhone or an Android phone, or whether to vote for a Republican or Democratic candidate. Such choices are naturally influenced by one's neighbors in the network – one may be more likely to buy an iPhone if he sees all his friends have one, corresponding to an Ising model with positive-weight edges[4] In our synthetic data study, we will leave these choices as abstract, referring to them only as "values," but in our Last.fm data study, these choices will be whether or not one listens to a particular artist.

Our general algorithmic approach is as follows. Given a single multivariate sample, we first run the maximum pseudo-likelihood estimator (MPLE) to obtain an estimate of the model's parameters. The MPLE is a canonical estimator for the parameters of the Ising model, and it enjoys strong consistency guarantees in many settings of interest [Cha07, BM16]. If the MPLE gives a large estimate of the model's edge parameter, this is sufficient evidence to reject the null hypothesis. Otherwise, we use Markov Chain Monte Carlo (MCMC) on a model with the MPLE parameters to determine a range of values for our statistic. We note that, to be precise, we would need to quantify the error incurred by the MPLE – in favor of simplicity in our exploratory investigation, we eschew this detail, and

at this point attempt to reject the null hypothesis of the model learned by the MPLE. Our statistic is bilinear in the Ising model, and thus enjoys the strong concentration properties explained earlier in this paper. Note that since the Ising model will be in the high-temperature regime, the Glauber dynamics mix rapidly, and we can efficiently sample from the model using MCMC. Finally, given the range of values for the statistic determined by MCMC, we reject the null hypothesis if $p \leq 0.05$.

## 5.1 Synthetic Data

We proceed with our investigation on synthetic data. Our null hypothesis is that the sample is generated from an Ising model in the high temperature regime on the grid, with no external field (i.e. $\theta_u = 0$ for all $u$) and a common (unknown) edge parameter $\theta$ (i.e., $\theta_{uv} = \theta$ iff nodes $u$ and $v$ are adjacent in the grid, and 0 otherwise). For the Ising model on the grid, the critical edge parameter for high-temperature is $\theta_c = \frac{\ln(1+\sqrt{2})}{2}$. In other words, we are in high-temperature if and only if $\theta \leq \theta_c$, and we can reject the null hypothesis if the MPLE estimate $\hat{\theta} > \theta_c$.

To generate departures from the null hypothesis, we give a construction parameterized by $\tau \in [0, 1]$. We provide a rough description of the departures, for a precise description, see the supplemental material. Each node $x$ selects a random node $y$ at Manhattan distance at most 2, and sets $y$'s value to $x$ with probability $\tau$. The intuition behind this construction is that each individual selects a friend or a friend-of-a-friend, and tries to convince them to take his value – he is successful with probability $\tau$. Selecting either a friend or a friend-of-a-friend is in line with the concept of strong triadic closure [EK10] from the social sciences, which suggests that two individuals with a mutual friend are likely to either already be friends (which the social network may not have knowledge of) or become friends in the future.

An example of a sample generated from this distribution with $\tau = 0.04$ is provided in Figure 1 of the supplementary material, alongside a sample from the Ising model generated with the corresponding MPLE parameters. We consider this distribution to pass the "eye test" – one can not easily distinguish these two distributions by simply glancing at them. However, as we will see, our multilinear statistic is able to correctly reject the null a large fraction of the time.

Our experimental process was as follows. We started with a $40 \times 40$ grid, corresponding to a distribution with $n = 1600$ dimensions. We generated values for this grid according to the depatures from the null described above, with some parameter $\tau$. We then ran the MPLE estimator to obtain an estimate for the edge parameter $\hat{\theta}$, immediately rejecting the null if $\hat{\theta} > \theta_c$. Otherwise, we ran the Glauber dynamics for $O(n \log n)$ steps to generate a sample from the grid Ising model with parameter $\hat{\theta}$. We repeated this process to generate 100 samples, and for each sample, computed the value of the statistic $Z_{local} = \sum_{u=(i,j)} \sum_{v=(k,l):d(u,v) \leq 2} X_u X_v$, where $d(\cdot, \cdot)$ is the Manhattan distance on the grid. This statistic can be justified since we wish to account for the possibility of connections between friends-of-friends of which the social network may be lacking knowledge. We then compare with the value of the statistic $Z_{local}$ on the provided sample, and reject the null hypothesis if this statistic corresponds to a $p$-value of $\leq 0.05$. We repeat this for a wide range of values of $\tau \in [0, 1]$, and repeat 500 times for each $\tau$.

Our results are displayed in Figure 1 The x-axis marks the value of parameter $\tau$, and the y-axis indicates the fraction of repetitions in which we successfully rejected the null hypothesis. The performance of the MPLE alone is indicated by the orange line, while the performance of our statistic is indicated by the blue line. We find that our statistic is able to correctly reject the null at a much earlier point than the MPLE alone. In particular, our statistic manages to reject the null for $\tau \geq 0.04$, while the MPLE requires a parameter which is an order of magnitude larger, at $0.4$. As mentioned before, in the former regime (when $\tau \approx 0.04$), it appears impossible to distinguish the distribution from a sample from the Ising model with the naked eye.

## 5.2 Last.fm Dataset

We now turn our focus to the Last.fm dataset from HetRec'11 [CBK11]. This dataset consists of data from $n = 1892$ users on the Last.fm online music system. On Last.fm, users can indicate (bi-directional) friend relationships, thus constructing a social network – our dataset has $m = 12717$ such edges. The dataset also contains users' listening habits – for each user we have a list of their

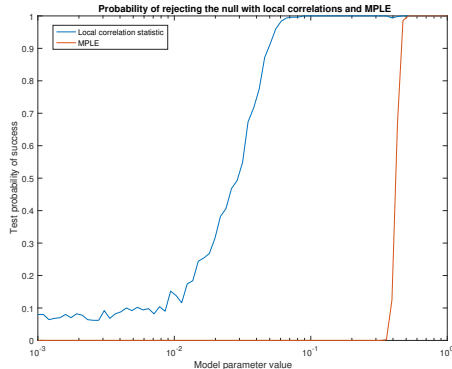

Figure 1: Power of our statistic on synthetic data.

fifty favorite artists, whose tracks they have listened to the most times. We wish to test whether users' preference for a particular artist is distributed according to a high-temperature Ising model.

Fixing some artist $a$ of interest, we consider the vector $X^{(a)}$, where $X_u^{(a)}$ is $+1$ if user $u$ has artist $a$ in his favorite artists, and $-1$ otherwise. We wish to test the null hypothesis, whether $X^{(a)}$ is distributed according to an Ising model in the high temperature regime on the known social network graph, with common (unknown) external field $h$ (i.e. $\theta_u = h$ for all $u$) and edge parameter $\theta$ (i.e., $\theta_{uv} = \theta$ iff $u$ and $v$ are neighbors in the graph, and $0$ otherwise).

Our overall experimental process was very similar to the synthetic data case. We gathered a list of the ten most-common favorite artists, and repeated the following process for each artist $a$. We consider the vector $X^{(a)}$ (defined above) and run the MPLE estimator on it, obtaining estimates $\hat{h}$ and $\hat{\theta}$. We then run MCMC to generate 100 samples from the Ising model with these parameters, and for each sample, computed the value of the statistics $Z_k = \sum_u \sum_{v:d(u,v) \leq k} (X_u - \tanh(\hat{h}))(X_v - \tanh(\hat{h}))$, where $d(\cdot, \cdot)$ is the distance on the graph, and $k = 1$ (the neighbor correlation statistic) or 2 (the local correlation statistic). Motivated by our theoretical results (Theorem 3), we consider a statistic where the variables are recentered by their marginal expectations, as this statistic experiences sharper concentration. We again consider $k = 2$ to account for the possibility of edges which are unknown to the social network.

Strikingly, we found that the plausibility of the Ising modelling assumption varies significantly depending on the artist. We highlight some of our more interesting findings here, see the supplemental material for more details. The most popular artist in the dataset was Lady Gaga, who was a favorite artist of 611 users in the dataset. We found that $X^{(\text{Lady Gaga})}$ had statistics $Z_1 = 9017.3$ and $Z_2 = 106540$. The range of these statistics computed by MCMC can be seen in Figure 2 of the supplementary material – clearly, the computed statistics fall far outside these ranges, and we can reject the null hypothesis with $p \ll 0.01$. Similar results held for other popular pop musicians, including Britney Spears, Christina Aguilera, Rihanna, and Katy Perry.

However, we observed qualitatively different results for The Beatles, the fourth most popular artist, being a favorite of 480 users. We found that $X^{(\text{The Beatles})}$ had statistics $Z_1 = 2157.8$ and $Z_2 = 22196$. The range of these statistics computed by MCMC can be seen in Figure 3 of the supplementary material. This time, the computed statistics fall near the center of this range, and we can not reject the null. Similar results held for the rock band Muse.

Based on our investigation, our statistic seems to indicate that for the pop artists, the null fails to effectively model the distribution, while it performs much better for the rock artists. We conjecture that this may be due to the highly divisive popularity of pop artists like Lady Gaga and Britney Spears – while some users may love these artists (and may form dense cliques within the graph), others have little to no interest in their music. The null would have to be expanded to accomodate heterogeneity to model such effects. On the other hand, rock bands like The Beatles and Muse seem to be much more uniform in their appeal: users seem to be much more homogeneous when it comes to preference for these groups.

## Acknowledgments

Research was supported by NSF CCF-1617730, CCF-1650733, and ONR N00014-12-1-0999. Part of this work was done while GK was an intern at Microsoft Research New England.

## Footnotes

*Authors are listed in alphabetical order.

[2]High temperature is a widely studied regime of the Ising model where it enjoys a number of useful properties such as decay of correlations and fast mixing of the Glauber dynamics. Throughout this paper we will take "high temperature" to mean that Dobrushin's conditions of weak dependence are satisfied. See Definition 1.

[3]Consider an Ising model with no external fields, comprising two disjoint cliques of half the vertices with infinitely strong bonds; i.e. $\theta_v = 0$ for all $v$, and $\theta_{u,v} = \infty$ if $u$ and $v$ belong to the same clique. Now consider the multilinear function $f(X) = \sum_{u \not\sim v} X_u X_v$, where $u \not\sim v$ denotes that $u$ and $v$ are not neighbors (i.e. belong to different cliques). It is easy to see that the maximum absolute value of $f(X)$ is $\Omega(n^2)$ and that there is no concentration at radius better than some $\Omega(n^2)$.

[4]Note that one may also decide *against* buying an iPhone in this scenario, if one places high value on individuality and uniqueness – this corresponds to negative-weight edges.

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
