[Supplementary Material]

# Supplementary Material for Concentration of Multilinear Functions of the Ising Model with Applications to Network Data

Constantinos Daskalakis[*]
EECS & CSAIL, MIT
costis@csail.mit.edu

Nishanth Dikkala[†]
EECS & CSAIL, MIT
nishanthd@csail.mit.edu

Gautam Kamath[‡]
EECS & CSAIL, MIT
g@csail.mit.edu

## 1 Supplementary Preliminaries

We will abuse notation, referring to both the probability distribution $p$ and the random vector $X$ that it samples in $\{\pm 1\}^V$ as the Ising model. That is, $X \sim p$. We will subscript $X$ as follows. At times, we will consider a sequence of $X$'s at various "time steps" – we will use $X_t$ or $X_i$ to denote random vectors in this sequence. Other times, we will need to consider the value of the vector $X$ at a particular node – we will use $X_u$ or $X_v$ to indicate random variables in this sequence. Whether we index based on time step versus node should be apparent from the choice of subscript variable, and otherwise clear from context. Occationally, we will use both: $X_{t,u}$ denotes the variable corresponding to node $u$ in the Ising model $X$ at some time step $t$. Throughout the paper we will refer to the set $\Omega = \{\pm 1\}^V$.

**Definition 1.** *A* degree-$d$ multilinear function *defined on $n$ variables $x_1, \ldots, x_n$ is a polynomial such that*

$$\sum_{S \subseteq [n]:|S| \leq d} a_S \prod_{i \in S} x_i,$$

*where $a : 2^{[n]} \to \mathbb{R}$ is a coefficient vector.*

When the degree $d = 1$, we will refer to the function as a linear function, and when the degree $d = 2$ we will call it a bilinear function. Note that since $X_u \in \{\pm 1\}$, any polynomial function of an Ising model is a multilinear function. We will use $a$ to denote the coefficient vector of such a multilinear function. Note that we will use permutations of the subscripts to refer to the same coefficient, i.e., $a_{uv}$ is the same as $a_{vu}$. Also we will use the term $d$-linear function to refer to a multilinear function of degree $d$.

We say an Ising model has *no external field* if $\theta_v = 0$ for all $v \in V$. An Ising model is *ferromagnetic* if $\theta_e \geq 0$ for all $e \in E$.

We now give a formal definition of the high-temperature regime, also known as Dobrushin's uniqueness condition – in this paper, we will use the terms interchangeably.

**Definition 2** (Dobrushin's Uniqueness Condition)**.** *Consider an Ising model $p$ defined on a graph $G = (V, E)$ with $|V| = n$ and parameter vector $\vec{\theta}$. Suppose $\max_{v \in V} \sum_{u \neq v} \tanh(|\theta_{uv}|) \leq 1 - \eta$ for*

---

[*]Supported by NSF CCF-1617730, CCF-1650733, and ONR N00014-12-1-0999.

[†]Supported by NSF CCF-1617730, CCF-1650733, and ONR N00014-12-1-0999.

[‡]Supported by NSF CCF-1617730, CCF-1650733, and ONR N00014-12-1-0999. Part of this work was done while the author was an intern at Microsoft Research New England.

*some $\eta > 0$. Then $p$ is said to satisfy Dobrushin's uniqueness condition, or be in the high temperature regime. In this paper, we use the notation that an Ising model is $\eta$-high temperature to parameterize the extent to which it is inside the high temperature regime. Note that since $\tanh(|x|) \leq |x|$ for all $x$, the above condition follows from more simplified conditions which avoid having to deal with hyperbolic functions. For instance, either of the following two conditions:*

$$\max_{v \in V} \sum_{u \neq v} |\theta_{uv}| \leq 1 - \eta \ \ or$$

$$\beta d_{\max} \leq 1 - \eta$$

*are sufficient to imply Dobrushin's condition (where $\beta = \max_{u,v} |\theta_{uv}|$ and $d_{\max}$ is the maximum degree of $G$).*

In some situations, we may use the parameter $\eta$ implicitly and simply say the Ising model is in the high temperature regime. In general, when one refers to the *temperature* of an Ising model, a high temperature corresponds to small $\theta_{uv}$ values, and a low temperature corresponds to large $\theta_{uv}$ values.

We will use the following lemma which shows concentration of measure for Lipschitz functions on the Ising model in high temperature. It is a well-known result and can be found for instance as Theorem 4.3 of [Cha05].

**Lemma 1** (Lipschitz Concentration Lemma). *Suppose that $f(X_1, \ldots, X_n)$ is a function of an Ising model in the high-temperature regime. Suppose the Lipschitz constants of $f$ are $l_1, l_2, \ldots, l_n$ respectively. That is,*

$$\left| f(X_1, \ldots, X_i, \ldots, X_n) - f(X_1, \ldots, X'_i, \ldots, X_n) \right| \leq l_i$$

*for all values of $X_1, \ldots, X_{i-1}, X_{i+1}, \ldots, X_n$ and for any $X_i$ and $X'_i$. Then,*

$$\Pr\left[ |f(X) - \mathbf{E}[f(X)]| > t \right] \leq 2 \exp\left( -\frac{\eta t^2}{2 \sum_{i=1}^{n} l_i^2} \right).$$

Note that this immediately implies sharp concentration bounds for linear functions on the Ising model.

We will refer to elements in $\Omega$ as both states and configurations of the Ising model. The name states will be more natural when considering Markov chains such as the Glauber dynamics. Glauber dynamics is the canonical Markov chain for sampling from an Ising model. Glauber dynamics define a reversible, ergodic Markov chain whose stationary distribution is identical to the corresponding Ising model. In many relevant settings, including the high-temperature regime, the dynamics are rapidly mixing (i.e., in $O(n \log n)$ steps) and hence offer an efficient way to sample from Ising models. We consider the basic variant known as single-site Glauber dynamics. The dynamics are a Markov chain defined on the set $\Omega$. They proceed as follows:

1. Let $X_t$ denote the state of the dynamics at time $t$. Start at any state $X_0 \in \Omega$.

2. Let $N(u)$ be the set of neighbors of node $u$. Pick a node $u$ uniformly at random and update $X_u$ as follows

$$X_{t+1,u} = 1 \quad \text{w.p.} \quad \frac{\exp\left( \theta_u + \sum_{v \in N(u)} \theta_{uv} X_{t,v} \right)}{\exp\left( \theta_u + \sum_{v \in N(u)} \theta_{uv} X_{t,v} \right) + \exp\left( -\theta_u - \sum_{v \in N(u)} \theta_{uv} X_{t,v} \right)}$$

$$X_{t+1,u} = -1 \quad \text{w.p.} \quad \frac{\exp\left( -\theta_u - \sum_{v \in N(u)} \theta_{uv} X_{t,v} \right)}{\exp\left( \theta_u + \sum_{v \in N(u)} \theta_{uv} X_{t,v} \right) + \exp\left( -\theta_u - \sum_{v \in N(u)} \theta_{uv} X_{t,v} \right)}$$

Glauber dynamics for an Ising model in the high temperature regime are fast mixing. In particular, they mix in $O(n \log n)$ steps. To be more concrete, for an Ising model $p$ in $\eta$-high temperature, we define

$$t_{\text{mix}} = \frac{n \log n}{\eta},\tag{1}$$

The dynamics for an Ising model in high temperature also display the cutoff phenomenon. Due to this, we have Lemma 2.

**Lemma 2.** *Let $x_0$ be any starting state for the Glauber dynamics and let $t^* = (\zeta + 2)t_{mix}$ for some $0 \leq d \leq n$. If $X_{t^*,x_0}$ is the state reached after $t^*$ steps of the dynamics, then*

$$d_{\text{TV}}(X_{t^*,x_0}, p) \leq \exp\left(-(\zeta + 1)n \log n\right)$$

*for all $x_0$.*

*Proof.* This follows in a straightforward manner from the cutoff phenomenon observed with respect to the mixing of the Glauber dynamics in this setting. The bound on the mixing time of Glauber dynamics for high temperature Ising models (Theorem 15.1 of [LPW09])[1] gives us that to achieve $d_{\text{TV}}(X_t, p) \leq \varepsilon$, we must run the dynamics for $t = \frac{n \log n + \log(1/\varepsilon)}{\eta}$ steps. This implies, that after $t^*$ steps, the total variation distance $\varepsilon$ achieved is

$$\varepsilon \leq \exp(-t^* \eta + n \log n)$$
$$= \exp(-(\zeta + 1)n \log n).$$

$\square$

**Definition 3.** *The* Hamming distance *between $x, y \in \{\pm 1\}^n$ is defined as $d_H(x, y) = \sum_{i \in [n]} \mathbb{1}_{\{x_i \neq y_i\}}$.*

**Definition 4** (The greedy coupling)**.** *Consider two instances of Glauber dynamics associated with the same Ising model $p$: $X_0^{(1)}, X_1^{(1)}, \ldots$ and $X_0^{(2)}, X_1^{(2)}, \ldots$. The following coupling procedure is known as the* greedy coupling. *Start chain 1 at $X_0^{(1)}$ and chain 2 at $X_0^{(2)}$ and in each time step $t$, choose a node $v \in V$ uniformly at random to update in both the runs. Let $p^{(1)}$ denote the probability that the first chain sets $X_{t,v}^{(1)} = 1$ and let $p^{(2)}$ be the probability that the second chain sets $X_{t,v}^{(2)} = 1$. Let $p_1 \leq p_2$ be a rearrangement of the $p^{(i)}$ values in increasing order. Also let $p_0 = 0$ and $p_3 = 1$. Draw a number $x$ uniformly at random from $[0, 1]$ and couple the updates according to the following rule:*

*If $x \in [p_l, p_{l+1}]$ for some $0 \leq l \leq 2$, set $X_{t,v}^{(i)} = -1$ for all $1 \leq i \leq l$ and $X_{t,v}^{(i)} = 1$ for all $l < i \leq 2$.*

We summarize some properties of this coupling in the following lemma, which appear in Chapter 15 of [LPW09].

**Lemma 3.** *The greedy coupling (Definition 4) satisfies the following properties.*

1. *It is a valid coupling.*

2. *If $p$ is an Ising model in $\eta$-high temperature, then*

$$\mathbf{E}\left[d_H(X_t^{(1)}, X_t^{(2)}) \Big| (X_0^{(1)}, X_0^{(2)})\right] \leq \left(1 - \frac{\eta}{n}\right)^t d_H(X_0^{(1)}, X_0^{(2)}).$$

3. *The distribution of $X_t^{(1)}$, for any $t \geq 0$, conditioned on $X_0^{(1)}$ is independent of $X_0^{(2)}$.*

## 1.1 Martingales

We briefly review some definitions from the theory of martingales in this section.

**Definition 5.** *A* probability space *is defined by a triple* $(O, \mathcal{F}, P)$ *where $O$ is the possible set of outcomes of the probability space. $\mathcal{F}$ is a $\sigma$-field which is a set of all measurable events of the space and $P$ is a function which maps events in $\mathcal{F}$ to probability values.*

**Definition 6.** *A sequence of random variables $X_0, X_1, \ldots, X_i, \ldots$ on the probability space $(O, \mathcal{F}, P)$ is a* martingale sequence *if for all $i \geq 0$, $\mathbf{E}[X_{i+1}|\mathcal{F}_i] = X_i$.*

**Definition 7.** *A* stopping time *with respect to a martingale sequence defined on $(O, \mathcal{F}, P)$ is a function $\tau : O \to \{1, 2, \ldots\}$ such that $\{\tau = n\} \in \mathcal{F}_n$ for all $n$. Also, $P[\tau = \infty] > 0$ is allowed.*

**Definition 8.** *Let $X_1, X_2, \ldots, X_n$ be a set of possibly dependent random variables. Consider any function $f(X_1, X_2, \ldots, X_n)$ on them. Then the sequence $\{B_i\}_{i \geq 1}$ where*

$$B_i = \mathbf{E}[f(X_1, X_2, \ldots, X_n)|X_1, X_2, \ldots, X_i] \tag{2}$$

*is a martingale sequence and is known as the* Doob martingale *of the function $f(.)$.*

A popular set of tools which have been used for showing concentration results such as McDiarmid's inequality come from the theory of martingales. In our proof, the following two martingale inequalities will be useful. The first is the well-known Azuma's inequality.

**Lemma 4** (Azuma's Inequality)**.** *Let $(\Omega, \mathcal{F}, P)$ be a probability space. Let $\mathcal{F}_0 \subset \mathcal{F}_1 \subset \mathcal{F}_2 \ldots$ be an increasing sequence of sub-$\sigma$-fields of $\mathcal{F}$. Let $X_0, X_1, \ldots, X_t$ be random variables on $(\Omega, \mathcal{F}, P)$ such that $X_i$ is $\mathcal{F}_i$-measurable. Suppose they represent a sequence of martingale increments. That is, $\mathbf{E}[X_i|\mathcal{F}_{i-1}] = 0$ or $S_i = \sum_{j=0}^i X_j$ forms a martingale sequence defined on the space $(\Omega, \mathcal{F}, P)$. Let $K \geq 0$ be such that $\Pr[|X_i| \leq K] = 1$ for all $i$. Then for all $r \geq 0$,*

$$\Pr[|S_t| \geq r] \leq 2 \exp\left(-\frac{r^2}{tK^2}\right)$$

The second inequality due to Freedman is a generalization of Azuma's inequality. It applies when a bound on the martingale increments $|X_i|$ only holds until some stopping time, unlike Azuma's, which requires a bound on the martingale increments for all times.

**Lemma 5** (Freedman's Inequality (Proposition 2.1 in [Fre75]))**.** *Let $(\Omega, \mathcal{F}, P)$ be a probability space. Let $\mathcal{F}_0 \subset \mathcal{F}_1 \subset \mathcal{F}_2 \ldots$ be an increasing sequence of sub-$\sigma$-fields of $\mathcal{F}$. Let $X_0, X_1, \ldots, X_t$ be random variables on $(\Omega, \mathcal{F}, P)$ such that $X_i$ is $\mathcal{F}_i$-measurable. Suppose they represent a sequence of martingale increments. That is, $S_i = \sum_{j=0}^i X_j$ forms a martingale sequence defined on the space $(\Omega, \mathcal{F}, P)$. Let $\tau$ be a stopping time defined on $\Omega$ and $K \geq 0$ be such that $\Pr[|X_i| \leq K] = 1$ for $i \leq \tau$. Let $v_i = \mathbf{Var}[X_i|\mathcal{F}_{i-1}]$ and $V_t = \sum_{i=0}^t v_i$. Then,*

$$\Pr[|S_t| \geq r \text{ and } V_t \leq b \text{ for some } t \leq \tau] \leq 2 \exp\left(-\frac{r^2}{2(rK+b)}\right) \tag{3}$$

$$\equiv \quad \Pr[\exists t \leq \tau \text{ s.t } |S_t| \geq r \text{ and } V_t \leq b] \leq 2 \exp\left(-\frac{r^2}{2(rK+b)}\right) \tag{4}$$

# 2 Supplementary Information for Concentration of Bilinear Functions

In this section, we prove the following theorem:

**Theorem 1.** *Consider any bilinear function $f_a(x) = \sum_{u,v} a_{uv} x_u x_v$ on an Ising model $p$ (defined on a graph $G = (V, E)$ such that $|V| = n$) in $\eta$-high-temperature regime with no external field. Let $\|a\|_\infty = \max_{u,v} a_{uv}$. If $X \sim p$, then for any $r \geq 300\|a\|_\infty n \log^2 n/\eta + 2$, we have*

$$\Pr\left[|f_a(X) - \mathbf{E}\left[f_a(X)\right]| \geq r\right] \leq 5 \exp\left(-\frac{\eta r}{1735\|a\|_\infty n \log n}\right).$$

Note that, for the sake of convenience in our proof, this theorem is stated for bilinear functions where all terms are of degree 2. One can immediately obtain concentration for all bilinear functions by combining this result with concentration bounds for linear functions. Since linear functions concentrate in a much tighter radius ($O(\sqrt{n})$, rather than $\tilde{O}(n)$), this comes at a minimal additional cost.

**Remark 1.** *We note that $\eta$-high-temperature is not strictly needed for our results to hold – we only need Hamming contraction of the "greedy coupling" (see Point 2 in Lemma 3). This condition implies rapid mixing of the Glauber dynamics (in $O(n \log n)$ steps) via path coupling (Theorem 15.1 of [LPW09]).*

The organization of this section is as follows. We will first focus on proving concentration for bilinear statistics with no external field. In Section 2.1, we state some additional preliminaries, and describe the martingale sequence and stopping time we will consider. In Section 2.2, we prove certain concentration properties of linear functions of the Ising model – in particular, these will be useful in showing that the stopping time is large. In Section 2.4, we show that our martingale sequence has bounded differences before the stopping time. In Section 2.5, we put the pieces together and prove bilinear concentration. In Section 2.6, we discuss how to prove concentration for bilinear statistics under an external field. Note that under an external field, not all bilinear functions of the Ising model concentrate, and thus our statistics require appropriate recentering. In Section 2.7, we briefly argue that the exponential behavior of the tail is inherent – for example, it could not be improved to a Gaussian tail.

## 2.1 Setup

We will consider functions where $\|a\|_\infty \leq 1$, Theorem 1 follows by a scaling argument. Let $a \in [-1,1]^{\binom{V}{2}}$ and define $f_a : \{\pm 1\}^V \to \mathbb{R}$ as follows:

$$f_a(x) = \sum_{u,v} a_{uv} x_u x_v.$$

The quantity of interest which we would like to bound is $\Pr[|f_a(X) - \mathbf{E}[f_a(X)]| > r]$ where $X \sim p$ is a sample from the Ising model $p$. For the time being, we will focus on the setting with *no external field* for ease of exposition[2].

A crucial quantity to the whole discussion will be $|f(X) - f(X')|$ where $X'$ is obtained by taking a single step of the Glauber dynamics associated with a high temperature Ising model $p$ starting from $X$. Define $f_a^u(X) = \sum_{u \neq v} a_{uv} X_u$. These $n$ linear functions $f_a^1(X), \ldots, f_a^n(X)$ will arise as a result of looking at $|f_a(X) - f_a(X')|$, as shown in the following claim:

**Claim 1.** *If $X'$ is obtained by taking a step of the Glauber dynamics starting from $X$, then*

$$\left|f_a(X) - f_a(X')\right| = \begin{cases} 0 & \text{w.p. } p_0 \\ 2\left|f_a^1(X)\right| & \text{w.p. } p_1 \\ \dots & \\ 2\left|f_a^n(X)\right| & \text{w.p. } p_n \end{cases}$$

*where $p_0 + \dots + p_n = 1$.*

*Proof.* In each step of the Glauber dynamics, a node $v$ is chosen uniformly at random and updated according to the distribution of $v$ conditioned on its neighbors under the Ising model. With some probability $p_0^v$, the dynamics leave node $v$ unchanged (i.e. update it to its current value $X_v$). In this scenario, $f_a(X) - f_a(X') = 0$. If, on the other hand, the dynamics flip the sign of node $v$, then $f_a(X) - f_a(X') = (\sum_{u \neq v} a_{uv} X_u)(2X_v)$. Since $X_v \in \{\pm 1\}$, $\left|(\sum_{u \neq v} a_{uv} X_u)(2X_v)\right| = 2\left|\sum_{u \neq v} a_{uv} X_u\right| = 2\left|f_a^v(X)\right|$. $\qquad\square$

Next, we define a martingale sequence associated with any bilinear function $f_a$ of the Ising model. A sufficiently strong tail inequality on the difference between the first and last terms of the martingale will get us very close to the desired concentration result.

**Definition 9.** *Let $t^* = 3t_{mix} = 3n\log n/\eta$. Let $X_0 \sim p$ be a sample from the Ising model $p$. Consider a walk of the Glauber dynamics starting at $X_0$ and running for $t^*$ steps: $X_0, X_1, \dots, X_{t^*}$. $X_{t^*}$ can be viewed as a function of all the random choices made by the dynamics up to that point. That is, $X_{t^*} = h(X_0, R_1, \dots, R_{t^*})$ where $R_i$ is a random variable representing the random choices made by the dynamics in step $i$. More precisely, $R_i$ represents the realization of the random choice of which node to (attempt to) update and a $Uniform([0,1])$ random variable (based upon which we decide whether or not to update the node's variable). Hence $f_a(X_{t^*}) = \tilde{f}_a(X_0, R_1, \dots, R_{t^*})$ where $\tilde{f}_a = f_a \circ h$. Consider the Doob martingale associated with $\tilde{f}_a$ defined on the probability space $(O, 2^O, P)$ where $O$ is the set of all possible values of the variables $X_0, X_1, X_2, \dots, X_{t^*}$ under the above described stochastic process and $P$ is the function which assigns probability to events in $2^O$ according to the underlying stochastic process. Also consider the increasing sequence of sub-$\sigma$-fields $\mathcal{F}_0 = 2^{O_0} \subset \mathcal{F}_1 = 2^{O_1} \subset \mathcal{F}_2 = 2^{O_2} \subset \dots \mathcal{F}_{t^*} = 2^{O_{t^*}} = 2^O$ where $O_i$ is the set of all possible values of the variables $X_0, X_1, X_2, \dots, X_i$. The terms in the martingale sequence are as follows.*

$$B_0 = \mathbf{E}[\tilde{f}_a(X_0, R_1, \dots, R_{t^*})|X_0] = \mathbf{E}\left[f_a(X_{t^*})|X_0\right]$$
$$\dots$$
$$B_i = \mathbf{E}[\tilde{f}_a(X_0, R_1, \dots, R_{t^*})|X_0, R_1, \dots, R_i] = \mathbf{E}\left[f_a(X_{t^*})|X_0, X_1, \dots, X_i\right] \qquad (5)$$
$$\dots$$
$$B_{t^*} = \tilde{f}_a(X_0, R_1, \dots, R_{t^*}) = \mathbf{E}\left[f_a(X_{t^*})|X_0, X_1, \dots, X_{t^*}\right] = f_a(X_{t^*})$$

*Since the dynamics are Markovian, we can also write $B_i$ as follows:*

$$B_i = \mathbf{E}[f_a(X_{t^*})|X_i] \quad \forall\, 0 \leq i \leq t^*.$$

Note that we deliberately choose to skip the term $\mathbf{E}[\tilde{f}_a(R_1, \dots, R_{t^*})]$ and start the martingale sequence at $\mathbf{E}[\tilde{f}_a(X_0, R_1, \dots, R_{t^*})|X_0]$ instead. This is crucial because it enables us to obtain strong bounds on the martingale increments. We have a good understanding over the behavior

of the difference in values of $f(X_{t^*})$ conditioned on $X_i$ versus $X_{i+1}$ but apriori we can't bound $|\mathbf{E}\left[f_a(X_{t^*})|X_0\right] - \mathbf{E}\left[f_a(X_{t^*})\right]|$.

At this point, we could try and apply Azuma's inequality by bounding the martingale increments $|B_{i+1} - B_i|$. However, these increments can be $\Omega(n)$ in magnitude which would yield a radius of concentration of $\approx n^{1.5}$ from Azuma's inequality. As was remarked earlier, this is weak and we will see how we can show a radius of concentration $\approx n$ by harnessing the fact that the martingale increments are rarely, if ever, of the order $\Omega(n)$. This is because of concentration of linear functions on the Ising model. To harness this fact, we appeal to Freedman's inequality (Lemma 5) and the first order of business in applying Freedman's inequality effectively is to define a stopping time on the martingale sequence such that two things hold:

1. The stopping time is larger than $t^*$ with high probability. Hence, with a good probability the process doesn't stop too early. The harm if the process stops too early (at $t < t^*$) is that we will not be able to effectively decouple $\mathbf{E}\left[f_a(X_t)|X_0\right]$ from the choice of $X_0$. $t^*$ was chosen to be larger than the mixing time of the Glauber dynamics precisely because it allows us to argue that $\mathbf{E}\left[f_a(X_{t^*})|X_0\right] \approx \mathbf{E}\left[f_a(X_{t^*})\right] = \mathbf{E}[f_a(X)]$.

2. For all $i + 1$ less than the stopping time, $|B_{i+1} - B_i| = O(\sqrt{n})$.

With the above criterion in mind, we define a stopping time $T_K$ on the martingale sequence.

**Definition 10.** *Consider the martingale sequence defined in Definition 9. Define the set $G_K^a(t)$ to be the following set of configurations:*

$$G_K^a(t) = \{x_t \in \Omega \mid |\mathbf{E}[f_a^v(X_{t^*})|X_t = x_t]| \leq K \ \text{and} \ |\mathbf{E}[f_a^v(X_{t^*-1})|X_t = x_t]| \leq K \ \forall v \in V\}$$

$$\bigcap \left\{ x_t \in \Omega \mid \Pr\left[|f_a^v(X_{t^*}) - \mathbf{E}\left[f_a^v(X_{t^*})|X_t\right]| > K|X_t = x_t\right] \leq 2\exp\left(-\frac{K^2}{16t^*}\right) \ \forall v \in V \right\}$$

$$\bigcap \left\{ x_t \in \Omega \mid \Pr\left[|f_a^v(X_{t^*-1}) - \mathbf{E}\left[f_a^v(X_{t^*-1})|X_t\right]| > K|X_t = x_t\right] \leq 2\exp\left(-\frac{K^2}{16t^*}\right) \ \forall v \in V \right\}$$

(6)

*where $\mathbf{E}[f_a^v(X_t)|X_{t_0}]$, for all $v \in V$, is defined as 0 for $t_0 > t$. Let $T_K : O \to \{0\} \bigcup \mathbb{N}$ be a stopping time defined as follows:*

$$T_K = \min\{t^* + 1, \min_{t \geq 0}\{t \mid t \notin G_K^a(t)\}\},$$

*Note that the event $\{T_K = t\}$ lies in the $\sigma$-field $2^{O_t}$ and hence the above definition is a valid stopping time.*

## 2.2  Properties of Linear Functions of the Ising Model

In this section, we prove the following lemma, concerned primarily with a particular type of concentration of linear functions on the Ising model.

**Lemma 6.** *Let $X_0$ be a sample from an Ising model $p$ at $\eta$-high temperature with no external field, and $X_t$ be obtained by taking $t$ steps along the Glauber dynamics corresponding to $p$ with the condition that the dynamics start at $X_0$. For any linear function $f(x) := \sum_{v \in V} a_v x_v$ such that*

$|a_v| \leq 1$, *define* $g^t(X_0) = \mathbf{E}[f(X_t)|X_0]$. *Then the following hold for any* $t \geq 0$,

$$\mathbf{E}[f(X_t)] = 0, \tag{7}$$

$$\Pr\left[\left|g^t(X_0)\right| > r\right] \leq 2\exp\left(-\frac{\eta r^2}{8n}\right), \tag{8}$$

$$\Pr\left[|f(X_t) - \mathbf{E}[f(X_t)|X_0]| > K\right] \leq 2\exp\left(-\frac{K^2}{4t}\right). \tag{9}$$

*Proof.* First, if $X_0 \sim p$, then since $p$ is the stationary distribution of the associated Glauber chain, $X_t \sim p$ as well. Hence, $\mathbf{E}[f(X_t)] = \mathbf{E}[f(X_0)] = \sum_{v \in V} a_v \mathbf{E}[X_{0,v}] = 0$ for all $t \geq 0$.

For showing the second property, we will first bound the Lipschitz constants of the function $g^t(.)$. We denote by $\vec{l}$ the vector of Lipschitz constants of $f(x)$. Since $f$ is a linear function, $\vec{l} = [2|a_1|, \ldots, 2|a_v|, \ldots, 2|a_n|]$. We have for any $x, x'$ such that $d_H(x, x') = 1$,

$$\left|g^t(x = x_1, \ldots, x_i, \ldots, x_n) - g^t(x' = x_1, \ldots, x_i', \ldots, x_n)\right| = \left|\mathbf{E}[f(X_t)|X_0 = x] - \mathbf{E}[f(X_t')|X_0' = x']\right|$$
$$= \left|\mathbf{E}[f(X_t) - f(X_t')|X_0 = x, X_0' = x']\right| \tag{10}$$
$$\leq \left|\mathbf{E}[(X_t - X_t') \cdot \vec{l}/2|X_0 = x, X_0' = x']\right|$$
$$\leq \mathbf{E}\left[2d_H(X_t, X_t')|X_0 = x, X_0' = x'\right] \tag{11}$$
$$\leq 2. \tag{12}$$

where (10) holds for any valid coupling of the two chains starting at $X_0$ and $X_0'$ respectively, in particular, we use the greedy coupling (Definition 4) here. (11) follows because $|a_i| \leq 1 \ \forall i$, and (12) follows because the expected Hamming distance between $X_t$ and $X_t'$, due to the contracting nature of the Glauber dynamics under the greedy coupling (Lemma 3), is smaller than $d_H(X_0, X_0')$ which is equal to 1. Also note that $\mathbf{E}[g^t(X_0)] = \mathbf{E}[f(X_t)] = 0$. Hence, applying Lemma 1 to $g^t(x)$, we get

$$\Pr\left[\left|g^t(X_0)\right| > r\right] \leq 2\exp\left(-\frac{\eta r^2}{8n}\right).$$

Note that we could apply Lemma 1 to $g^t(.)$ because $X_0$ was drawn from the stationary distribution of the Glauber dynamics.

To show the third property, we will define a martingale similar to the one defined in Definition 9 and apply Azuma's inequality to it. Consider a run of the Glauber dynamics starting at $X_0 \sim p$ and running for $t$ steps. We will view $X_t$ as a function of all the random choices made by the dynamics up to step $t^*$. That is, $X_t = h(X_0, R_1, \ldots, R_t)$ where $R_i$ denotes the random choices made by the dynamics during step $i$. More precisely, $R_i$ represents the realization of the random choice of which node to (attempt to) update and a $Uniform([0, 1])$ random variable (based upon which we decide whether or not to update the node's variable). Hence $f(X_t) = \tilde{f}(X_0, R_1, \ldots, R_t)$ where $\tilde{f} = f \circ h$. Consider the Doob martingale defined on $\tilde{f}$:

$$D_0 = \mathbf{E}[\tilde{f}(X_0, R_1, \ldots, R_t)|X_0]$$
$$\ldots$$
$$D_i = \mathbf{E}[\tilde{f}(X_0, R_1, \ldots, R_t)|X_0, R_1, \ldots, R_i] \tag{13}$$
$$\ldots$$
$$D_t = \tilde{f}(X_0, R_1, \ldots, R_t)$$

Since the dynamics are Markovian, we can also write $D_i$ as follows:

$$D_i = \mathbf{E}[f(X_t)|X_i].$$

Next we will bound the increments of the above martingale and apply Azuma's inequality to get the desired tail bound. In the following calculation, we will use the notation $x \to y$ where $x, y \in \{\pm 1\}^n$, to denote that $y$ is a possible transition according to a single step of the dynamics starting from $x$. For any $0 \le i \le t-1$,

$$|D_{i+1} - D_i| = |\mathbf{E}\left[f(X_t)|X_{i+1}\right] - \mathbf{E}\left[f(X_t)|X_i\right]| \tag{14}$$

$$\le \max_{x,y:d_H(x,y)=1} \left|\mathbf{E}\left[f(X_t)|X_{i+1}=x\right] - \mathbf{E}\left[f(X_t')|X_i'=y\right]\right| \tag{15}$$

$$= \max_{x,y:d_H(x,y)=1} \left|\mathbf{E}\left[f(X_t)|X_{i+1}=x\right] - \sum_{y':y\to y'} \Pr[y \to y']\mathbf{E}\left[f(X_t')|X_{i+1}'=y'\right]\right| \tag{16}$$

$$\le \max_{x,y':d_H(x,y')\le 2} \left|\mathbf{E}\left[f(X_t)|X_{i+1}=x\right] - \mathbf{E}\left[f(X_t')|X_{i+1}'=y'\right]\right| \tag{17}$$

$$= \max_{x,y':d_H(x,y')\le 2} \left|\mathbf{E}\left[f(X_t) - f(X_t')|X_{i+1}=x, X_{i+1}'=y'\right]\right| \tag{18}$$

$$= \max_{x,y':d_H(x,y')\le 2} \left|\mathbf{E}\left[\sum_v a_v(X_{t,v} - X_{t,v}')|X_{i+1}=x, X_{i+1}'=y'\right]\right|$$

$$\le \max_{x,y':d_H(x,y')\le 2} \mathbf{E}\left[2d_H(X_t, X_t')|X_{i+1}=x, X_{i+1}'=y'\right] \le 2. \tag{19}$$

where in (15) we relabeled the variables in the second expectation to avoid notational confusion in the later steps of our bounding, maintaining the understanding that the sequence $\{X_i', Y_i'\}_i$ has the same distribution as $\{X_i, Y_i\}_i$, (18) holds for any valid coupling of the $X_i$ and $X_i'$ chains, in particular, it holds for the greedy coupling between the runs (Definition 4). (19) follows from the condition $|a_v| \le 1$ and the contracting nature of the Glauber dynamics (Lemma 3) under the greedy coupling.

Hence, for all $0 \le i \le t-1$, $|D_{i+1} - D_i| \le 2$. Azuma's inequality applied on the martingale sequence $\{D_i\}_{i\ge 0}$ yields

$$\Pr\left[|D_t - D_0| > K\right] \le 2\exp\left(-\frac{K^2}{4t}\right)$$

$$\implies \Pr\left[|f(X_t) - \mathbf{E}[f(X_t)|X_0]| > K\right] \le 2\exp\left(-\frac{K^2}{4t}\right)$$

$\square$

## 2.3 Showing that the Martingale Process Doesn't Stop Too Early

As a consequence of Lemma 6, we can show that with sufficiently large probability the stopping time $T_K$ defined above is larger than $t^*$, and thus Freedman's inequality gives guarantees for all $t$ up to $t^*$ with high probability. The main lemma will be the following one, which shows that for any $t$, $X_t \in G_K^a(t)$ with high probability.

**Lemma 7.** *For any $t \ge 0$, for $t^* = 3t_{mix}$,*

$$\Pr\left[X_t \notin G_K^a(t)\right] \le 8n\exp\left(-\frac{K^2}{8t^*}\right).$$

*Proof.* Since $X_0$ is a sample from the stationary distribution $p$ of the dynamics, it follows from the property of stationary distributions that $X_t$ is also a sample from $p$. Hence we have, from Lemma 6, and a union bound, that

$$\Pr\left[\exists\, v \in V \text{ s.t. } \max\{|\mathbf{E}\left[f_a^v(X_{t^*})|X_t\right]|, |\mathbf{E}\left[f_a^v(X_{t^*-1})|X_t\right]|\} > K\right] \le 4n \exp\left(-\frac{\eta K^2}{8n}\right). \qquad (20)$$

Let $D_K^a(t)$ be the event defined as

$$D_K^a(t) = \{\exists\, v \in V \text{ s.t. } \max\{|f_a^v(X_{t^*}) - \mathbf{E}\left[f_a^v(X_{t^*})|X_t\right]|, |f_a^v(X_{t^*-1}) - \mathbf{E}\left[f_a^v(X_{t^*-1})|X_t\right]|\} > K\}.$$

From Lemma 6, and a union bound, we have,

$$\Pr\left[D_K^a(t)\right] = \mathbf{E}\left[\Pr\left[D_K^a(t)|X_t\right]\right] \le 4n \exp\left(-\frac{K^2}{4t^*}\right)$$

$$\implies \Pr\left[\Pr\left[D_K^a(t)|X_t\right] > \exp\left(-\frac{K^2}{8t^*}\right)\right] \le 4n \exp\left(-\frac{K^2}{8t^*}\right) \qquad (21)$$

where (21) follows from a simple application of Markov's inequality. Hence,

$$\Pr\left[\exists\, v \in V \text{ s.t. } \max\{\Pr\left[|f_a^v(X_{t^*}) - \mathbf{E}\left[f_a^v(X_{t^*})|X_t\right]| > K\right], \Pr\left[|f_a^v(X_{t^*-1}) - \mathbf{E}\left[f_a^v(X_{t^*-1})|X_t\right]| > K\right]\} > \exp\left(-\frac{K^2}{8t^*}\right)\right]$$

$$\le \Pr\left[\Pr\left[D_K^a(t)|X_t\right] > \exp\left(-\frac{K^2}{8t^*}\right)\right] \le 4n \exp\left(-\frac{K^2}{8t^*}\right). \qquad (22)$$

From (20) and (22), we have,

$$\Pr\left[X_t \notin G_K^a(t)\right] \le 4n \exp\left(-\frac{K^2}{8t^*}\right) + 4n \exp\left(-\frac{\eta K^2}{8n}\right) \le 8n \exp\left(-\frac{K^2}{8t^*}\right).$$

$\square$

Given Lemma 7, the proof of the stopping time being large with high probability follows by a simple application of the union bound.

**Lemma 8.** *For $t^* = 3t_{mix}$,*

$$\Pr\left[t^* \ge T_K\right] \le 8nt^* \exp\left(-\frac{K^2}{8t^*}\right).$$

*Proof.* From Lemma 7, we have

$$\Pr\left[X_t \notin G_K^a(t)\right] \le 8n \exp\left(-\frac{K^2}{8t^*}\right)$$

$$\implies \Pr[t^* \ge T_K] = \Pr\left[\bigcup_{t=0}^{t^*} X_t \notin G_K^a(t)\right]$$

$$\le \sum_{t=0}^{t^*} \Pr\left[X_t \notin G_K^a(t)\right] \le 8t^* n \exp\left(-\frac{K^2}{8t^*}\right)$$

$$= 8nt^* \exp\left(-\frac{K^2}{8t^*}\right).$$

$\square$

## 2.4 Bounding the Martingale Differences

In this section, we prove a bound on the increments $B_{i+1} - B_i$ for the Doob martingale defined in Definition 9 which holds with probability 1 for all $i + 1 < T_K$. We begin by showing a bound which holds pointwise when $X_i \in G_K^a(i)$ and $X_{i+1} \in G_K^a(i+1)$ in the form of Lemma 9.

**Lemma 9.** *Consider the Doob martingale defined in Definition 9. Suppose $X_i \in G_K^a(i)$ and $X_{i+1} \in G_K^a(i+1)$. Then*

$$|B_{i+1} - B_i| \leq 16K + 16n^2 \exp\left(-\frac{K^2}{16t^*}\right).$$

*Proof.* For ease of exposition, we will refer to $G_K^a(i)$ as simply $G_i$ in the following proof.

$$|B_{i+1} - B_i| = |\mathbf{E}\left[f_a(X_{t^*})|X_{i+1}\right] - \mathbf{E}\left[f_a(X_{t^*})|X_i\right]| \tag{23}$$

$$= \left|\mathbf{E}\left[f_a(X_{t^*})|X_{i+1}\right] - \mathbf{E}\left[f_a(X'_{t^*})\big|X'_i\right]\right| \tag{24}$$

$$\leq \left|\mathbf{E}\left[f_a(X_{t^*})|X_{i+1}\right] - \mathbf{E}\left[f_a(X'_{t^*-1})\big|X'_i\right]\right| + \left|\mathbf{E}\left[f_a(X'_{t^*-1})\big|X'_i\right] - \mathbf{E}\left[f_a(X'_{t^*})\big|X'_i\right]\right| \tag{25}$$

$$\leq \left|\mathbf{E}\left[f_a(X_{t^*}) - f_a(X'_{t^*-1})\big|X_{i+1}, X'_i\right]\right| + \left|\mathbf{E}\left[f_a(X'_{t^*}) - f_a(X'_{t^*-1})\big|X'_i\right]\right| \tag{26}$$

$$= \left|\mathbf{E}\left[\sum_v (X_{t^*,v} - X'_{t^*-1,v})\left(\sum_u a_{uv}X_{t^*,u}\right) + \sum_u (X_{t^*,u} - X'_{t^*-1,u})\left(\sum_v a_{uv}X'_{t^*-1,v}\right)\Big|X_{i+1}, X'_i\right]\right|$$

$$+ \left|\mathbf{E}\left[f_a(X'_{t^*}) - f_a(X'_{t^*-1})\big|X'_i\right]\right| \tag{27}$$

$$\leq \mathbf{E}\left[\sum_v \left|X_{t^*,v} - X'_{t^*-1,v}\right|\left|\sum_u a_{uv}X_{t^*,u}\right| + \sum_u \left|X_{t^*,u} - X'_{t^*-1,u}\right|\left|\sum_v a_{uv}X'_{t^*-1,v}\right|\Big|X_{i+1}, X'_i\right]$$

$$+ \left|\mathbf{E}\left[f_a(X'_{t^*}) - f_a(X'_{t^*-1})\big|X'_i\right]\right| \tag{28}$$

where in (24) we relabeled the variables in the second expectation to avoid notational confusion in the later steps of our bounding, maintaining the understanding that the sequence $\{X'_i, Y'_i\}_i$ has the same distribution as $\{X_i, Y_i\}_i$, in (25) we added and subtracted the term $\mathbf{E}[f_a(X'_{t^*-1})|X'_i]$, (26) holds for any valid coupling of the two chains, one starting at $X_{i+1}$ and the other starting at $X'_i$, and both running for $t^* - 1 - i$ steps. In particular, we use the greedy coupling between these two runs (Definition 4). Consider the first term in (28). Since $X_{i+1} \in G_{i+1}$, we have

$$\left|\mathbf{E}\left[\sum_u a_{uv}X_{t^*,u}\Big|X_{i+1}\right]\right| \leq K \text{ and} \tag{29}$$

$$\Pr\left[\left|\sum_u a_{uv}X_{t^*,u} - \mathbf{E}\left[\sum_u a_{uv}X_{t^*,u}\Big|X_{i+1}\right]\right| > K \Big|X_{i+1}\right] \leq 2\exp\left(-\frac{K^2}{16t^*}\right). \tag{30}$$

$$\implies \Pr\left[\left|\sum_u a_{uv}X_{t^*,u}\right| > 2K \Big|X_{i+1}\right] \leq 2\exp\left(-\frac{K^2}{16t^*}\right). \tag{31}$$

where (29) and (30) together imply (31). Similarly, we also get that

$$\Pr\left[\left|\sum_v a_{uv}X'_{t^*-1,v}\right| > 2K \Big|X'_i\right] \leq 2\exp\left(-\frac{K^2}{16t^*}\right). \tag{32}$$

Since $\sum_v \left| X_{t^*,v} - X'_{t^*-1,v} \right| \left| \sum_u a_{uv} X_{t^*,u} \right| \le 2n^2$ for all $X_{t^*}, X'_{t^*-1}$, and since $\sum_v \left| X_{t^*,v} - X'_{t^*-1,v} \right| = 2d_H(X_{t^*}, X'_{t^*-1})$, (31) and (32) imply that

$$\mathbf{E}\left[ \sum_v \left| X_{t^*,v} - X'_{t^*-1,v} \right| \left| \sum_u a_{uv} X_{t^*,u} \right| + \sum_u \left| X_{t^*,u} - X'_{t^*-1,u} \right| \left| \sum_v a_{uv} X'_{t^*-1,v} \right| \,\middle|\, X_{i+1}, X'_i \right] \tag{33}$$

$$\le \mathbf{E}\left[ 4d_H(X_{t^*}, X'_{t^*-1})K + 4d_H(X_{t^*}, X'_{t^*-1})K \,\middle|\, X_{i+1}, X'_i \right] + 8n^2 \exp\left( -\frac{K^2}{16t^*} \right) \tag{34}$$

$$\le 8K + 8n^2 \exp\left( -\frac{K^2}{16t^*} \right), \tag{35}$$

where (35) follows because of the contracting nature of Hamming distance under the greedy coupling of the Glauber dynamics (Lemma 3).

The same bound can be proven by following the same steps for the second term in (28), completing the proof. □

As a consequence of Lemma 9, we get the following two useful corollaries.

**Corollary 1.** *Consider the martingale sequence defined in Definition 9.*

$$\Pr\left[ \forall \, 0 < i+1 < T_K, \, |B_{i+1} - B_i| \le 16K + 16n^2 \exp\left( -\frac{K^2}{16t^*} \right) \right] = 1.$$

*Proof.* Let $\kappa = 16K + 16n^2 \exp\left( -\frac{K^2}{16t^*} \right)$.

$$\Pr\left[ \forall \, 0 < i+1 < T_K, \, |B_{i+1} - B_i| \le \kappa \right]$$
$$= 1 - \Pr\left[ \exists \, 0 < i+1 < T_K, \, |B_{i+1} - B_i| > \kappa \right]$$
$$= 1 - \Pr\left[ \exists \, 0 < i+1 < T_K, \, (X_i \in G^a_K(i), X_{i+1} \in G^a_K(i+1) \text{ and } |B_{i+1} - B_i| > \kappa) \right.$$
$$\left. \quad \text{or } ((X_i \notin G^a_K(i) \text{ or } X_{i+1} \notin G^a_K(i+1)) \text{ and } |B_{i+1} - B_i| > \kappa) \right]$$
$$= 1 - \Pr\left[ \exists \, 0 < i+1 < T_K, \, (X_i \in G^a_K(i), X_{i+1} \in G^a_K(i+1) \text{ and } |B_{i+1} - B_i| > \kappa) \right] \tag{36}$$
$$= 1 - 0 \tag{37}$$

where (36) follows because by the definition of $T_K$, $\forall 0 < i+1 < T_K, (X_i \in G^a_K(i) \text{ and } X_{i+1} \notin G^a_K(i+1))$, and (37) follows because $X_i \in G^a_K(i), X_{i+1} \in G^a_K(i+1) \implies |B_{i+1} - B_i| \le \kappa$ (Lemma 9). □

Corollary 1, will give us one of the required conditions to apply Freedman's inequality.

As a corollary of Lemma 9, we get a bound on the variance of the martingale differences which holds with high probability. To show it we first show Claim 2 which states that, informally, for any time step $i$, with a large probability we hit an $X_i$ such that the probability of transitioning from $X_i$ to an $X_{i+1} \in G^a_K(i+1)$ is large.

**Claim 2.** *Denote by $N^a_K(i)$ the following set of configurations:*

$$N^a_K(i) = \left\{ x_i \in \Omega \,\middle|\, \Pr\left[ X_{i+1} \notin G^a_K(i+1) | X_i = x_i \right] \le \exp\left( -\frac{K^2}{16t^*} \right) \right\}. \tag{38}$$

*Then,*

$$\Pr\left[ X_i \notin N^a_K(i) \right] \le 8n \exp\left( -\frac{K^2}{16t^*} \right).$$

*Proof.* We have from Lemma 7, that

$$\Pr\left[X_i \notin G_K^a(i)\right] \le 8n \exp\left(-\frac{K^2}{8t^*}\right) \text{ and} \tag{39}$$

$$\Pr\left[X_{i+1} \notin G_K^a(i+1)\right] \le 8n \exp\left(-\frac{K^2}{8t^*}\right). \tag{40}$$

From the definition of the set $N_K^a(i)$ we have,

$$\Pr\left[X_{i+1} \in G_K^a(i) | X_i \notin N_K^a(i)\right] \le 1 - \exp\left(-\frac{K^2}{16t^*}\right). \tag{41}$$

Then we have,

$$1 - 8n \exp\left(-\frac{K^2}{8t^*}\right) \le \Pr\left[X_{i+1} \in G_K^a(i)\right] \tag{42}$$

$$= \Pr\left[X_{i+1} \in G_K^a(i) | X_i \in N_K^a(i)\right] \Pr\left[X_i \in N_K^a(i)\right] + \Pr\left[X_{i+1} \in G_K^a(i) | X_i \notin N_K^a(i)\right] \Pr\left[X_i \notin N_K^a(i)\right] \tag{43}$$

$$\le \Pr\left[X_i \in N_K^a(i)\right] + \left(1 - \exp\left(-\frac{K^2}{16t^*}\right)\right) \Pr\left[X_i \notin N_K^a(i)\right] \tag{44}$$

$$= \left(1 - \exp\left(-\frac{K^2}{16t^*}\right)\right) + \exp\left(-\frac{K^2}{16t^*}\right) \Pr\left[X_i \in N_K^a(i)\right]. \tag{45}$$

(45) implies,

$$\Pr\left[X_i \in N_K^a(i)\right] \ge \frac{\exp\left(-\frac{K^2}{16t^*}\right) - 8n \exp\left(-\frac{K^2}{8t^*}\right)}{\exp\left(-\frac{K^2}{16t^*}\right)} \tag{46}$$

$$= 1 - 8n \exp\left(-\frac{K^2}{16t^*}\right). \tag{47}$$

$\square$

**Lemma 10.** *Consider the martingale sequence defined in Definition 9. Let* $b = \left(16K + 16n^2 \exp\left(-\frac{K^2}{16t^*}\right)\right)^2 + n^4 \exp\left(-\frac{K^2}{16t^*}\right)$. *Denote by* $N_K^a(i)$ *the following set of configurations (as was defined in Claim 2):*

$$N_K^a(i) = \left\{x_i \in \Omega \,\middle|\, \Pr\left[X_{i+1} \notin G_K^a(i+1) | X_i = x_i\right] \le \exp\left(-\frac{K^2}{16t^*}\right)\right\}. \tag{48}$$

*Then,*

$$\Pr\left[\mathbf{Var}[B_{i+1} - B_i | \mathcal{F}_i] > b | X_i \in G_K^a(i) \cap N_K^a(i)\right] = 0.$$

*where* $\mathcal{F}_i = 2^{O_i}$.

*Proof.* Since, the random variables $X_0, \ldots, X_i$ together characterize every event in $\mathcal{F}_i$, we have,

$$\mathbf{Var}[B_{i+1} - B_i | \mathcal{F}_i] = \mathbf{Var}[B_{i+1} - B_i | X_0, X_1, \ldots, X_i] = \mathbf{Var}[B_{i+1} - B_i | X_i] \tag{49}$$

where the last equality follows from the Markov property of the Glauber dynamics. By the definition of $N_K^a(i)$, we have that

$$\mathbf{Var}\left[B_{i+1} - B_i | X_i \in G_K^a(i) \cap N_K^a(i)\right] = \mathbf{E}\left[\left(B_{i+1} - B_i - \mathbf{E}\left[B_{i+1} - B_i | X_i \in G_K^a(i) \cap N_K^a(i)\right]\right)^2 \Big| X_i \in G_K^a(i) \cap N_K^a(i)\right] \tag{50}$$

$$= \mathbf{E}\left[(B_{i+1} - B_i)^2 \Big| X_i \in G_K^a(i) \cap N_K^a(i)\right] \tag{51}$$

$$\leq \mathbf{E}\left[(B_{i+1} - B_i)^2 \Big| X_i \in G_K^a(i) \cap N_K^a(i) \cap X_{i+1} \in G_K^a(i+1)\right]$$

$$\quad + \mathbf{E}\left[(B_{i+1} - B_i)^2 \Big| X_i \in G_K^a(i) \cap N_K^a(i) \cap X_{i+1} \notin G_K^a(i+1)\right] \Pr\left[X_{i+1} \notin G_K^a(i+1) | X_i \in G_K^a(i) \cap N_K^a(i)\right]$$

$$\leq \left(16K + 16n^2 \exp\left(-\frac{K^2}{16t^*}\right)\right)^2 + 4n^4 \Pr\left[X_{i+1} \notin G_K^a(i+1) | X_i \in G_K^a(i) \cap N_K^a(i)\right] \tag{52}$$

$$\leq \left(16K + 16n^2 \exp\left(-\frac{K^2}{16t^*}\right)\right)^2 + 4n^4 \exp\left(-\frac{K^2}{16t^*}\right), \tag{53}$$

where (51) holds because $\mathbf{E}\left[B_{i+1} - B_i | X_i = x_i\right] = 0$ for all $x_i$ since $\{B_i\}_{i \geq 0}$ is a martingale, (52) holds because $\Pr[|B_{i+1} - B_i| \leq 16K + 16n^2 \exp\left(-\frac{K^2}{16t^*}\right) | X_i \in G_K^a(i) \cap X_{i+1} \in G_K^a(i+1)] = 1$ and the maximum $\mathbf{E}[(B_{i+1} - B_i)^2]$ can be is at most $2n^2$, (53) follows from Claim 2. The last inequality implies the statement of the lemma. $\qquad\square$

## 2.5 Applying Freedman's Inequality and Completing the Proof

With Lemma 9 and Lemma 10 to bound the martingale increments, and Lemma 8 to show that the stopping time is large, we are ready to apply Freedman's inequality on the martingale defined in Definition 9.

**Lemma 11.** *For all $r \geq 300n \log^2 n / \eta$,*

$$\Pr\left[|f_a(X_{t^*}) - \mathbf{E}\left[f_a(X_{t^*})|X_0\right]| \geq r\right] \leq 5 \exp\left(-\frac{\eta r}{1734n \log n}\right).$$

*Proof.* From Freedman's inequality (Lemma 5) applied on the martingale sequence (Definition 9), we get

$$\Pr\left[\exists t < T_K \text{ s.t. } |B_t - B_0| \geq r \text{ and } V_t \leq B\right] \leq 2 \exp\left(-\frac{r^2}{2(rK_1 + B)}\right) \tag{54}$$

where $K_1 = 16K + 16n^2 \exp\left(-\frac{K^2}{16t^*}\right)$ (Lemma 9) and $V_t$ is defined as follows:

$$V_t = \sum_{i=0}^{t-1} \mathbf{Var}\left[B_{i+1} - B_i | \mathcal{F}_i\right]. \tag{55}$$

Set $B = t^*\left(16K + 16n^2 \exp\left(-\frac{K^2}{16t^*}\right)\right)^2 + 4t^*n^4 \exp\left(-\frac{K^2}{16t^*}\right)$ and $K = \sqrt{r}$. Next, we note that if $r > 2n^2$, the statement of the theorem holds vacuously. From now on we handle the case when $r \leq 2n^2$.

**Proposition 1.** *If $2n^2 \geq r > 64n \log^2 n / \eta$, then $rK_1 + B \leq 867rn \log n / \eta$.*

*Proof.* We note that $n \leq r \leq n^2$: the former is in the condition of the proposition statement, and the latter is since the lemma is trivial for $r > n^2$. Since $K = \sqrt{r}$, this implies $\sqrt{n} \leq K \leq n$.

We first focus on $rK_1$. Since $r \geq 24n \log^2 n/\eta$, we have that $16n^2 \exp\left(-\frac{K^2}{16t^*}\right) \leq 16\sqrt{n}$, and therefore $K_1 \leq 16K + 16\sqrt{n} \leq 32K$, and thus $rK_1 \leq 32rK \leq 32rt^*$, where the latter inequality follows since $r \leq 2n^2$ while $t^* \geq n \log n$.

By a similar calculation, we have that $B \leq t^* \left(8K + 8\sqrt{n}\right)^2 + t^* \leq 257K^2 t^* = 257rt^*$.

Adding the two, we have that $rK_1 + B \leq 289rt^* = 867rn \log n/\eta$, as desired. $\qquad\square$

Hence, (54) becomes

$$\Pr\left[\exists t < T_K \text{ s.t. } |B_t - B_0| \geq r \text{ and } V_t \leq B\right] \leq 2\exp\left(-\frac{r^2}{2(rK_1 + B)}\right) \tag{56}$$

$$\leq 2\exp\left(-\frac{\eta r}{1734 n \log n}\right). \tag{57}$$

Next we will bound, $\Pr\left[V_{t^*} > B\right]$ which will be useful for obtaining the desired concentration bound from (57).

$$\Pr\left[V_{t^*} > B\right] \leq \Pr\left[V_{t^*} > B \mid \forall\, 0 \leq t \leq t^*\ X_t \in G_K^a(t) \cap N_K^a(t)\right] + \Pr\left[\exists\, 0 \leq t \leq t^*\ X_t \notin G_K^a(t) \cap N_K^a(t)\right]$$

$$\leq \Pr\left[\exists\, 0 \leq t < t^* \text{ s.t. } \mathbf{Var}\left[B_{t+1} - B_t \mid X_t\right] > \frac{B}{t^*} \middle| \forall\, 0 \leq t \leq t^*\ X_t \in G_K^a(t) \cap N_K^a(t)\right] \tag{58}$$

$$+ \sum_{t=0}^{t^*} \left(\Pr\left[X_t \notin G_K^a(t)\right] + \Pr\left[X_t \notin N_K^a(t)\right]\right) \tag{59}$$

$$\leq 0 + t^*\left(8n\exp\left(-\frac{K^2}{8t^*}\right) + 8n\exp\left(-\frac{K^2}{16t^*}\right)\right) \leq \frac{48n^2 \log n}{\eta} \exp\left(-\frac{K^2}{16t^*}\right). \tag{60}$$

where (58) holds because $V_{t^*} > B$ implies that there exists a $0 \leq t \leq t^*$ such that $\mathbf{Var}\left[B_{t+1} - B_t \mid X_t\right] > B/t^*$, (59) follows by an application of the union bound, and (60) follows from Lemma 10, Lemma 7 and Claim 2.

Now,

$$\Pr\left[|f(X_{t^*}) - \mathbf{E}\left[f(X_{t^*})|X_0\right]| > r\right] = \Pr\left[|B_{t^*} - B_0| > r\right]$$

$$\leq \Pr\left[|B_{t^*} - B_0| > r \text{ and } V_{t^*} \leq B\right] + \Pr\left[V_{t^*} > B\right] \tag{61}$$

$$\leq \Pr\left[|B_{t^*} - B_0| > r \text{ and } V_{t^*} \leq B \text{ and } t^* < T_K\right] + \Pr[t^* \geq T_K] + \Pr\left[V_{t^*} > B\right] \tag{62}$$

$$\leq \Pr\left[(\exists t \leq t^* \text{ s.t. } |B_t - B_0| > r \text{ and } V_t \leq B) \text{ and } t^* < T_K\right] + \Pr[t^* \geq T_K] + \Pr\left[V_{t^*} > B\right] \tag{63}$$

$$\leq \Pr\left[\exists t < T_K \text{ s.t. } |B_t - B_0| > r \text{ and } V_t \leq B\right] + \Pr[t^* \geq T_K] + \Pr\left[V_{t^*} > B\right]$$

$$\leq 2\exp\left(-\frac{\eta r}{1734 n \log n}\right) + \frac{24n^2 \log n}{\eta} \exp\left(-\frac{\eta r}{24n \log n}\right) + \Pr\left[V_{t^*} > B\right] \tag{64}$$

$$\leq 3\exp\left(-\frac{\eta r}{1734 n \log n}\right) + \Pr\left[V_{t^*} > B\right] \tag{65}$$

$$\leq 3\exp\left(-\frac{\eta r}{1734 n \log n}\right) + \frac{48n^2 \log n}{\eta} \exp\left(-\frac{\eta r}{48n \log n}\right) \tag{66}$$

$$\leq 5\exp\left(-\frac{\eta r}{1734 n \log n}\right). \tag{67}$$

where (61) and (62) follow from the fact that $\Pr[A] \leq \Pr[A \cap B] + \Pr[\neg B]$, (63) follows from the fact that $\Pr[A] \leq \Pr[A \cup B]$, (64) follows from (57) and from Lemma 8, (65) holds because

$r \geq 300n \log^2 n/\eta$, (66) follows from (60) and (67) again holds because $r \geq 300n \log^2 n/\eta$. Note that we have implicitly assumed that $\eta > 1/n$, since otherwise the concentration bounds obtained are trivial.

$\square$

We note that this statement conditions on an initial state $X_0$. In order to remove this conditioning, we must argue that after $t^*$ steps, the Glauber dynamics have mixed, and $X_{t^*}$ is very close in total variation distance to the true Ising model, for any starting point $x_0$.

We use Lemma 2 to remove the conditioning in our previous tail bound, which implies Theorem 1.

**Lemma 12.** *For all $r \geq 300n \log^2 n/\eta + 2$ and $n$ sufficiently large,*

$$\Pr\left[|f_a(X_{t^*}) - \mathbf{E}\left[f_a(X_{t^*})\right]| \geq r\right] \leq 5 \exp\left(-\frac{\eta r}{1735n \log n}\right).$$

*Proof.* Since $X_0 \sim p$, we have that, $X_{t^*} \sim p$ as well. From Lemma 2, we have that for $t^* = 3t_{\mathrm{mix}}$,

$$d_{\mathrm{TV}}(X_{t^*}|X_0, X_{t^*}) \leq \exp\left(-2n \log n\right)$$
$$\implies |\mathbf{E}\left[f_a(X_{t^*})|X_0\right] - \mathbf{E}\left[f_a(X_{t^*})\right]| \leq 2n^2 \exp(-2n \log n) \leq 2 \exp(-n).$$

Now,

$$\Pr\left[|f_a(X_{t^*}) - \mathbf{E}\left[f_a(X_{t^*})\right]| > ger\right] \leq \Pr\left[|f_a(X_{t^*}) - \mathbf{E}\left[f_a(X_{t^*})|X_0\right]| \geq r - 2 \exp(-n)\right]$$
$$\leq 5 \exp\left(-\frac{\eta(r-2)}{1734n \log^2 n}\right)$$
$$\leq 5 \exp\left(-\frac{\eta r}{1735n \log^2 n}\right), \tag{68}$$

where (68) holds for sufficiently large $n$. $\square$

## 2.6 Concentration under an External Field

Under an external field, not all bilinear functions concentrate nicely even in the high temperature regime. This can be seen easily, for instance, in the case of $f_a(X) = \sum_{u \neq v} X_u X_v$. On an empty graph with a uniform external field $h$ on each node, $\mathbf{Var}(f_a(X)) = c(h)n^3$ (where $c(\cdot)$ is a function depending only on $h$). Hence the best scale of concentration one could hope for is at a distance $n^{1.5}$ from $\mathbf{E}[f_a(X)]$. However, tighter concentration akin to the one we achieve when there is no external field can be shown for classes of appropriately centered bilinear functions. We briefly describe the reason a non-centered function such as the one above doesn't concentrate and then argue at a high level how a correctly 'centered' function has sharper tails. To see where our framework fails when trying to show concentration of measure for arbitrary bilinear functions, let us look at $f_a(X) = \sum_{u \neq v} X_u X_v$. Under an external field, the linear functions associated with taking a step along the censored Glauber dynamics starting at $X$, are no longer zero mean. Although linear functions still concentrate around their expectation with a radius of $\tilde{O}(\sqrt{n})$, the expectations can be of the order $\Omega(n)$. Hence we *can't* use concentration of linear functions to argue that $|f_a^v(X)| \approx O(\sqrt{n})$. And the example described above shows that indeed the variance is higher for this function and the best concentration of measure one could hope to show has tails bounds which kick in at deviations of $O(n^{1.5})$ from the mean. To get stronger tails, the fix is to center our bilinear functions so that the linear functions arising from $f_a(X) - f_a(X')$ are zero mean thereby

enabling application of concentration at radius $\tilde{O}(\sqrt{n})$ on the quantity $|f_a^v(X) - \mathbf{E}[f_a^v(X)]|$ rather than having to bound $|f_a^v(X)|$ itself. There are multiple ways to achieve this. We present two simple and natural ways of doing so in Theorem 2.

**Theorem 2** (Concentration of Measure for Bilinear Functions Under an External Field). *1. Bilinear functions on the Ising model of the form $f_a(X) = \sum_{u,v} a_{uv}(X_u - \mathbf{E}[X_u])(X_v - \mathbf{E}[X_v])$ satisfy the following inequality at high temperature. There exist absolute constants $c$ and $c'$ such that, for $r \geq cn \log^2 n/\eta$,*

$$\Pr\left[|f_a(X) - \mathbf{E}[f_a(X)]| \geq r\right] \leq 4\exp\left(-\frac{r}{c'n \log n}\right).$$

2. *Bilinear functions on the Ising model of the form $f_a(X^{(1)}, X^{(2)}) = \sum_{u,v} a_{uv}(X_u^{(1)} - X_u^{(2)})(X_v^{(1)} - X_v^{(2)})$, where $X^{(1)}, X^{(2)}$ are two i.i.d samples from the Ising model, satisfy the following inequality at high temperature. There exist absolute constants $c$ and $c'$ such that, for $r \geq cn \log^2 n/\eta$,*

$$\Pr\left[\left|f_a(X^{(1)}, X^{(2)}) - \mathbf{E}[f_a(X^{(1)}, X^{(2)})]\right| \geq r\right] \leq 4\exp\left(-\frac{r}{c'n \log n}\right).$$

*Proof.* Since most of the proof follows along similar lines as that of the case with no external field, we briefly sketch the outline and highlight the major differences here. The calculations are straightforward to verify. The first step would be to prove a version of Lemma 6 for linear functions in the case of external field with the main difference being that wherever we had $f(x) = \sum_v a_v x_v$ before, we replace it with $f(x) = \sum_v a_v(x_v - \mathbf{E}[X_v])$. With this replacement it can be seen that the lemma follows in the presence of an external field. Next, we proceed to define a martingale sequence in the same way as was done in the case without external field. The linear functions in the stopping time definition are now replaced with their centered versions (i.e. are made zero mean). When studying the martingale differences we end up having to bound the difference in the expected value of our function at some future time $t^*$, conditioning on starting at two different starting states $X$ and $X'$, where $X'$ is obtained by doing one step of the Glauber dynamics from $X$. For the first style of centered functions listed in the theorem statement, if we unravel our bounding procedure we end up needing to bound functions of the form $\left|2\sum_{u \neq v} a_{uv}(X_u - \mathbf{E}[X_u])\right|$ for different $v$'s. The linear function inside the absolute value is zero mean and hence we can bound it in absolute value with high probability using concentration of measure for linear functions. Similarly, for the second style of functions in the theorem statement, we end up needing to bound functions of the form $\left|2\sum_{u \neq v} a_{uv}(X_u^{(1)} - X_u^{(2)})\right|$ which again are zero mean, and we can still use concentration of measure of linear functions to bound them. The rest of the proof follows in the same way as in the case without external field. □

## 2.7 An Exponential Tail is Inherent for Bilinear Statistics

In this section, we show that our tail bound of Theorem 1 is asymptotically tight upto a $\log n$ factor in the radius of concentration. Informally this means that exponential tails are the best one could hope to get for bilinear functions and sharper tails, (e.g. a Gaussian tail: $\exp(-r^2/n^2)$), can't be obtained. The tightness will follow from the following theorem which shows that the tail given by the Chernoff bound is asymptotically tight for sums of bounded i.i.d. random variables.

**Theorem 3.** *(Folklore) Let $X_1, \ldots, X_n$ be i.i.d. samples from $Ber(1/2)$ and let $g(X) = \sum_{i=1}^n X_i$. Then for any $r > 0$,*

$$\Pr\left[g(X) - \mathbf{E}[g(X)] > r\right] \geq \exp\left(-\frac{9r^2}{n}\right),$$

$$\Pr\left[g(X) - \mathbf{E}[g(X)] < -r\right] \geq \exp\left(-\frac{9r^2}{n}\right).$$

One possible proof of the above theorem follows by the application of Stirling's inequalities. Now, consider the bilinear function $f(X) = \sum_{u \neq v} X_u X_v$ on an Ising model on an empty graph. Hence for each $u$, $(X_u + 1)/2 \sim Ber(1/2)$ independently. Note that $\mathbf{E}[f(X)] = 0$.

$$2f(X) = \left(\sum_u X_u\right)^2 - n$$

$$\implies \Pr\left[|f(X)| > r\right] \geq \Pr\left[f(X) > r\right]$$

$$= \Pr\left[\left|\sum_u X_u\right| > \sqrt{r/2 + n}\right] = \Pr\left[\left|\sum_u \frac{X_u + 1}{2} - \frac{n}{2}\right| > \frac{\sqrt{r/2 + n}}{2}\right]$$

$$\geq \exp\left(-\frac{9(r/2 + n)}{4n}\right) \geq \exp(-9/4)\exp\left(-\frac{9r}{8n}\right),$$

where the last inequality follows from Theorem 3. This shows that the tail bound obtained from Theorem 1 is asymptotically nearly-tight (up to a $O(\log n)$ factor in the radius of concentration and $O(1/\log n)$ factor in the exponent of the tail bound).

# 3 Supplementary Information for Concentration of Multilinear Functions

In this section, we prove the following theorem:

**Theorem 4.** *Consider any degree-d multilinear function*

$$f_a(x) = \sum_{U \subseteq V : |U| = d} a_U \prod_{u \in U} x_u$$

*on an Ising model $p$ (defined on a graph $G = (V, E)$ such that $|V| = n$) in $\eta$-high-temperature regime with no external field. Let $\|a\|_\infty = \max_{U \subseteq V : |U| = d} |a_U|$. There exist constants $C_1 = C_1(d) > 0$ and $C_2 = C_2(d) > 0$ depending only on $d$, such that if $X \sim p$, then for any $r \geq C_1 \|a\|_\infty (n \log^2 n / \eta)^{d/2}$, we have*

$$\Pr\left[|f_a(X) - \mathbf{E}\left[f_a(X)\right]| > r\right] \leq 2\exp\left(-\frac{\eta r^{2/d}}{C_2 \|a\|_\infty^{2/d} n \log n}\right).$$

Note that, like our bilinear theorem statement, this statement is phrased for multilinear functions of degree exactly equal to $d$. This is for convenience of notation in our proof. A general purpose theorem for all degree-$d$ multilinear functions can be obtained by simply partitioning the terms based on their degree, and applying this theorem for each degree from 1 to $d$. This will not incur significant costs in the concentration bound, as the terms of order lower than $d$ have much tighter radii of concentration. Similar to Remark 1, our theorem statement still holds under the weaker assumption of Hamming contraction.

**Remark 2.** *The bound presented in Theorem 4 is asymptotically tight up to an $O_d(\log^d n)$ factor in the radius of concentration and a $O(1/\log n)$ factor in the exponent of the tail bound. This can be shown via an argument similar to that employed in Section 2.7. In particular, for an Ising model on an empty graph (where each node is completely independent of the others), we have for the d-linear function $f(x) = \sum_{U \subseteq V: |U|=d} \prod_{u \in U} x_u$, the following inequality for a big enough constant $C(d)$*

$$\Pr\left[|f(X) - \mathbf{E}\left[f(X)\right]| > r\right] \geq 2 \exp\left(-\frac{r^{2/d}}{C(d)n}\right).$$

In Section 3.1 we state some lemmata and definitions. We then show a bound on the expected value of $d$-linear functions on the Ising model in high temperature (Section 3.2). We proceed by showing our main result (Theorem 5) in Section 3.3. This result requires us to relate the expected values and tail probabilities of *hybrid* terms to those of non-hybrid terms, which we do as Theorem 6 in Section 3.4.

## 3.1   Setup

We will now proceed with the setup of our argument for concentration of $d$-linear functions.

Recall from Claim 1 the linear functions that arose when looking at the difference in the value of a bilinear function due to a step of the Glauber dynamics. In a similar vein, we define a family $F_a^d$ of multilinear functions on the Ising model of degree $\leq d - 1$ associated with any $d$-linear function $f_a(x)$.

**Definition 11.**

$$F_a^d = \bigcup_{l=0}^{d-1} F_a(l) \text{ where} \tag{69}$$

$$F_a^d(l) = \{f_a^{v_1, v_2, \ldots, v_{d-l}} \mid \forall \ distinct \ v_1, v_2, \ldots, v_{d-l} \in V\} \text{ and} \tag{70}$$

$$f_a^{v_1, v_2, \ldots, v_k}(x) = \sum_{u_1, u_2, \ldots, u_{d-k} \in V \setminus \{v_1, v_2, \ldots, v_k\}} a_{u_1 u_2 \ldots u_{d-k} v_1 v_2 \ldots v_k} X_{u_1} X_{u_2} \ldots X_{u_{d-k}}. \tag{71}$$

In the set of functions defined above, the degree $d - 1$ functions arise (up to scaling) from looking at the difference in values of $f_a(X)$ when a single step of the Glauber dynamics is taken. More generally, the degree $l - 1$ functions in the definition arise when looking at the difference in values of a degree $l$ function from $F_a^d(l)$ when a single step of the Glauber dynamics is taken.

We will also need to generalize the greedy coupling (Definition 4) used in Section 2 to couple two runs of the Glauber dynamics. The generalization will provide a way of coupling an arbitrary number of runs of the Glauber dynamics on a common Ising model $p$.

**Definition 12** (The $k$-Greedy Coupling). *Given an Ising model $p$ in high temperature, for any $k > 0$, consider the following process: Let $x_0^{(1)}, x_0^{(2)}, \ldots, x_0^{(k)} \in \Omega$ be $k$ starting configurations. Run $k$ instances of the Glauber dynamics associated with $p$ with the $i^{th}$ instance starting at state $X_0^{(i)} = x_0^{(i)}$. Let the sequence of states observed in the $i^{th}$ run of the dynamics be $X_0^{(i)}, X_1^{(i)}, X_2^{(i)}, \ldots$. Couple the $k$ runs in the following way: At each time step $t$ choose a vertex $v \in V$ uniformly at random to update in all of the $k$ runs. Let $p^i$ denote the probability that the $i^{th}$ Glauber dynamics instance sets $X_{t,v}^{(i)} = 1$. Let $p_1 \leq p_2 \leq \ldots p_k$ be a rearrangement of the $p^i$ values in increasing order. Also let $p_0 = 0$ and $p_{k+1} = 1$. Draw a number $x$ uniformly at random from $[0, 1]$ and couple the updates*

*according to the following rule:*

*If $x \in [p_l, p_{l+1}]$ for some $0 \le l \le k$, set $X_{t,v}^{(i)} = -1$ for all $1 \le i \le l$ and $X_{t,v}^{(i)} = 1$ for all $l < i \le k$.*

*We call this coupling the generalized greedy coupling of the $k$ runs or the $k$-greedy coupling.*

Now we list some properties the generalized greedy coupling (Definition 12) satisfies.

**Lemma 13** (Properties of the $k$-Greedy Coupling). *The $k$-Greedy coupling (Definition 12) is a valid coupling of $k$ runs of Glauber dynamics with the following properties.*

1. *If $X_0^{(i)} \sim p$, then $X_t^{(i)} \sim p$ for all $t \ge 0$ and for all $1 \le i \le k$.*

2. *If $p$ is an Ising model in $\eta$-high temperature, for any pair of runs $i \ne j$,*

$$\mathbf{E}\left[ d_H(X_t^{(i)}, X_t^{(j)}) \Big| (X_0^{(i)}, X_0^{(j)}) \right] \le \left(1 - \frac{\eta}{n}\right)^t d_H(X_0^{(i)}, X_0^{(j)}).$$

   *That is, the joint distribution of any two of the runs is a greedy coupling as described in Definition 4.*

3. *For any pair of runs $i \ne j$, the distribution of $X_t^{(i)}$, for any $t \ge 0$, conditioned on $X_0^{(i)}$ is independent of $X_0^{(j)}$.*

*Proof.* First we will argue that the $k$-greedy coupling is a valid coupling. Consider the marginal distribution of any one of the $k$ runs: $X_0^{(j)}, X_1^{(j)}, \ldots$. The process of generating $X_{t+1}^{(j)}$ from $X_t^{(j)}$ corresponds precisely to a step of the Glauber dynamics. Firstly, the sampling of a node among all choices is common to all runs and hence also to run $j$. Secondly, the update probabilities for the selected node are exactly what Glauber dynamics would have prescribed. Hence, it is a valid coupling of the $k$ runs. Since $p$ is the stationary distribution corresponding to all the $k$ runs, $X_0^{(i)} \sim p \implies X_t^{(i)} \sim p$ for all $t \ge 0$. We will now argue that any pair of runs $i \ne j$ are coupled according to the greedy coupling of Definition (4). Since the node to be updated in any step is chosen to be the same for all runs it is also the same for runs $i$ and $j$. Moreover, the updates of the selected node in runs $i$ and $j$ are coupled in precisely the same way as they were under the greedy coupling. Hence, the $k$-greedy coupling is a greedy coupling for any pair of runs $i$ and $j$. Hence, from Lemma 3, we have

$$\mathbf{E}\left[ d_H(X_t^{(i)}, X_t^{(j)}) \Big| (X_0^{(i)}, X_0^{(j)}) \right] \le \left(1 - \frac{\eta}{n}\right)^t d_H(X_0^{(i)}, X_0^{(j)}).$$

Also from Lemma 3, we have that the distribution of $X_t^{(i)}$, for any $t \ge 0$, conditioned on $X_0^{(i)}$ is independent of $X_0^{(j)}$. $\square$

The $k$-greedy coupling we have defined above will be useful in showing the following property about the concentration of the Hamming distance between two greedily coupled runs which is stated as Lemma 14.

**Lemma 14.** *Let $x_0, y_0 \in \Omega$ be two configurations for a high temperature Ising model $p$ on $n$ nodes. Let $\{X_t\}_{t \ge 0}, \{Y_t\}_{t \ge 0}$ be two runs of Glauber dynamics associated with $p$, coupled greedily with $X_0 = x_0, Y_0 = y_0$. Then, for any integer $t > 0$ and any real $K > 0$,*

$$\Pr\left[ |d_H(X_t, Y_t) - \mathbf{E}\left[ d_H(X_t, Y_t) | X_0, Y_0 \right]| > K | X_0 = x_0, Y_0 = y_0 \right] \le 2 \exp\left(-\frac{K^2}{16t}\right).$$

*Proof.* We will use Azuma's inequality (Lemma 4). Consider the Doob martingale associated with $d_H(X_t, Y_t)$ (similar to that of Definition 9)but now defined using the greedily coupled dynamics), parameterized by $x_0, y_0$. The $i^{th}$ term in the martingale sequence is $H_i = \mathbf{E}\left[d_H(X_t, Y_t)|X_i, Y_i\right]$ (where we have used the Markovian property of the Glauber dynamics). We look at $|H_{i+1} - H_i|$ for any $0 < i+1 \le t$.

$$
\begin{align}
|H_{i+1} - H_i| &= |\mathbf{E}\left[d_H(X_t, Y_t)|X_{i+1}, Y_{i+1}\right] - \mathbf{E}\left[d_H(X_t, Y_t)|X_i, Y_i\right]| \tag{72} \\
&\le \left|\mathbf{E}\left[d_H(X_t, Y_t)|X_{i+1}, Y_{i+1}\right] - \mathbf{E}\left[d_H(X'_{t-1}, Y'_{t-1})|X'_i, Y'_i\right]\right| \tag{73} \\
&\quad + |\mathbf{E}\left[d_H(X_t, Y_t)|X_i, Y_i\right] - \mathbf{E}\left[d_H(X_{t-1}, Y_{t-1})|X_i, Y_i\right]| \tag{74} \\
&\le \left|\mathbf{E}\left[d_H(X_t, Y_t) - d_H(X'_{t-1}, Y'_{t-1})|X_{i+1}, Y_{i+1}, X'_i, Y'_i\right]\right| + 2 \tag{75} \\
&\le \mathbf{E}\left[\left|d_H(X_t, Y_t) - d_H(X'_{t-1}, Y'_{t-1})\right||X_{i+1}, Y_{i+1}, X'_i, Y'_i\right] + 2 \\
&\le \mathbf{E}\left[\left|d_H(X_t, X'_{t-1}) + d_H(Y_t, Y'_{t-1})\right||X_{i+1}, Y'_{i+1}, X'_i, Y'_i\right] + 2 \tag{76} \\
&\le 4 \tag{77}
\end{align}
$$

where (73) and (74) follows by adding and subtracting the term $\mathbf{E}\left[d_H(X_{t-1}, Y_{t-1})|X_i, Y_i\right]$ to the difference inside the absolute value. We have also renamed $\mathbf{E}\left[d_H(X_{t-1}, Y_{t-1})|X_i, Y_i\right]$ as $\mathbf{E}\left[d_H(X'_{t-1}, Y'_{t-1})|X'_i, Y'_i\right]$ in (73) to avoid notational confusion in the later steps of our bounding, maintaining the understanding that the sequence $\{X'_t, Y'_t\}_t$ has the same distribution as $\{X_t, Y_t\}_t$. The first term in (75) bounds the term of (73) for any valid coupling of the two greedily coupled probability spaces, namely $\{X_t, Y_t\}_{t \ge i}$ and $\{X'_t, Y'_t\}_{t \ge i}$. Here we couple them using the 4-greedy coupling (Definition 12). Also, by triangle inequality, $\mathbf{E}[|d_H(X_t, Y_t) - d_H(X_{t-1}, Y_{t-1})||X_i, Y_i] \le \mathbf{E}[d_H(X_t, X_{t-1}) + d_H(Y_t, Y_{t-1})|X_i, Y_i] \le 2$. Hence (75) follows. Similarly, (76) follows because $\left|d_H(X_t, Y_t) - d_H(X'_{t-1}, Y'_{t-1})\right| \le d_H(X_t, X'_{t-1}) + d_H(Y_t, Y'_{t-1})$ and (77) follows because $\mathbf{E}[d_H(X_t, X'_{t-1})|X_{i+1}, X'_i] \le d_H(X_{i+1}, X'_i) \le 1$ (Lemma 3) and similarly $\mathbf{E}[d_H(Y_t, Y'_{t-1})|Y_{i+1}, Y'_i] \le d_H(Y_{i+1}, Y'_i) \le 1$.

Hence by Azuma's inequality applied on the martingale sequence from $0$ to $t$, we get

$$
\Pr\left[\left|d_H(X_t, X'_t) - \mathbf{E}\left[d_H(X_t, X'_t)|X_0, X'_0\right]\right| > K \Big| X_0 = x_0, X'_0 = x'_0\right] \le 2\exp\left(-\frac{K^2}{16t}\right).
$$

$\square$

## 3.2 Bounding Marginals of an Ising Model in High Temperature

The goal of this section will be to obtain a bound on the expected values of the $d$-linear functions under consideration when computed over a sample from a high temperature Ising model. We start by bounding the marginals of ferromagnetic Ising models ($\theta_{uv} \ge 0$ for all $u, v$). We will show later, using a generalization of the Fortuin-Kastelyn (FK) model, that this suffices to yield the result for non-ferromagnetic Ising models as well. The FK model connects bond percolation with the Ising model and offers powerful tools to show stochastic domination inequalities which will enable us to bound the marginals of the Ising model. We assume familiarity with the FK model as described in Chapter 10 of [RAS15].

**Lemma 15.** *Consider a ferromagnetic Ising model p ($\theta_{uv} \ge 0$ for all $(u, v) \in E$) at high temperature. Let d be a positive integer. We have*

$$
\sum_{u_1, \dots, u_d} \mathbf{E}\left[\prod_{i=1}^d X_{u_i}\right] \le 2\left(\frac{4nd\log n}{\eta}\right)^{d/2}.
$$

*Proof.* We have,

$$\left(\sum_{v \in V} X_v\right)^d = \sum_{u_1,\ldots,u_d \in V} \prod_{i=1}^d X_{u_i} \tag{78}$$

$$\implies \mathbf{E}\left[\sum_{u_1,\ldots,u_d} \prod_{i=1}^d X_{u_i}\right] = \mathbf{E}\left[\left(\sum_v X_v\right)^d\right]. \tag{79}$$

Since we are in high temperature, we have from Lemma 1 $\Pr\left[|\sum_v X_v| > K\right] \le 2\exp\left(-\frac{\eta K^2}{8n}\right)$. Since the maximum value of $\sum_v X_v = n$, we have for all $K > 0$,

$$\mathbf{E}\left[\left(\sum_v X_v\right)^d\right] \le K^d + 2n^d \exp\left(-\frac{\eta K^2}{8n}\right). \tag{80}$$

Setting $K = 2\sqrt{nd\log n/\eta}$ we get,

$$\mathbf{E}\left[\sum_{u_1,\ldots,u_d} \prod_{i=1}^d X_{u_i}\right] = \mathbf{E}\left[\left(\sum_v X_v\right)^d\right] \le 2\left(\frac{4nd\log n}{\eta}\right)^{d/2}. \tag{81}$$

□

For any Ising model $p$ on graph $G = (V, E)$ with parameter vector represented by $\theta$, we associate a ferromagnetic Ising model denoted by $p^+$ defined on the same graph $G$ where all the edges retain their magnitude but are now forced to be ferromagnetic interactions. That is, $|\theta_{uv}^p| = \theta_{uv}^{p^+}$ for all $u, v$. We have the following relation between the marginals of $p$ and those of $p^+$.

**Lemma 16.** *Consider any Ising model $p$ defined on $G = (V, E)$. Consider any subset of $k$ nodes $\{u_1, \ldots, u_k\} \subseteq V$. Then,*

$$\mathbf{E}_p\left[X_{u_1} X_{u_2} \ldots X_{u_k}\right] \le \mathbf{E}_{p^+}\left[X_{u_1} X_{u_2} \ldots X_{u_k}\right].$$

*Proof.* If $k$ is odd, then the quantities on the LHS and RHS are both 0 and hence the Lemma holds. To handle the case when $k$ is even, we consider a generalization of the FK model to possible non-ferromagnetic Ising model. This generalization is discussed in detail, for instance, in Newman's paper [New90]. The generalization retains many nice properties of the FK model. In particular, when $k$ is even, $\mathbf{E}_p\left[\prod_{i=1}^k X_{u_i}\right] = \Pr\left[X_{u_1}, X_{u_2}, \ldots, X_{u_k}\right.$ belong to same cluster ] still holds. Equation (20) in Section 5 of [New90] notes that the percolation measure associated with any Ising model is stochastically dominated by the measure associated with its corresponding ferromagnetic Ising model. A consequence of this stochastic dominance, noted in homework problem 3 in the chapter on Phase Transitions by Griffiths in [DS71], is the desired inequality

$$\mathbf{E}_p\left[X_{u_1} X_{u_2} \ldots X_{u_k}\right] \le \mathbf{E}_{p^+}\left[X_{u_1} X_{u_2} \ldots X_{u_k}\right].$$

□

**Lemma 17.** *Consider any Ising model $p$ defined on $G = (V, E)$. Consider any subset of $k$ nodes $\{u_1, \ldots, u_k\} \subseteq V$. Then,*

$$\left|\mathbf{E}_p\left[X_{u_1} X_{u_2} \ldots X_{u_k}\right]\right| \le \mathbf{E}_{p^+}\left[X_{u_1} X_{u_2} \ldots X_{u_k}\right].$$

*Proof.* If we show that $-\mathbf{E}_p\left[X_{u_1} X_{u_2} \ldots X_{u_k}\right] \leq \mathbf{E}_{p^+}\left[X_{u_1} X_{u_2} \ldots X_{u_k}\right]$, then together with Lemma 16 we get the desired result. To show the above inequality we build an Ising model $\tilde{p}$ and apply Lemma 16 to it. $\tilde{p}$ is defined as follows. The set of vertices $V$ on which $p$ is defined is augmented with $k$ dummy vertices, $\tilde{u}_1, \tilde{u}_2, \ldots, \tilde{u}_k$ and the set of edges $E$ is augmented by the addition of the set of edges $\tilde{E}$:

$$\tilde{E} = \{(u_i, \tilde{u}_i) \text{ for } i = 1, 2, \ldots, k\}.$$

The parameters for the new edges are all set to $+\infty$ except for the edge $(u_1, \tilde{u}_1)$ whose parameter is set to $-\infty$. Under this construction, we have that

$$\mathbf{E}_{\tilde{p}}\left[X_{\tilde{u}_1} X_{\tilde{u}_2} \ldots X_{\tilde{u}_k}\right] = -\mathbf{E}_p\left[X_{u_1} X_{u_2} \ldots X_{u_k}\right]. \tag{82}$$

From Lemma 16, we have

$$\mathbf{E}_{\tilde{p}}\left[X_{\tilde{u}_1} X_{\tilde{u}_2} \ldots X_{\tilde{u}_k}\right] \leq \mathbf{E}_{\tilde{p}^+}\left[X_{\tilde{u}_1} X_{\tilde{u}_2} \ldots X_{\tilde{u}_k}\right] \tag{83}$$

And since all edges of the form $(u_i, \tilde{u}_i)$ for $i = 1, 2, \ldots, k$ have a parameter of $+\infty$ under $\tilde{p}^+$, we have

$$\mathbf{E}_{\tilde{p}^+}\left[X_{\tilde{u}_1} X_{\tilde{u}_2} \ldots X_{\tilde{u}_k}\right] = \mathbf{E}_{\tilde{p}^+}\left[X_{u_1} X_{u_2} \ldots X_{u_k}\right] \tag{84}$$

Finally, we observe that this construction doesn't change the values of any of the original marginals. In particular,

$$\mathbf{E}_{\tilde{p}^+}\left[X_{u_1} X_{u_2} \ldots X_{u_k}\right] = \mathbf{E}_{p^+}\left[X_{u_1} X_{u_2} \ldots X_{u_k}\right]. \tag{85}$$

(82),(83),(84) and (85) combined give us the desired result. □

Lemma 15 together with Lemma 17 gives Corollary 2.

**Corollary 2.** *Consider any Ising model $p$ at high temperature. Let $d$ be a positive integer. We have*

$$\left|\sum_{u_1, \ldots, u_d} \mathbf{E}_p[X_{u_1} X_{u_2} \ldots, X_{u_d}]\right| \leq 2\left(\frac{4nd\log n}{\eta}\right)^{d/2}.$$

*Proof.* We have,

$$\left|\sum_{u_1, \ldots, u_d} \mathbf{E}_p[X_{u_1} X_{u_2} \ldots, X_{u_d}]\right| \leq \sum_{u_1, \ldots, u_d} |\mathbf{E}_p[X_{u_1} X_{u_2} \ldots, X_{u_d}]|$$

$$\leq \sum_{u_1, \ldots, u_d} \mathbf{E}_{p^+}[X_{u_1} X_{u_2} \ldots, X_{u_d}] \leq 2\left(\frac{4nd\log n}{\eta}\right)^{d/2}.$$

□

## 3.3   Main Theorem Statement for $d$-Linear Functions

We are now ready to show concentration of measure for $d$-linear functions. First, we define a notion of a 'good' set of configurations corresponding to a $d$-linear function $f_a(x)$ similar to how it was defined in Section 2. Doing so will help us define a stopping time for the martingale sequence we

consider later on in the argument. For any multilinear function $f_a(x)$ of degree $d$, $K > 0$, and $t_1 \geq t$, define the set $G_K^{a,d}(t_1, t)$ to be the following set of configurations:

$$G_K^{a,d}(t_1, t) = \left\{ x_t \in \Omega \,\middle|\, \forall 1 \leq l \leq d-1, \, \forall f \in F_a^d(l) \, \max\{|\mathbf{E}[f(X_{t_1})|X_t = x_t]|, |\mathbf{E}[f(X_{t_1-1})|X_t = x_t]|\} \leq K^{l/(d-1)} \right\}$$

$$\bigcap \left\{ x_t \,\middle|\, \forall 1 \leq l \leq d-1, \, \forall f \in F_a^d(l) \, \Pr\left[|f(X_{t_1}) - \mathbf{E}[f(X_{t_1})|X_t]| > K^{l/(d-1)} \,\middle|\, X_t = x_t\right] \leq 2 \exp\left(-\frac{K^{2/(d-1)}}{c_1(l)t_1}\right) \right\}$$
(86)

$$\bigcap \left\{ x_t \,\middle|\, \forall 1 \leq l \leq d-1, \, \forall f \in F_a^d(l) \, \Pr\left[|f(X_{t_1-1}) - \mathbf{E}[f(X_{t_1-1})|X_t]| > K^{l/(d-1)} \,\middle|\, X_t = x_t\right] \leq 2 \exp\left(-\frac{K^{2/(d-1)}}{c_1(l)t_1}\right) \right\}$$

where $\mathbf{E}[f(X_{t_1})|X_t]$, is defined as $0$ for $t > t_1$ for any function $f$ and $c_1(l) > 0$ is a function of $l$ which is sufficiently large. The definition may seem complicated at this moment but its usefulness will become more apparent once we delve into the proof of Theorem 5.

$G_K^{a,d}(t_1, t)$ was deliberately constructed so as to satisfy the following layering property which will be very useful in the inductive argument of Theorem 5. The following corollary is immediate from the definition of $G_K^{a,d}(t_1, t)$.

**Corollary 3.** *Suppose $f_a(x)$ is a $d$-linear function and $x \in G_K^{a,d}(t_1, t)$. For any $v \in V$, let $a^v$ represent the coefficient vector of the $(d-1)$-linear function $f_a^v(x)$. Then $x \in G_{\hat{K}}^{a^v, d-1}(t_1, t)$ where $\hat{K} = K^{(d-2)/(d-1)}$.*

We will obtain the desired concentration bound by showing that the following set of statements hold for any $d$-linear function $f_a(X_{t^*})$ (where $d$ is a constant) with bounded coefficients ($\|a\|_\infty \leq 1$). To show Statement (1) of Theorem 5, we will use Theorem 6.

**Theorem 5.** *Consider an Ising model $p$ in the $\eta$-high temperature regime. Let $t_{mix} = n \log n / \eta$ denote the mixing time of the Glauber dynamics associated with $p$. Let $f_a : \Omega \to \mathbb{R}$ be any $d$-linear function for some $d \geq 1$, such that $f_a(x) = \sum_{u_1, u_2, \ldots, u_d} a_{u_1 u_2 \ldots u_d} x_{u_1} x_{u_2} \ldots x_{u_d}$ where $a \in [-1, 1]^{\binom{V}{d}}$. Let $2t_{mix} \leq t^* \leq (n+1)t_{mix}$.*

1. *Let $X_0 \sim p$. Consider a run of the Glauber dynamics associated with $p$ running for $t^*$ steps: $X_0, X_1, \ldots, X_{t^*}$. For any $0 \leq t_0 \leq t^*$, there exist $c(d), c_2(d) > 0$ which are increasing functions of $d$ only, such that, for any $r > c(d)(n \log^2 n / \eta)^{d/2}$, we have,*

$$\Pr\left[|f_a(X_{t^*}) - \mathbf{E}[f_a(X_{t^*})|X_{t_0}]| \geq r\right] \leq 2 \exp\left(-\frac{r^{2/d}}{c_2(d)t^*}\right).$$

2. *If $X \sim p$ is a sample from the Ising model, there exist $c(d), c_3(d) > 0$ which are increasing functions of $d$ only, such that, for any $r > c(d)(n \log^2 n / \eta)^{d/2}$,*

$$\Pr\left[|f_a(X) - \mathbf{E}[f_a(X)]| \geq r\right] \leq 2 \exp\left(-\frac{r^{2/d}\eta}{c_3(d)n \log n}\right).$$

3. *For any $0 \leq t_0 \leq t^*$, there exist $c(d), c_4(d) > 0$ which are increasing functions of $d$ alone, such that, for any $r > c(d)(n \log^2 n / \eta)^{d/2}$,*

$$\Pr\left[|\mathbf{E}[f_a(X_{t^*})|X_{t_0}]| \geq r\right] \leq 2 \exp\left(-\frac{r^{2/d}}{c_4(d)t^*}\right).$$

*Proof.* The proof will proceed by induction on $d$.

**Base Case** $d = 1$**:** Statement 1 follows from Statement 3 of Lemma 6. Statement 2 of the Theorem follows immediately from Lemma 1 applied to linear functions. Statement 3 follows from Statement 2 of Lemma 6. Hence, we have shown that Theorem 5 holds when $d = 1$.

**Inductive Hypothesis:** Suppose the statements of the theorem hold for some $d > 1$. We will now show that they hold for $d + 1$.

**Statement 1:** We aim to show this statement for $d + 1$-linear functions. We will use Freedman's inequality in a similar manner as was done in Section 2. We begin by defining a martingale sequence associated with $f_a(x)$.

**Definition 13** (The $d$-Linear Martingale Sequence)**.** *Let $X_0 \in \Omega = \{\pm 1\}^n$ be a starting state. Consider a walk of the Glauber dynamics starting at $X_0$ and running for $t^*$ steps: $X_0, X_1, \ldots, X_{t^*}$. $X_{t^*}$ can be viewed as a function of all the random choices made by the dynamics up to that point. That is, $X_{t^*} = h(X_0, R_1, \ldots, R_{t^*})$ where $R_i$ is a random variable representing the random choices made by the dynamics in step $i$. Hence $f_a(X_{t^*}) = \tilde{f}_a(X_0, R_1, \ldots, R_{t^*})$ where $\tilde{f}_a = f_a \circ h$. Consider the Doob martingale associated with $\tilde{f}_a$ defined on the probability space $(O, 2^O, P)$ where $O$ is the set of all possible values of the variables $X_0, X_1, X_2, \ldots, X_{t^*}$ under the Glauber dynamics and $P$ is the function which assigns probability to events in $2^O$ according to the underlying Glauber dynamics. Also consider the increasing sequence of sub-$\sigma$-fields $2^{O_0} \subset 2^{O_1} \subset 2^{O_2} \subset \ldots 2^{O_{t^*}} = 2^O$ where $O_i$ is the set of all possible values to the variables $X_0, X_1, X_2, \ldots, X_i$ under the Glauber dynamics. The terms in the martingale sequence are as follows:*

$$B_0 = \mathbf{E}\left[\tilde{f}_a(X_0, R_1, \ldots, R_{t^*}) \Big| X_0\right]$$
$$\ldots$$
$$B_i = \mathbf{E}[\tilde{f}_a(X_0, R_1, \ldots, R_{t^*})|X_0, R_1, \ldots, R_i] \tag{87}$$
$$\ldots$$
$$B_{t^*} = \tilde{f}_a(X_0, R_1, \ldots, R_{t^*})$$

*Since the dynamics are Markovian, we can also write $B_i$ as follows:*

$$B_i = \mathbf{E}[f_a(X_{t^*})|X_i] \quad \forall \, 0 \le i \le t^*.$$

Next, we define a stopping time $T_K$ on the above martingale sequence. The definition generalizes the stopping time defined in Section 2 by requiring that many conditional expectations are small together.

**Definition 14** (Stopping Time for $d$-linear functions)**.** *Consider the martingale sequence defined in Definition 13. Let $T_K : O \to \{0\} \bigcup \mathbb{N}$ be a stopping time defined as follows:*

$$T_K = \min\{\min_{t \ge 0} \left\{ t \ \Big| \ t \notin G_K^{a,d+1}(t^*, t) \right\}, t^* + 1\}.$$

*Note that the event $\{T_K = t\}$ lies in the $\sigma$-field $2^{O_t}$ and hence the above definition is a valid stopping time.*

Using the induction hypothesis, we will show that the stopping time defined above is large with a good probability for the parameter range which is of interest to us.

**Lemma 18.** *For any $t \geq 0$, $t^* \leq (n+1)t_{mix}$, there exists $c(d) > 0$ such that, for any $K > c(d)(n\log^2 n/\eta)^{d/2}$,*

$$\Pr\left[X_t \notin G_K^{a,d+1}(t^*,t)\right] \leq 8dn^{d+1}\exp\left(-\frac{K^{2/d}}{2c_1(d)t^*}\right),$$

*where $c_1(d)$ is as defined in (86).*

*Proof.* For any $1 \leq k \leq d+1$, and $v_1, v_2, \ldots, v_k \in V$, let $E_K(v_1, v_2, \ldots, v_k)$ be the following event:

$$E_K(v_1, v_2, \ldots, v_k) = \max\left\{|\mathbf{E}[f_a^{v_1,v_2,\ldots,v_k}(X_{t^*})|X_t = x_t]|, |\mathbf{E}[f_a^{v_1,v_2,\ldots,v_k}(X_{t^*-1})|X_t = x_t]|\right\} > K^{(d+1-k)/d}.$$

Since $X_0$ is a sample from the stationary distribution $p$ of the dynamics, it follows from the property of stationary distributions that $X_t$ is also a sample from $p$. Hence we have, from the induction hypothesis, Statement 3 for multilinear functions of degree $\leq d$, and a union bound, that for any $1 \leq k \leq d+1$, $v_1, v_2, \ldots, v_k \in V$, and for $K > c(d)(n\log^2 n/\eta)^{d/2}$,

$$\Pr\left[E_K(v_1, v_2, \ldots, v_k)\right] \leq 4\exp\left(-\frac{K^{2/d}}{c_4(d+1-k)t^*}\right) \tag{88}$$

From (88) and a union bound, we get

$$\Pr\left[\exists 1 \leq k \leq d+1 \text{ and } v_1, v_2, \ldots, v_k \in V \text{ s.t. } E_K(v_1, v_2, \ldots, v_k)\right]$$

$$\leq \sum_{k=1}^{d+1} 4n^k \exp\left(-\frac{K^{2/d}}{c_4(d+1-k)t^*}\right) \leq 4dn^{d+1}\exp\left(-\frac{K^{2/d}}{c_4(d+1)t^*}\right). \tag{89}$$

For any $1 \leq k \leq d+1$, $v_1, v_2, \ldots, v_k \in V$, let $D_K(v_1, v_2, \ldots, v_k)$ denote the following event:

$$D_K(v_1, v_2, \ldots, v_k) = \max\left\{|f_a^{v_1,v_2,\ldots,v_k}(X_{t^*}) - \mathbf{E}\left[f_a^{v_1,v_2,\ldots,v_k}(X_{t^*})|X_t\right]|, \right.$$

$$\left. |f_a^{v_1,v_2,\ldots,v_k}(X_{t^*-1}) - \mathbf{E}\left[f_a^{v_1,v_2,\ldots,v_k}(X_{t^*-1})|X_t\right]|\right\} > K^{(d+1-k)/d}.$$

Let $D_K^a(t,k)$ be the event defined as

$$D_K^a(t,k) = \exists\, v_1, v_2, \ldots, v_k \in V \text{ such that } D_K(v_1, v_2, \ldots, v_k).$$

From the inductive hypothesis, Statement 1 for multilinear functions of degree $(d+1-k)(\leq d)$, and a union bound, we have,

$$\Pr\left[D_K^a(t,k)\right] = \mathbf{E}\left[\Pr\left[D_K^a(t,k)|X_t\right]\right] \leq 4n^k \exp\left(-\frac{K^{2/d}}{c_2(d+1-k)t^*}\right)$$

$$\implies \Pr\left[\Pr\left[D_K^a(t,k)|X_t\right] > \exp\left(-\frac{K^{2/d}}{c_1(d+1-k)t^*}\right)\right] \leq 4n^k \exp\left(-\frac{K^{2/d}}{2c_2(d+1-k)t^*}\right) \tag{90}$$

$$\implies \Pr\left[\bigcup_{k=1}^{d+1} \Pr\left[D_K^a(t,k)|X_t\right] > \exp\left(-\frac{K^{2/d}}{c_1(d+1-k)t^*}\right)\right] \leq \sum_{k=1}^{d+1} 4n^l \exp\left(-\frac{K^{2/d}}{2c_2(d+1-k)t^*}\right)$$

$$\leq 4dn^{d+1}\exp\left(-\frac{K^{2/d}}{2c_2(d)t^*}\right) \tag{91}$$

where (90) follows from Markov's inequality (and holds for sufficiently large $c_2(d)$) and (91) follows from a union bound. Hence,

$$\Pr\left[\bigcup_{k=1}^{d+1} \exists\, v_1, v_2, \ldots, v_k \in V \mid \Pr[D_K(v_1, v_2, \ldots, v_k)] > \exp\left(-\frac{K^{2/d}}{c_1(d+1-k)t^*}\right)\right]$$

$$\leq \Pr\left[\bigcup_{k=1}^{d+1} \Pr\left[D_K^a(t, k)|X_t\right] > \exp\left(-\frac{K^{2/d}}{c_1(d+1-k)t^*}\right)\right] \leq 4dn^{d+1}\exp\left(-\frac{K^{2/d}}{2c_2(d)t^*}\right). \qquad (92)$$

From (89) and (92), we have,

$$\Pr\left[\neg G_K^{a,d+1}(t^*, t)\right] \leq 4dn^{d+1}\exp\left(-\frac{K^{2/d}}{c_4(d)t^*}\right) + 4dn^{d+1}\exp\left(-\frac{K^{2/d}}{2c_2(d)t^*}\right) \leq 8dn^{d+1}\exp\left(-\frac{K^{2/d}}{2c_2(d)t^*}\right)$$

$\square$

**Lemma 19.** *For $t^* \leq (n+1)t_{mix}$, there exists $c(d) > 0$ such that, for any $K > c(d)(n\log^2 n/\eta)^{d/2}$,*

$$\Pr\left[t^* \geq T_K\right] \leq 8dt^*n^{d+1}\exp\left(-\frac{K^{2/d}}{2c_2(d)t^*}\right),$$

*where $c_2(d)$ is as defined in Theorem 5.*

*Proof.* From Lemma 18, we have,

$$\Pr\left[X_t \notin G_K^{a,d+1}(t^*, t)\right] \leq 8dn^{d+1}\exp\left(-\frac{K^{2/d}}{2c_2(d)t^*}\right)$$

$$\implies \Pr[t^* \geq T_K] = \Pr\left[\bigcup_{t=0}^{t^*} X_t \notin G_K^{a,d+1}(t^*, t)\right]$$

$$\leq \sum_{t=0}^{t^*}\Pr\left[X_t \notin G_K^{a,d+1}(t^*, t)\right] \leq 8dt^*n^{d+1}\exp\left(-\frac{K^{2/d}}{2c_2(d)t^*}\right).$$

$\square$

Now we will argue that the increments of the martingale are bounded up until stopping time.

**Lemma 20.** *Consider the Doob martingale defined in Definition 9. Suppose $X_i \in G_K^{a,d+1}(t^*, i)$ and $X_{i+1} \in G_K^{a,d+1}(t^*, i+1)$. For $K > c(d)(n\log^2 n/\eta)^{d/2}$, and a large enough constant $c_5$,*

$$|B_{i+1} - B_i| \leq 2d^2c_5K.$$

*Proof.* For ease of exposition, we will refer to $G_K^{a,d+1}(t^*, i)$ as simply $G_i$ in the following proof.

$$|B_{i+1} - B_i| = |\mathbf{E}\left[f_a(X_{t^*})|X_{i+1}\right] - \mathbf{E}\left[f_a(X_{t^*})|X_i\right]| \tag{93}$$

$$\leq \max_{\substack{x,y:d_H(x,y)=1,\\x\in G_{i+1},y\in G_i}} |\mathbf{E}\left[f_a(X_{t^*})|X_{i+1}=x\right] - \mathbf{E}\left[f_a(X'_{t^*})|X'_i=y\right]| \tag{94}$$

$$\leq \max_{\substack{x,y:d_H(x,y)=1,\\x\in G_{i+1},y\in G_i}} \left|\mathbf{E}\left[f_a(X_{t^*})|X_{i+1}=x\right] - \mathbf{E}\left[f_a(X'_{t^*-1})\big|X'_i=y\right]\right| + \left|\mathbf{E}\left[f_a(X'_{t^*-1})\big|X'_i=y\right] - \mathbf{E}\left[f_a(X'_{t^*})|X'_i=y\right]\right| \tag{95}$$

$$\leq \max_{\substack{x,y:d_H(x,y)=1,\\x\in G_{i+1},y\in G_i}} \left|\mathbf{E}\left[f_a(X_{t^*}) - f_a(X'_{t^*-1})\big|X_{i+1}=x, X'_i=y\right]\right| \tag{96}$$

$$+ \max_{\substack{x,y:d_H(x,y)=1,\\x\in G_{i+1},y\in G_i}} \left|\mathbf{E}\left[f_a(X'_{t^*-1}) - f_a(X'_{t^*})\big|X'_i=y\right]\right| \tag{97}$$

where in (94) we relabeled the variables in the second expectation to avoid notational confusion in the later steps of our bounding, maintaining the understanding that the sequence $\{X'_i, Y'_i\}_i$ has the same distribution as $\{X_i, Y_i\}_i$, in (95) we added and subtracted the term $\mathbf{E}[f_a(X'_{t^*-1})|X'_i = y]$, (96) holds for any valid coupling of the two chains, one starting at $X_{i+1}$ and the other starting at $X'_i$, and both running for $t^* - 1 - i$ steps. In particular, we use the greedy coupling between these two runs (Definition 4). Consider (96).

$$(96) = \max_{\substack{x,y:d_H(x,y)=1,\\x\in G_{i+1},y\in G_i}} \left|\mathbf{E}\left[\sum_{u_1,...,u_d}\prod_{e=1}^{d}X_{t^*,u_e}\left(\sum_{u_{d+1}}a_{u_1u_2...u_{d+1}}(X_{t^*,u_{d+1}} - X'_{t^*-1,u_{d+1}})\right) + \right.\right.$$

$$+ \sum_{u_1,...,u_{d-1},u_{d+1}}\prod_{e=1}^{d-1}X_{t^*,u_e}X'_{t^*-1,u_{d+1}}\left(\sum_{u_d}a_{u_1u_2...u_{d+1}}(X_{t^*,u_d} - X'_{t^*-1,u_d})\right) + ...$$

$$+ \left.\left.\sum_{u_2,...,u_{d+1}}\prod_{e=2}^{d+1}X'_{t^*-1,u_e}\left(\sum_{u_1}a_{u_1u_2...u_{d+1}}\left(X_{t^*,u_1} - X'_{t^*-1,u_1}\right)\right)\right|X_{i+1}=x, X'_i=y\right]\right|$$

$$\leq \max_{\substack{x,y:d_H(x,y)=1,\\x\in G_{i+1},y\in G_i}} \mathbf{E}\left[\sum_{u_{d+1}}\left|X_{t^*,u_{d+1}} - X'_{t^*-1,u_{d+1}}\right|\left|\sum_{u_1,...,u_d}a_{u_1u_2...u_{d+1}}\prod_{e=1}^{d}X_{t^*,u_e}\right|\right.$$

$$+ \sum_{u_d}\left|X_{t^*,u_d} - X'_{t^*-1,u_d}\right|\left|\sum_{u_1,...,u_{d-1},u_{d+1}}a_{u_1u_2...u_{d+1}}\prod_{e=1}^{d-1}X_{t^*,u_e}X'_{t^*-1,u_{d+1}}\right| + ...$$

$$+ \left.\sum_{u_1}\left|X_{t^*,u_1} - X'_{t^*-1,u_1}\right|\left|\sum_{u_2,u_3,...,u_{d+1}}a_{u_1u_2...u_{d+1}}\prod_{e=2}^{d+1}X'_{t^*-1,u_e}\right|\Bigg|X_{i+1}=x, X'_i=y\right] \tag{98}$$

We see the *hybrid* terms arise in (98). A generic term in (98) looks as follows:

$$\mathbf{E}\left[\sum_{u_l}\left|X_{t^*,u_l} - X'_{t^*-1,u_l}\right|\left|\sum_{u_1,u_2,...,u_{l-1},u_{l+1},...,u_{d+1}}a_{u_1u_2...u_{d+1}}\prod_{e=1}^{l-1}X_{t^*,u_e}\prod_{e=l+1}^{d+1}X'_{t^*-1,u_e}\right|\Bigg|X_{i+1}=x, X'_i=y\right] \tag{99}$$

Since $x \in G_{i+1}$ and $y \in G_i$, from Statements 1 and 2 of Theorem 6

$$\left| \mathbf{E}\left[ \sum_{u_1,u_2,\ldots,u_{l-1},u_{l+1},\ldots,u_{d+1}} a_{u_1 u_2 \ldots u_{d+1}} \prod_{e=1}^{l-1} X_{t^*,u_e} \prod_{e=l+1}^{d+1} X'_{t^*-1,u_e} \middle| X_{i+1} = x, X'_i = y \right] \right| \leq dK \text{ and} \tag{100}$$

$$\Pr\left[ \left| \sum_{u_1,u_2,\ldots,u_{l-1},u_{l+1},\ldots,u_{d+1}} a_{u_1 u_2 \ldots u_{d+1}} \prod_{e=1}^{l-1} X_{t^*,u_e} \prod_{e=l+1}^{d+1} X'_{t^*-1,u_e} - \right. \right.$$
$$\left. \left. \mathbf{E}\left[ \sum_{u_1,u_2,\ldots,u_{l-1},u_{l+1},\ldots,u_{d+1}} a_{u_1 u_2 \ldots u_{d+1}} \prod_{e=1}^{l-1} X_{t^*,u_e} \prod_{e=l+1}^{d+1} X'_{t^*-1,u_e} \middle| X_{i+1}, X'_i \right] \right| > K \middle| X_{i+1} = x, X'_i = y \right] \leq 2\exp\left( -\frac{K^{2/d}}{c_6(d)t^*} \right), \tag{101}$$

where $c_6(d)$ is the constant function in Statement 2 of Theorem 6. (100) and (101) together with the Hamming contraction property of the greedy coupling (Lemma 3) imply, there exists a constant $c_5 > 0$, such that, for $K > c(d)(n \log^2 n/\eta)^{d/2}$,

$$(99) \leq c_5 dK \implies (98) \leq c_5 d^2 K. \tag{102}$$

Now, we consider (97) and bound it using the same approach as was used to bound (96).

$$(97) = \max_{y \in G_i} \left| \mathbf{E}\left[ \sum_{u_1,\ldots,u_d} \prod_{e=1}^{d} X'_{t^*,u_e} \left( \sum_{u_{d+1}} a_{u_1 u_2 \ldots u_{d+1}} (X'_{t^*,u_{d+1}} - X'_{t^*-1,u_{d+1}}) \right) + \right. \right. \tag{103}$$

$$\left. + \sum_{u_1,\ldots,u_{d-1},u_{d+1}} \prod_{e=1}^{d-1} X'_{t^*,u_e} X'_{t^*-1,u_{d+1}} \left( \sum_{u_d} a_{u_1 u_2 \ldots u_{d+1}} (X'_{t^*,u_d} - X'_{t^*-1,u_d}) \right) + \ldots \right.$$

$$\left. \left. + \sum_{u_2,\ldots,u_{d+1}} \prod_{e=2}^{d+1} X'_{t^*-1,u_e} \left( \sum_{u_1} a_{u_1 u_2 \ldots u_{d+1}} \left( X'_{t^*,u_1} - X'_{t^*-1,u_1} \right) \right) \middle| X'_i = y \right] \right|$$

$$\leq \max_{y \in G_i} \mathbf{E}\left[ \sum_{u_{d+1}} \left| X'_{t^*,u_{d+1}} - X'_{t^*-1,u_{d+1}} \right| \left| \sum_{u_1,\ldots,u_d} a_{u_1 u_2 \ldots u_{d+1}} \prod_{e=1}^{d} X'_{t^*,u_e} \right| \right.$$

$$\left. + \sum_{u_d} \left| X'_{t^*,u_d} - X'_{t^*-1,u_d} \right| \left| \sum_{u_1,\ldots,u_{d-1},u_{d+1}} a_{u_1 u_2 \ldots u_{d+1}} \prod_{e=1}^{d-1} X'_{t^*,u_e} X'_{t^*-1,u_{d+1}} \right| \ldots \right.$$

$$\left. + \sum_{u_1} \left| X'_{t^*,u_1} - X'_{t^*-1,u_1} \right| \left| \sum_{u_2,u_3,\ldots,u_{d+1}} a_{u_1 u_2 \ldots u_{d+1}} \prod_{e=2}^{d+1} X'_{t^*-1,u_e} \right| \middle| X'_i = y \right] \tag{104}$$

where in (103) we have used Statement 3 of Lemma 3. A generic term in (104) looks as follows:

$$\mathbf{E}\left[ \sum_{u_l} \left| X'_{t^*,u_l} - X'_{t^*-1,u_l} \right| \left| \sum_{u_1,u_2,\ldots,u_{l-1},u_{l+1},\ldots,u_{d+1}} a_{u_1 u_2 \ldots u_{d+1}} \prod_{e=1}^{l-1} X'_{t^*,u_e} \prod_{e=l+1}^{d+1} X'_{t^*-1,u_e} \right| \middle| X'_i = y \right] \tag{105}$$

Since $y \in G_i$, from the definition of $G_i$ and the fact that $d_H(X'_{t^*}, X'_{t^*-1}) \leq 1$ we have that,

$$\left| \mathbf{E} \left[ \sum_{u_1, u_2, \ldots, u_{l-1}, u_{l+1}, \ldots, u_{d+1}} a_{u_1 u_2 \ldots u_{d+1}} \prod_{e=1}^{l-1} X'_{t^*, u_e} \prod_{e=l+1}^{d+1} X'_{t^*-1, u_e} \middle| X'_i = y \right] \right| \leq 2K \text{ and} \qquad (106)$$

$$\Pr \left[ \left| \sum_{u_1, u_2, \ldots, u_{l-1}, u_{l+1}, \ldots, u_{d+1}} a_{u_1 u_2 \ldots u_{d+1}} \prod_{e=1}^{l-1} X'_{t^*, u_e} \prod_{e=l+1}^{d+1} X'_{t^*-1, u_e} - \right. \right.$$
$$\left. \left. \mathbf{E} \left[ \sum_{u_1, u_2, \ldots, u_{l-1}, u_{l+1}, \ldots, u_{d+1}} a_{u_1 u_2 \ldots u_{d+1}} \prod_{e=1}^{l-1} X'_{t^*, u_e} \prod_{e=l+1}^{d+1} X'_{t^*-1, u_e} \middle| X'_i \right] \right| > 2K \middle| X'_i = y \right] \leq 2 \exp \left( -\frac{(2K)^{2/d}}{c_1(d) t^*} \right). \qquad (107)$$

(106) and (107) together with the property that $d_H(X'_{t^*}, X'_{t^*-1}) \leq 1$ imply that for $c_5$ sufficiently large and $K > c(d)(n \log^2 n/\eta)^{d/2}$,

$$(105) \leq c_5 dK \implies (104) \leq c_5 d^2 K. \qquad (108)$$

Hence we get that, when $X_i \in G_i$ and $X_{i+1} \in G_{i+1}$,

$$|B_{i+1} - B_i| \leq 2c_5 d^2 K. \qquad (109)$$

$\square$

As a consequence of Lemma 20, we get the following two useful corollaries.

**Corollary 4.** *Consider the martingale sequence defined in Definition 13.*

$$\Pr \left[ \forall\, 0 < i+1 < T_K,\ |B_{i+1} - B_i| \leq 2c_5 d^2 K \right] = 1.$$

*Proof.* Let $\kappa = 2c_5 d^2 K$.

$$\Pr \left[ \forall\, 0 < i+1 < T_K,\ |B_{i+1} - B_i| \leq \kappa \right]$$
$$= 1 - \Pr \left[ \exists\, 0 < i+1 < T_K,\ |B_{i+1} - B_i| > \kappa \right]$$
$$= 1 - \Pr \left[ \exists\, 0 < i+1 < T_K,\ \left( X_i \in G_K^{a,d+1}(t^*, i), X_{i+1} \in G_K^{a,d+1}(t^*, i+1) \text{ and } |B_{i+1} - B_i| > \kappa \right) \right.$$
$$\left. \text{or } \left( \left( X_i \notin G_K^{a,d+1}(t^*, i) \text{ or } X_{i+1} \notin G_K^{a,d+1}(t^*, i+1) \right) \text{ and } |B_{i+1} - B_i| > \kappa \right) \right]$$
$$= 1 - \Pr \left[ \exists\, 0 < i+1 < T_K,\ (X_i \in G_K^a(i), X_{i+1} \in G_K^a(i+1) \text{ and } |B_{i+1} - B_i| > \kappa) \right] \qquad (110)$$
$$= 1 - 0 \qquad (111)$$

where (110) follows because by the definition of $T_K$, $\Pr \left[ \exists\, 0 < i+1 < T_K,\ (X_i \notin G_K^a(i) \text{ or } X_{i+1} \notin G_K^a(i+1)) \right] = 0$, and (111) follows because $X_i \in G_K^a(i), X_{i+1} \in G_K^a(i+1) \implies |B_{i+1} - B_i| \leq \kappa$ (Lemma 20). $\square$

Corollary 4 will give us one of the required conditions to apply Freedman's inequality.

As a corollary of Lemma 20, we get a bound on the variance of the martingale differences which holds with high probability and to show it we first show Claim 3 which states that, informally, for any time step $i$, with a large probability we hit an $X_i$ such that the probability of transitioning from $X_i$ to an $X_{i+1} \in G_K^a(i+1)$ is large.

**Claim 3.** *Denote by $N_K^{a,d+1}(t^*, i)$ the following set of configurations:*

$$N_K^{a,d+1}(t^*, i) = \left\{ x_i \in \Omega \,\middle|\, \Pr\left[X_{i+1} \notin G_K^{a,d+1}(t^*, i+1)\middle|X_i = x_i\right] \leq \exp\left(-\frac{K^{2/d}}{4c_2(d)t^*}\right) \right\}. \tag{112}$$

*Then,*

$$\Pr\left[X_i \notin N_K^{a,d+1}(t^*, i)\right] \leq 8dn^{d+1}\exp\left(-\frac{K^{2/d}}{4c_2(d)t^*}\right).$$

*Proof.* We have from Lemma 18, that

$$\Pr\left[X_i \notin G_K^{a,d+1}(t^*, i)\right] \leq 8dn^{d+1}\exp\left(-\frac{K^{2/d}}{2c_2(d)t^*}\right) \quad \text{and} \tag{113}$$

$$\Pr\left[X_{i+1} \notin G_K^{a,d+1}(t^*, i+1)\right] \leq 8dn^{d+1}\exp\left(-\frac{K^{2/d}}{2c_2(d)t^*}\right). \tag{114}$$

From the definition of the set $N_K^{a,d+1}(t^*, i)$ we have,

$$\Pr\left[X_{i+1} \in G_K^{a,d+1}(t^*, i)\middle|X_i \notin N_K^{a,d+1}(t^*, i)\right] \leq 1 - \exp\left(-\frac{K^{2/d}}{4c_2(d)t^*}\right). \tag{115}$$

Then we have,

$$1 - 8dn^{d+1}\exp\left(-\frac{K^{2/d}}{2c_2(d)t^*}\right) \leq \Pr\left[X_{i+1} \in G_K^{a,d+1}(t^*, i)\right] \tag{116}$$

$$= \Pr\left[X_{i+1} \in G_K^{a,d+1}(t^*, i)\middle|X_i \in N_K^{a,d+1}(t^*, i)\right]\Pr\left[X_i \in N_K^{a,d+1}(t^*, i)\right]$$

$$+ \Pr\left[X_{i+1} \in G_K^{a,d+1}(t^*, i)\middle|X_i \notin N_K^{a,d+1}(t^*, i)\right]\Pr\left[X_i \notin N_K^{a,d+1}(t^*, i)\right] \tag{117}$$

$$\leq \Pr\left[X_i \in N_K^{a,d+1}(t^*, i)\right] + \left(1 - \exp\left(-\frac{K^{2/d}}{4c_2(d)t^*}\right)\right)\Pr\left[X_i \notin N_K^{a,d+1}(t^*, i)\right] \tag{118}$$

$$= \left(1 - \exp\left(-\frac{K^{2/d}}{4c_2(d)t^*}\right)\right) + \exp\left(-\frac{K^{2/d}}{4c_2(d)t^*}\right)\Pr\left[X_i \in N_K^{a,d+1}(t^*, i)\right]. \tag{119}$$

(119) implies,

$$\Pr\left[X_i \in N_K^{a,d+1}(t^*, i)\right] \geq \frac{\exp\left(-\frac{K^{2/d}}{4c_2(d)t^*}\right) - 8dn^{d+1}\exp\left(-\frac{K^{2/d}}{2c_2(d)t^*}\right)}{\exp\left(-\frac{K^{2/d}}{4c_2(d)t^*}\right)} \tag{120}$$

$$= 1 - 8dn^{d+1}\exp\left(-\frac{K^{2/d}}{4c_2(d)t^*}\right). \tag{121}$$

$$\square$$

As a corollary of Lemma 20, we get a bound on the variance of the martingale differences which holds with high probability.

**Lemma 21.** *Consider the martingale sequence defined in Definition 13. Denote by $N_K^{a,d+1}(t^*, i)$ the following set of configurations:*

$$N_K^{a,d+1}(t^*, i) = \left\{ x_i \in \Omega \middle| \Pr\left[X_{i+1} \notin G_K^{a,d+1}(t^*, i+1)\middle| X_i = x_i\right] \leq \exp\left(-\frac{K^{2/d}}{4c_2(d)t^*}\right) \right\}. \quad (122)$$

*Let $b = (c_5 d^2 K/2)^2 + n^{2d+2} \exp\left(-\frac{K^{2/d}}{4c_2(d)t^*}\right)$ where $c_5$ is the constant from Lemma 20. Then,*

$$\Pr\left[\mathbf{Var}[B_{i+1} - B_i|\mathcal{F}_i] > b \middle| X_i \in G_K^{a,d+1}(t^*, i) \cap N_K^{a,d+1}(t^*, i)\right] = 0.$$

*where $\mathcal{F}_i = 2^{O_i}$.*

*Proof.* Since, the random variables $X_0, \ldots, X_i$ together characterize every event in $\mathcal{F}_i$, we have,

$$\mathbf{Var}[B_{i+1} - B_i|\mathcal{F}_i] = \mathbf{Var}[B_{i+1} - B_i|X_0, X_1, \ldots, X_i] = \mathbf{Var}[B_{i+1} - B_i|X_i] \quad (123)$$

where the last equality follows from the Markov property of the Glauber dynamics. By the definition of $N_K^{a,d+1}(t^*, i)$, we have that

$$\Pr\left[X_{i+1} \notin G_K^a(i+1)|X_i \in N_K^a(i)\right] \leq \exp\left(-\frac{K^{2/d}}{4c_2(d)t^*}\right). \quad (124)$$

This implies that,

$$\Pr\left[X_{i+1} \in G_K^a(i+1) \text{ and } X_i \in G_K^a(i)|X_i \in G_K^a(i) \text{ and } X_i \in N_K^a(i)\right] \geq 1 - \exp\left(-\frac{K^{2/d}}{4c_2(d)t^*}\right)$$

$$(125)$$

$$\implies \Pr\left[|B_{i+1} - B_i| < c_5 d^2 K \middle| X_i \in G_K^{a,d+1}(t^*, i) \cap N_K^{a,d+1}(t^*, i)\right] \geq 1 - \exp\left(-\frac{K^{2/d}}{4c_2(d)t^*}\right) \quad (126)$$

$$\implies \mathbf{Var}\left[B_{i+1} - B_i\middle| X_i \in G_K^{a,d+1}(t^*, i) \cap N_K^{a,d+1}(t^*, i)\right] \leq (c_5 d^2 K/2)^2 + n^{2d+2} \exp\left(-\frac{K^{2/d}}{4c_2(d)t^*}\right)$$

$$(127)$$

where (126) follows from Lemma 20, and (127) follows from the law of total variance and from the fact that $\mathbf{Var}(X) \leq (b-a)^2/4$ when $X \in [a, b]$ with probability 1. The last inequality implies the statement of the lemma. $\qquad \square$

With Lemma 20 and Lemma 21 to bound the martingale increments, and Lemma 19 to show that the stopping time is large, we are ready to apply Freedman's inequality on the martingale defined in Definition 13 to yield Lemma 22.

**Lemma 22.** *For any $0 \leq t_0 \leq t^*$, there exists $c(d) > 0$ which is a function of $d$ alone, such that for any $r > c(d+1)(n \log^2 n/\eta)^{(d+1)/2}$,*

$$\Pr\left[|f_a(X_{t^*}) - \mathbf{E}\left[f_a(X_{t^*})|X_{t_0}\right]| \geq r\right] \leq 4 \exp\left(-\frac{r^{2/(d+1)}}{c_2(d+1)t^*}\right).$$

*Proof.* From Freedman's inequality (Lemma 5) applied on the martingale sequence (Definition 13) starting from $t_0$, we get

$$\Pr\left[\exists t < T_K \text{ s.t. } |B_t - B_{t_0}| \geq r \text{ and } V_t \leq B\right] \leq 2\exp\left(-\frac{r^2}{2(rK_1 + B)}\right) \tag{128}$$

where $K_1 \leq c_5 d^2 K$ (Lemma 20) and $V_t$ is defined as follows:

$$V_t = \sum_{i=0}^{t-1} \mathbf{Var}\left[B_{i+1} - B_i | \mathcal{F}_i\right]. \tag{129}$$

Set $B = t^*(c_5 d^2 K/2)^2 + t^* n^{2d+2}\exp\left(-\frac{K^{2/d}}{4c_2(d)t^*}\right)$ and $K = r^{d/(d+1)}$. Then we have, $rK_1 = c_5 d^2 K^{(2d+1)/d} \leq c_5 d^2 K^2 n$ where the last inequality holds because $r \leq n^{d+1}$ which in turn implies $K \leq n^d$. Similarly, since $r \geq c(d+1)(n\log^2 n/\eta)^{(d+1)/2}$, we have $K \geq c(d+1)^{d/(d+1)}(n\log^2 n/\eta)^{d/2}$. Combined with the fact that $t^* \leq (d+1)n\log n/\eta$, this implies that $t^* n^{2d+2}\exp\left(-\frac{K^{2/d}}{4c_2(d)t^*}\right) \leq t^*$ for a sufficiently large value of $c(d+1)$. This in turn implies that $B \leq t^* c_5^2 d^4 K^2/2$.

Hence (128) becomes,

$$\Pr\left[\exists t < T_K \text{ s.t. } |B_t - B_0| \geq r \text{ and } V_t \leq B\right] \leq 2\exp\left(-\frac{r^2}{2(c_5 d^2 K^2 n + t^* c_5^2 d^4 K^2/2)}\right)$$

$$\leq 2\exp\left(-\frac{r^2}{3c_5^2 d^4 K^2 t^*}\right). \tag{130}$$

Next we will bound, $\Pr\left[V_{t^*} > B\right]$ which will be useful for obtaining the desired concentration bound from (57).

$$\Pr\left[V_{t^*} > B\right] \leq \Pr\left[V_{t^*} > B \,\middle|\, \forall\, 0 \leq t \leq t^*\ X_t \in G_K^{a,d+1}(t^*,t) \cap N_K^{a,d+1}(t^*,t)\right]$$

$$+ \Pr\left[\exists\, 0 \leq t \leq t^*\ X_t \notin G_K^{a,d+1}(t^*,t) \cup N_K^{a,d+1}(t^*,t)\right]$$

$$\leq \Pr\left[\exists\, 0 \leq t < t^*\ \text{s.t. } \mathbf{Var}\left[B_{t+1} - B_t | X_t\right] > B/t^* \,\middle|\, \forall\, 0 \leq t \leq t^*\ X_t \in G_K^{a,d+1}(t^*,t) \cap N_K^{a,d+1}(t^*,t)\right]$$

$$\tag{131}$$

$$+ \sum_{t=0}^{t^*}\left(\Pr\left[X_t \notin G_K^{a,d+1}(t^*,t)\right] + \Pr\left[X_t \notin N_K^{a,d+1}(t^*,t)\right]\right) \tag{132}$$

$$\leq 0 + (t^*+1)\left(8dn^{d+1}\exp\left(-\frac{K^{2/d}}{2c_1(d)t^*}\right) + 8dn^{d+1}\exp\left(-\frac{K^{2/d}}{4c_2(d)t^*}\right)\right) \leq \frac{16(d+2)^2 n^{d+2}\log n}{\eta}\exp\left(-\frac{K^{2/d}}{4c_2(d)t^*}\right)$$

$$\tag{133}$$

where (131) holds because $V_{t^*} > B$ implies that there exists a $0 \leq t \leq t^*$ such that $\mathbf{Var}\left[B_{t+1} - B_t | X_t\right] > B/t^*$, (132) follows by an application of the union bound, and (133) follows from Lemma 21, Lemma 18 and Claim 3.

Now,

$$\Pr\left[|f(X_{t^*}) - \mathbf{E}\left[f(X_{t^*})|X_{t_0}\right]| \geq r\right] = \Pr\left[|B_{t^*} - B_0| \geq r\right]$$

$$\leq \Pr\left[|B_{t^*} - B_0| \geq r \text{ and } V_{t^*} \leq B\right] + \Pr\left[V_{t^*} > B\right] \tag{134}$$

$$\leq \Pr\left[|B_{t^*} - B_0| \geq r \text{ and } V_{t^*} \leq B \text{ and } t^* < T_K\right] + \Pr[t^* \geq T_K] + \Pr\left[V_{t^*} > B\right] \tag{135}$$

$$\leq \Pr\left[(\exists t \leq t^* \text{ s.t. } |B_t - B_0| \geq r \text{ and } V_t \leq B) \text{ and } t^* < T_K\right] + \Pr[t^* \geq T_K] + \Pr\left[V_{t^*} > B\right] \tag{136}$$

$$\leq \Pr\left[\exists t < T_K \text{ s.t. } |B_t - B_0| \geq r \text{ and } V_t \leq B\right] + \Pr[t^* \geq T_K] + \Pr\left[V_{t^*} > B\right]$$

$$\leq 2\exp\left(-\frac{r^2}{3c_5^2 d^4 K^2 t^*}\right) + 8dt^* n^{d+1}\exp\left(-\frac{r^{2/(d+1)}}{2c_2(d)t^*}\right) + \Pr\left[V_{t^*} > B\right] \tag{137}$$

$$\leq 3\exp\left(-\frac{r^2}{3c_5^2 d^4 K^2 t^*}\right) + \Pr\left[V_{t^*} > B\right] \tag{138}$$

$$\leq 3\exp\left(-\frac{r^2}{3c_5^2 d^4 K^2 t^*}\right) + \frac{16(d+2)^2 n^{d+2}\log n}{\eta}\exp\left(-\frac{K^{2/d}}{4c_2(d)t^*}\right) \tag{139}$$

$$\leq 2\exp\left(-\frac{r^{2/(d+1)}}{c_2(d+1)t^*}\right) \tag{140}$$

where (134) and (135) follow from the fact that $\Pr[A] \leq \Pr[A \cap B] + \Pr[\neg B]$, (136) follows from the fact that $\Pr[A] \leq \Pr[A \cup B]$, (137) follows from (130) and from Lemma 19, (138) holds for a sufficiently large $c(d+1)$ because $r > c(d+1)(n\log^2 n/\eta)^{(d+1)/2}$, (139) follows from (133) and (140) again holds for a sufficiently large $c(d+1)$, $c_2(d+1)$, because $r > c(d+1)(n\log^2 n/\eta)^{(d+1)/2}$ and $K = r^{d/(d+1)}$. Note that we have implicitly assumed that $\eta > 1/n$, since otherwise the concentration bounds obtained are trivial. $\qquad\square$

**Statement 2:** This statement will follow from Statement 1 applied to the case $t_0 = 0$ together with an application of the mixing time properties of the Glauber dynamics. Set $t^* = (d+2)t_{\mathrm{mix}}$. From Lemma 2, we have that

$$d_{\mathrm{TV}}(X_{t^*}|X_0, p) \leq \exp\left(-(d+1)n\log n\right)$$
$$\implies |\mathbf{E}\left[f_a(X_{t^*})|X_0\right] - \mathbf{E}\left[f_a(X_{t^*})\right]| \leq 2n^{d+1}\exp(-(d+1)n\log n) \leq 2\exp(-n). \tag{141}$$

Hence,

$$\Pr\left[|f_a(X_{t^*}) - \mathbf{E}\left[f_a(X_{t^*})\right]| \geq r\right] \leq \Pr\left[|f_a(X_{t^*}) - \mathbf{E}\left[f_a(X_{t^*})|X_0\right]| > r - 2\exp(-n)\right] \tag{142}$$

$$\leq 4\exp\left(-\frac{(r-2)^{2/(d+1)}}{c_2(d+1)t^*}\right) \tag{143}$$

$$\leq 2\exp\left(-\frac{\eta r^{2/(d+1)}}{c_3(d+1)n\log n}\right)$$

(142) follows from (141), (143) follows from Lemma 22 and (144) holds for a sufficiently large constant $c_3(d)$.

**Statement 3:** This follows from Corollary 2, and Statements 2, 1 of the theorem. Indeed, since $X_{t^*} \sim p$, we have for any $r > c(d+1)(n \log^2 n/\eta)^{(d+1)/2}$,

$$\Pr\left[|\mathbf{E}[f_a(X_{t^*})|X_{t_0}] - \mathbf{E}[f_a(X_{t^*})]| \geq r\right] \tag{144}$$

$$\leq \Pr\left[|f_a(X_{t^*}) - \mathbf{E}[f_a(X_{t^*})]| \geq r/2\right] + [|f_a(X_{t^*}) - \mathbf{E}[f_a(X_{t^*})|X_{t_0}]| \geq r/2] \tag{145}$$

$$\leq 2\exp\left(-\frac{\eta r^{2/(d+1)}}{4^{1/(d+1)}c_3(d)n\log n}\right) + 2\exp\left(-\frac{r^{2/(d+1)}}{4^{1/(d+1)}c_2(d+1)t^*}\right) \tag{146}$$

$$\leq 2\exp\left(-\frac{r^{2/(d+1)}}{c_7(d+1)t^*}\right). \tag{147}$$

where (145) follows because $|\mathbf{E}[f_a(X_{t^*})|X_{t_0}] - \mathbf{E}[f_a(X_{t^*})]| \geq r \implies |f_a(X_{t^*}) - \mathbf{E}[f_a(X_{t^*})|X_{t_0}]| \geq r/2$ or $|f_a(X_{t^*}) - \mathbf{E}[f_a(X_{t^*})]| \geq r/2$, (146) follows from Statements 2 and 1 respectively and (147) holds for a sufficiently large constant $c_7(d+1)$. Since $X_{t^*} \sim p$, from Corollary 2, and from the fact that $r > c(d+1)(n \log^2 n/\eta)^{(d+1)/2}$ we get that $|\mathbf{E}[f_a(X_{t^*})]| \leq 2(4n(d+1)\log n/\eta)^{(d+1)/2} \leq r/2$ for $c(d+1)$ sufficiently large. This implies in turn that,

$$\Pr\left[|\mathbf{E}[f_a(X_{t^*})|X_{t_0}]| \geq r\right] \leq \Pr\left[|\mathbf{E}[f_a(X_{t^*})|X_{t_0}] - \mathbf{E}[f_a(X_{t^*})]| \geq r/2\right] \tag{148}$$

$$\leq 2\exp\left(-\frac{r^{2/(d+1)}}{4^{1/(d+1)}c_7(d+1)t^*}\right) \leq 2\exp\left(-\frac{r^{2/(d+1)}}{c_4(d+1)t^*}\right), \tag{149}$$

for a sufficiently large constant $c_4(d+1)$. This shows the theorem holds by induction. $\square$

Note that a straightforward corollary of Theorem 5 is the desired statement for concentration of $d$-linear functions.

## 3.4 Supplementary Theorem Statement for *Hybrid* Functions

**Theorem 6.** *Let $p$ be an Ising model in the $\eta$-high temperature regime. Let $t_{mix} = n \log n/\eta$ denote the mixing time of the Glauber dynamics associated with $p$. Let $f_a(x) = \sum_{u_1,u_2,\ldots,u_d} a_{u_1u_2\ldots u_d}x_{u_1}x_{u_2}\ldots x_{u_d}$ be a $d$-linear function. Let $G_K^{a,d}(t_1,t)$ be the 'good' set associated with $f_a(.)$ as defined in (86). Additionally, define $G_K^{a,0}(t_1,t) = \{\pm1\}^n$. Also let $2t_{mix} \leq t^* \leq (n+1)t_{mix}$, $0 \leq t_0 \leq t^*$ and let $x_{t_0}^{(1)}$ be a starting state such that $x_{t_0}^{(1)} \in G_K^{a,d}(t^*,t_0)$. Let $x_{t_0}^{(2)}$ be a state obtained by taking a step of the Glauber dynamics starting from $x_{t_0}^{(1)}$. Suppose we also have that $x_{t_0}^{(2)} \in G_K^{a,d}(t^*,t_0)$. Consider the 2 runs of the Glauber dynamics associated with $p$ with the $j^{th}$ run starting at $X_{t_0}^{(j)} = x_{t_0}^{(j)}$ respectively, and coupled together using the greedy coupling (Definition 4). Denote the state of run $j$ at time $t \geq t_0$ by $X_t^{(j)}$. Consider any $l$-linear function from $F_a^d(l)$: $f_a^{v_1,v_2,\ldots,v_{d-l}}$. Denote its coefficient vector by $\alpha$. That is,*

$$f_a^{v_1,v_2,\ldots,v_{d-l}}(x) = \sum_{u_1,u_2,\ldots,u_l} \alpha_{u_1u_2\ldots u_l}x_{u_1}x_{u_2}\ldots x_{u_l}.$$

*Note that $\alpha_{u_1u_2\ldots u_l} = a_{u_1,u_2,\ldots,u_l,v_1,v_2,\ldots,v_{d-l}}$. For each $f_a^{v_1,v_2,\ldots,v_{d-l}}$ we define an associated class of hybrid functions defined over the concatenated states from the two runs of Glauber dynamics*

*described above as follows:*

$$f_a^{v_1,v_2,\ldots,v_{d-l}}\left(x^{(1:2)}\right) = \sum_{u_1,u_2,\ldots,u_l} \alpha_{u_1u_2\ldots u_l} x_{u_1}^{(1)} x_{u_2}^{(1)} \ldots x_{u_{l_1}}^{(1)} x_{u_{l_1+1}}^{(2)} x_{u_l}^{(2)} \tag{150}$$

$$= \sum_{u_1,u_2,\ldots,u_l} \alpha_{u_1u_2\ldots u_l} \prod_{b=1}^{2} \prod_{e=1}^{l_b} x_{u_{(b-1)l_1+e}}^{(b)} \tag{151}$$

*where $l_2 = l - l_1$. Then, the following two statements hold for all $0 \le l \le d-1$, and for any $f \in F_a^d(l)$, there exist constants $c(d), c_6(l)$ such that:*

1. *For any $K > c(d)(n\log^2 n/\eta)^{(d-1)/2}$*

$$\left| \mathbf{E}\left[ f\left(X_{t^*}^{(1:2)}\right) \middle| X_{t_0}^{(j)} = x_{t_0}^{(j)} \text{ for } j = 1,2 \right] \right| \le (l+1)K^{l/(d-1)}.$$

2. *For any $K > c(d)(n\log^2 n/\eta)^{(d-1)/2}$,*

$$\Pr\left[ \left| f\left(X_{t^*}^{(1:2)}\right) - \mathbf{E}\left[ f\left(X_{t^*}^{(1:2)}\right) \middle| X_{t_0}^{(j)} \text{ for } j = 1,2 \right] \right| > K^{l/(d-1)} \middle| X_{t_0}^{(j)} = x_{t_0}^{(j)} \text{ for } j = 1,2 \right] \le 2\exp\left( -\frac{K^{2/(d-1)}}{c_6(l)t^*} \right).$$

*Proof.* The proof will proceed by induction on $l$.

**Base Cases: $l = 0, 1$:** When $l = 0$, the functions under consideration are all just constant functions and hence both the statements hold immediately. Consider the next case $l = 1$ as well. In this case the functions are linear and hence no hybrid terms can arise. The statements of the theorem follow immediately from the definition of $G_K^{a,d}(t^*, t_0)$.

We will assume the statements of the theorem hold for some $1 < l < d-1$. And proceed to show them for $l+1$.

**Induction Step for Statement 1:** We will begin with Statement 1. We wish to show it for $l+1$-linear hybrid functions. At a high level, we will try to express any hybrid function of degree $l+1$ as a non-hybrid function of degree $l+1$ plus functions which resemble hybrid functions of degree $l$ multiplied with the Hamming distance between the two runs at time $t^*$. The definition of the 'good' set will allow us to bound the conditional expectation of the non-hybrid function of degree $l+1$. The inductive hypothesis together with Hamming contraction properties will help us bound the other functions. The total number of such functions we will encounter is $\text{poly}(d)$ and hence we incur a constant factor $(\text{poly}(d))$ loss in the final bound. Consider any function $f_a^{v_1,v_2,\ldots,v_{d-l-1}}\left(x^{(1:2)}\right)$ from the family $F_a^d(l+1)$ with coefficient vector $\alpha$. We will show the statement by inducting on $l_2 = l+1 - l_1$. For a given degree $l+1$ and a certain value of $l_2$ the inductive claim is as follows. For any $K > c(d)(n\log^2 n/\eta)^{(d-1)/2}$,

$$\left| \mathbf{E}\left[ \sum_{u_1,u_2,\ldots,u_{l+1}} \alpha_{u_1u_2\ldots u_{l+1}} \prod_{b=1}^{2} \prod_{e=1}^{l_b} X_{t^*,u_{s_b+e}}^{(b)} \middle| X_{t_0}^{(j)} = x_{t_0}^{(j)} \text{ for } j = 1,2 \right] \right| \le (l_2+1)K^{(l+1)/(d-1)}. \tag{152}$$

As a base case consider the scenario when $l_2 = 0$. Then the function under consideration is a vanilla non-hybrid $l+1$-linear function from $F_a^d(l+1)$ and the statement holds by the definition of $G_K^{a,d}(t_0)$. Suppose the statement holds for some $l_2$. We will show that it holds for any $l+1$-linear hybrid

function in $F_a^d(l+1)$ with $l_2+1$ terms coming from the 2nd run. Then the LHS of Statement 1 is of the form,

$$\left| \mathbf{E}\left[ \sum_{u_1,u_2,\ldots,u_{l+1}} \alpha_{u_1 u_2 \ldots u_{l+1}} \prod_{e=1}^{l_1-1} X_{t^*,u_e}^{(1)} \prod_{e=l_1}^{l+1} X_{t^*,u_e}^{(2)} \middle| X_{t_0}^{(j)} = x_{t_0}^{(j)} \text{ for } j=1,2 \right] \right|$$

$$\leq \left| \mathbf{E}\left[ \sum_{u_1,u_2,\ldots,u_{l+1}} \alpha_{u_1 u_2 \ldots u_{l+1}} \prod_{e=1}^{l_1-1} X_{t^*,u_e}^{(1)} \prod_{e=l_1}^{l} X_{t^*,u_e}^{(2)} X_{t^*,u_{l+1}}^{(1)} \middle| X_{t_0}^{(j)} = x_{t_0}^{(j)} \text{ for } 1 \leq j \leq 2 \right] \right| \qquad (153)$$

$$+ \left| \mathbf{E}\left[ \sum_{u_{l+1}} \left( X_{t^*,u_{l+1}}^{(2)} - X_{t^*,u_{l+1}}^{(1)} \right) \sum_{u_1,u_2,\ldots,u_l} \alpha_{u_1 u_2 \ldots u_l u_{l+1}} \prod_{e=1}^{l_1-1} X_{t^*,u_e}^{(1)} \prod_{e=l_1}^{l} X_{t^*,u_e}^{(2)} \middle| X_{t_0}^{(j)} = x_{t_0}^{(j)} \text{ for } j=1,2 \right] \right|.$$

$$(154)$$

We have, by the inductive hypothesis, Statement 1 for functions in $F_a^d(l+1)$ with $l_2$ terms from the second run that,

$$(153) \leq l_2 K^{(l+1)/(d-1)}. \qquad (155)$$

Similarly, from the inductive hypothesis for functions in $F_a^d(l)$, Statements 1, 2 and Corollary 3, we have

$$\left| \mathbf{E}\left[ \sum_{u_1,u_2,\ldots,u_l} \alpha_{u_1 u_2 \ldots u_l u_{l+1}} \prod_{e=1}^{l_1-1} X_{t^*,u_e}^{(1)} \prod_{e=l_1}^{l} X_{t^*,u_e}^{(2)} \middle| X_{t_0}^{(j)} = x_{t_0}^{(j)} \text{ for } j=1,2 \right] \right| \leq (l+1)K^{l/(d-1)}, \quad (156)$$

$$\Pr\left[ \left| \sum_{u_1,u_2,\ldots,u_l} \alpha_{u_1 u_2 \ldots u_l u_{l+1}} \prod_{e=1}^{l_1-1} X_{t^*,u_e}^{(1)} \prod_{e=l_1}^{l} X_{t^*,u_e}^{(2)} - \right. \right.$$

$$\left. \left. \mathbf{E}\left[ \sum_{u_1,u_2,\ldots,u_l} \alpha_{u_1 u_2 \ldots u_l u_{l+1}} \prod_{e=1}^{l_1-1} X_{t^*,u_e}^{(1)} \prod_{e=l_1}^{l} X_{t^*,u_e}^{(2)} \middle| X_i^{(j)} \text{ for } j=1,2 \right] \right| > K^{l/(d-1)} \middle| X_{t_0}^{(j)} = x_{t_0}^{(j)} \text{ for } j=1,2 \right]$$

$$\leq 2 \exp\left( -\frac{K^{2/(d-1)}}{c_6(d) t^*} \right). \qquad (157)$$

Putting together (156) and (157) with the Hamming contraction properties for any pair of runs in the coupled dynamics and the fact that $(154) \leq 2n^{l+1}$ always (similar to how it was shown in Section 2), we get

$$(154) \leq (l+1)K^{l/(d-1)} + 4n^{l+1}\exp\left( -\frac{K^{2/(d-1)}}{c_6(d) t^*} \right) \leq K^{(l+1)/(d-1)} \qquad (158)$$

where the last inequality holds because $K > c(d)(n \log^2 n/\eta)^{(d-1)/2}$. (155) and (158) together imply the desired bound for the case $l_2+1$. Hence this proves Statement 1 for $l+1$-linear functions.

**Induction Step for Statement 2:** Next we look at Statement 2 for $l+1$-linear hybrid functions in $F_a^d(l+1)$. The high level approach is similar to that used above in the proof for Statement 1. We will try to express any hybrid function of degree $l+1$ as a non-hybrid function of degree $l+1$ plus functions which resemble hybrid functions of degree $l$ multiplied with the Hamming distance between

the two runs at time $t^*$. To bound the probability of deviation of the non-hybrid function we use the definition of the 'good' set and to bound the probability of deviation of the other functions we appeal to the induction hypothesis and Hamming contraction properties. We incur an additional factor which is $\exp(d)$ in the final bound (recall that we are treating $d$ as fixed). Consider any function $f_a^{v_1,v_2,...,v_{d-l-1}}(x)$ from the family $F_a^d(l+1)$. Denote its coefficient vector by $\alpha$. That is, $\alpha_{u_1 u_2 ... u_l u_{l+1}} = a_{u_1 u_2 ... u_{l+1} v_1 v_2 ... v_{d-l-1}}$. We again induct on $l_2$, the number of terms in the function corresponding to the second run of the Glauber dynamics. Given an $l+1$-linear $f \in F_a^d(l+1)$ with coefficient vector $\alpha$, the inductive claim for the class of hybrid functions associated with $f$ with a certain value of $l_2$ is as follows:

$$\Pr\left[\left|f\left(X_{t^*}^{(1:2)}\right) - \mathbf{E}\left[f\left(X_{t^*}^{(1:2)}\right)\Big| X_{t_0}^{(j)} \,\forall\, 1 \le j \le 2\right]\right| > K^{(l+1)/(d-1)}\Big| X_{t_0}^{(j)} = x_{t_0}^{(j)} \,\forall\, 1 \le j \le 2\right]$$
$$\le 2\exp\left(-\frac{K^{2/(d-1)}}{c_8(l+1+l_2)t^*}\right), \tag{159}$$

where $c_8(l)$ is an increasing function of $l$. As a base case consider the scenario when $l_2 = 0$. Then the function under consideration is a vanilla non-hybrid $l+1$-linear function from $F_a^d(l+1)$ and the statement holds by the definition of $G_K^{a,d}(t_0)$. Suppose the statement holds for some $l_2 > 0$. We will show that it holds for any $l+1$-linear function from $F_a^d(l+1)$ with number of terms corresponding to the second run equal to $l_2 + 1$. Consider the LHS of the statement for any such function:

$$\Pr\Bigg[\Bigg| \sum_{u_1,u_2,...,u_{l+1}} \alpha_{u_1 u_2 ... u_{l+1}} \prod_{e=1}^{l_1-1} X_{t^*,u_e}^{(1)} \prod_{e=l_1}^{l+1} X_{t^*,u_e}^{(2)}$$
$$-\mathbf{E}\left[\sum_{u_1,u_2,...,u_{l+1}} \alpha_{u_1 u_2 ... u_{l+1}} \prod_{e=1}^{l_1-1} X_{t^*,u_e}^{(1)} \prod_{e=l_1}^{l+1} X_{t^*,u_e}^{(2)} \Bigg| X_{t_0}^{(j)} \,\forall\, 1 \le j \le 2\right]\Bigg| > K^{(l+1)/(d-1)}\Bigg| X_{t_0}^{(j)} = x_{t_0}^{(j)} \,\forall\, 1 \le j \le 2\Bigg] \tag{160}$$

$$\le \Pr\Bigg[\Bigg| \sum_{u_1,u_2,...,u_{l+1}} \alpha_{u_1 u_2 ... u_{l+1}} \prod_{e=1}^{l_1-1} X_{t^*,u_e}^{(1)} \prod_{e=l_1}^{l} X_{t^*,u_e}^{(2)} X_{t^*,u_{l+1}}^{(1)}$$
$$-\mathbf{E}\left[\sum_{u_1,u_2,...,u_{l+1}} \alpha_{u_1 u_2 ... u_{l+1}} \prod_{e=1}^{l_1-1} X_{t^*,u_e}^{(1)} \prod_{e=l_1}^{l} X_{t^*,u_e}^{(2)} X_{t^*,u_{l+1}}^{(1)} \Bigg| X_{t_0}^{(j)} \,\forall\, 1 \le j \le 2\right]\Bigg| \ge K^{(l+1)/(d-1)}/2\Bigg| X_{t_0}^{(j)} = x_{t_0}^{(j)} \,\forall\, 1 \le j \le 2\Bigg] \tag{161}$$

$$+\Pr\Bigg[\Bigg| \sum_{u_{l+1}} \left(X_{t^*,u_{l+1}}^{(1)} - X_{t^*,u_{l+1}}^{(2)}\right) \sum_{u_1,u_2,...,u_l} \alpha_{u_1 u_2 ... u_l u_{l+1}} \prod_{e=1}^{l_1-1} X_{t^*,u_e}^{(1)} \prod_{e=l_1}^{l} X_{t^*,u_e}^{(2)}$$
$$-\mathbf{E}\left[\sum_{u_{l+1}} \left(X_{t^*,u_{l+1}}^{(1)} - X_{t^*,u_{l+1}}^{(2)}\right) \sum_{u_1,u_2,...,u_l} \alpha_{u_1 u_2 ... u_l u_{l+1}} \prod_{e=1}^{l_1-1} X_{t^*,u_e}^{(1)} \prod_{e=l_1}^{l} X_{t^*,u_e}^{(2)}\right]\Bigg| > K^{(l+1)/(d-1)}/2\Bigg| X_{t_0}^{(j)} = x_{t_0}^{(j)} \,\forall\, 1 \le j \le 2\Bigg] \tag{162}$$

We have, by the inductive hypothesis for $l+1$-linear functions in $F_a^d(l+1)$ with $l_2$ terms from the 1st run that,

$$(161) \le 2\exp\left(-\frac{K^{2/(d-1)}}{c_8(l+1+l_2)t^*}\right). \tag{163}$$

From the property of the 'good' set $G_K^{a,d}(t_0)$ (Corollary 3), and the inductive hypothesis, Statements 1 and 2 for functions in $F_a^d(l)$ we get,

$$\left| \mathbf{E}\left[ \sum_{u_1,u_2,\ldots,u_l} \alpha_{u_1 u_2 \ldots u_l u_{l+1}} \prod_{e=1}^{l_1-1} X_{t^*,u_e}^{(1)} \prod_{e=l_1}^{l} X_{t^*,u_e}^{(2)} \middle| X_{t_0}^{(j)} = x_{t_0}^{(j)} \, \forall \, 1 \le j \le 2 \right] \right| \le (l+1) K^{l/(d-1)} \tag{164}$$

$$\Pr\left[ \left| \sum_{u_1,u_2,\ldots,u_l} \alpha_{u_1 u_2 \ldots u_l u_{l+1}} \prod_{e=1}^{l_1-1} X_{t^*,u_e}^{(1)} \prod_{e=l_1}^{l} X_{t^*,u_e}^{(2)} \right. \right.$$

$$\left. -\mathbf{E}\left[ \sum_{u_1,u_2,\ldots,u_l} \alpha_{u_1 u_2 \ldots u_l u_{l+1}} \prod_{e=1}^{l_1-1} X_{t^*,u_e}^{(1)} \prod_{e=l_1}^{l} X_{t^*,u_e}^{(2)} \middle| X_{t_0}^{(j)} \, \forall \, 1 \le j \le 2 \right] \right| > K^{l/(d-1)}/2 \left| X_{t_0}^{(j)} = x_{t_0}^{(j)} \, \forall \, 1 \le j \le 2 \right]$$

$$\le 2 \exp\left( -\frac{K^{2/(d-1)}}{4^{1/l} c_8 (l+l_2) t^*} \right). \tag{165}$$

When $K > c(d)(n \log^2 n/\eta)^{(d-1)/2}$, (164) and (165) together with the Hamming contraction property of the coupled dynamics (Lemma 3) imply that

$$\mathbf{E}\left[ \sum_{u_{l+1}} \left( X_{t^*,u_{l+1}}^{(1)} - X_{t^*,u_{l+1}}^{(2)} \right) \sum_{u_1,u_2,\ldots,u_l} \alpha_{u_1 u_2 \ldots u_l u_{l+1}} \prod_{e=1}^{l_1-1} X_{t^*,u_e}^{(1)} \prod_{e=l_1}^{l} X_{t^*,u_e}^{(2)} \middle| X_{t_0}^{(j)} = x_{t_0}^{(j)} \, \forall \, 1 \le j \le 2 \right] \le (l+1) K^{l/d} \tag{166}$$

$$\implies (162) \le$$

$$\Pr\left[ \left| \sum_{u_{l+1}} \left( X_{t^*,u_{l+1}}^{(1)} - X_{t^*,u_{l+1}}^{(2)} \right) \sum_{u_1,u_2,\ldots,u_l} \alpha_{u_1 u_2 \ldots u_l u_{l+1}} \prod_{e=1}^{l_1-1} X_{t^*,u_e}^{(1)} \prod_{e=l_1}^{l} X_{t^*,u_e}^{(2)} \right| > K^{(l+1)/(d-1)}/4 \middle| X_{t_0}^{(j)} = x_{t_0}^{(j)} \, \forall \, 1 \le j \le 2 \right] \tag{167}$$

Note this follows since $(l+1) K^{l/d} \le K^{(l+1)/(d-1)}/2$ when $K > c(d)(n \log^2 n/\eta)^{(d-1)/2}$ and $c(d)$ is sufficiently large.

From Lemma 14 we have, for any $K_1 > 2$,

$$\Pr\left[ \left| \sum_{u_{l+1}} \left( X_{t^*,u_{l+1}}^{(2)} - X_{t^*,u_{l+1}}^{(1)} \right) \right| > K_1 \middle| X_{t_0}^{(j)} = x_{t_0}^{(j)} \, \forall \, 1 \le j \le 2 \right] \tag{168}$$

$$\le \Pr\left[ d_H(X_{t^*}^{(2)}, X_{t^*}^{(1)}) > K_1/2 \middle| X_{t_0}^{(j)} = x_{t_0}^{(j)} \, \forall \, 1 \le j \le 2 \right] \tag{169}$$

$$\le \Pr\left[ \left| d_H(X_{t^*}^{(2)}, X_{t^*}^{(1)}) - \mathbf{E}\left[ d_H(X_{t^*}^{(2)}, X_{t^*}^{(1)}) \middle| X_{t_0}^{(j)} = x_{t_0}^{(j)} \, \forall \, 1 \le j \le 2 \right] \right| > K_1/2 - 1 \middle| X_{t_0}^{(j)} = x_{t_0}^{(j)} \, \forall \, 1 \le j \le 2 \right]$$

$$\le 2 \exp\left( -\frac{(K_1-2)^2}{64(t^*-t_0)} \right) \le 2 \exp\left( -\frac{K_1^2}{70 t^*} \right) \tag{170}$$

where (170) follows because $t^* - t_0 \le t^*$, and $\mathbf{E}[d_H(X_{t^*}^{(1)}, X_{t^*}^{(2)})|X_{t_0}^{(j)} = x_{t_0}^{(j)} \, \forall \, 1 \le j \le 2] \le 1$. Now,

set $K_1 = K^{1/(d-1)}/2$. Applying (170) for these parameter values, we get,

$$(167) \leq \Pr\left[\left|\sum_{u_{l+1}}\left(X^{(1)}_{t^*,u_{l+1}} - X^{(2)}_{t^*,u_{l+1}}\right)\right| > K^{1/(d-1)}/2 \middle| X^{(j)}_{t_0} = x^{(j)}_{t_0} \,\forall\, 1 \leq j \leq 2\right] \tag{171}$$

$$+ \Pr\left[\exists u_{l+1} \text{ s.t. } \left|\sum_{u_1,u_2,\ldots,u_l} \alpha_{u_1 u_2 \ldots u_{l+1}} \prod_{e=1}^{l_1-1} X^{(1)}_{t^*,u_e} \prod_{e=l_1}^{l} X^{(2)}_{t^*,u_e}\right| > K^{l/(d-1)}/2 \middle| X^{(j)}_{t_0} = x^{(j)}_{t_0} \,\forall\, 1 \leq j \leq 2\right] \tag{172}$$

$$\leq 2\exp\left(-\frac{K^{2/(d-1)}}{280 t^*}\right) + n\exp\left(-\frac{K^{2/(d-1)}}{c_8(l+l_2)t^*}\right) \leq (n+1)\exp\left(-\frac{K^{2/(d-1)}}{c_8(l+l_2)t^*}\right). \tag{173}$$

(173) with (163) implies

$$(160) \leq (n+1)\exp\left(-\frac{K^{2/(d-1)}}{c_8(l+l_2)t^*}\right) + 2\exp\left(-\frac{K^{2/(d-1)}}{c_8(l+1+l_2)t^*}\right) \leq 2\exp\left(-\frac{K^{2/(d-1)}}{c_8(l+2+l_2)t^*}\right), \tag{174}$$

for sufficiently large $c_8(.)$. This gives the desired bound for $l+1$-linear functions in $F^d_a(l+1)$ with $l_2 + 1$ terms from the second run. By induction, this proves Statement 2 for all $1 \leq l \leq d-1$ and all functions from $F^d_a(l)$.

$\square$

# 4 Additional details about Experiments

## 4.1 Details about synthetic experiments

Our departures from the null hypothesis are generated in the following manner, parameterized by some parameter $\tau \in [0,1]$. The grid is initialized by setting each node independently to be $-1$ or $1$ with equal probability. We then iterate over the nodes in column major order. For the node $x$ at position $(i,j)$, we select a node $y$ at one of the following positions uniformly at random: $(i, j+1), (i, j+2), (i+1, j+1), (i+1, j), (i+2, j), (i+1, j-1)$. Then, with probability $\tau$, we set $y$ to have the same value as $x$. We imagine this construction as a type of social network model, where each individual tries to convert one of his nearby connections in the network to match his signal, and is successful with probability $\tau$.

## 4.2 Details about experiments on Last.fm dataset

We report additional statistics extracted from the Last.fm dataset [CBK11].

The network has $n = 1892$ and $17632$ artists. There are $m = 12717$ edges, with an average degree of $13.443$. There are $92834$ user-listened artist relations, where the artists listed for a user is truncated at $50$. On average, $5.265$ users listened to each artist, but we focus on artists who had significatly more listens ($\sim 400$ or more).

(a) Our deviation from the null with $\tau = 0.04$

(b) A sample from the Ising model with $\theta = 0.035$

Figure 1: A visual comparison between the null and a deviation from the null

(a) MCMC $Z_1$ for Lady Gaga

(b) MCMC $Z_2$ for Lady Gaga

Figure 2: MCMC Statistics for Lady Gaga

| Artist | # of favorites | MPLE $h$ | MPLE $\theta$ | $Z_1$ | $Z_2$ | Reject $Z_1$? | Reject $Z_2$? |
|---|---|---|---|---|---|---|---|
| Lady Gaga | 611 | $-0.481$ | 0.0700 | 9017.3 | 106540 | Yes | Yes |
| Britney Spears | 522 | $-0.6140$ | 0.0960 | 10585 | 119560 | Yes | Yes |
| Rihanna | 484 | $-0.715$ | 0.1090 | 11831 | 126750 | Yes | Yes |
| The Beatles | 480 | $-0.3550$ | 0.0310 | 2157.8 | 22196 | No | No |
| Katy Perry | 473 | $-0.6150$ | 0.0890 | 8474 | 90762 | Yes | Yes |
| Madonna | 429 | $-0.5400$ | 0.0860 | 4580.9 | 40395 | Yes | No |
| Avril Lavigne | 417 | $-0.5580$ | 0.1020 | 5145.9 | 48639 | Yes | Yes |
| Christina Aguilera | 407 | $-0.7810$ | 0.1060 | 9979.8 | 101210 | Yes | Yes |
| Muse | 400 | $-0.5430$ | 0.0160 | 923.55 | 6911 | No | No |
| Paramore | 399 | $-0.4530$ | 0.0480 | 2047.1 | 18119 | Yes | Yes |

(a) MCMC $Z_1$ for The Beatles

(b) MCMC $Z_2$ for The Beatles

Figure 3: MCMC Statistics for The Beatles

## Footnotes

[1]Note that Theorem 15.1 of [LPW09] uses a definition of high temperature which is less general than the one we present here. But it can also be shown via very similar calculations to hold for our more general version of the high temperature regime.

[2]Concentration under an external field (with appropriate re-centering) is discussed in Section 2.6.