[Reviews · NeurIPS 2017]

Reviewer 1



This paper discusses concentration of bilinear functions of Ising systems and shows concentration of optimal up to logarithms order. I have read thoroughly the appendix, and the main results appear to be correct -- and I find them interesting. My main concerns with the paper are: (i) The writing: the paper refers to the supplementary material too much to my liking. For example sections 3.2.x for x in [2,...,5], do not read well, since the details provided are insufficient or vague. Part of the applications section also refers to the supplement multiple times. (ii) Although the datasets used in the application are interesting I think the analysis does not relate well to the theoretical study. First, using MPLE may be unstable and inconsistent in such high-dimensional data (n = 1600) which would render the second step -- running an MCMC based on those parameters useless. Note that while [BM16] prove the consistency of the MPLE, that typically requires the dimension to be fixed, and much smaller than the sample size. Second, why the fact that the bilinear functions concentrate almost sub-exponentially justify their use as test statistics -- as one can use any type of test statistic with the same idea?

Reviewer 2



Summary: The paper considers concentration of bilinear functions of samples from an ferromagnetic Ising model under high temperature (satisyfing Dobrushin's condition which implies fast mixing). The authors show that for the case when there is no external field, that any bilinear function with bounded coefficients concentrate around a radius of O(n) around the mean where n is the number of nodes in the Ising Model. The paper also shows that for Ising models with external fields, bilinear functions of mean shifted variables again obey such concentration properties. The key technical contribution is adapting the theory of exhangeable pairs from [Cha05] which has been applied for linear functions of the Ising model and apply it to bilinear functions. Naive applications give concentration radius of n^{1.5} which the authors improve upon. The results are also optimal due to the lower bound for the worst case. Strengths: a) The key strength is the proof technique - the authors show that the usual strategy of creating an exchangeable pair from a sample X from the Ising model and X' which is one step of glauber dynamics is not sufficient. So they restrict glauber dynamics to a desirable part of the configuration space and show that the same exchangeable trick can be done on this censored glauber dynamics. They also show that this glauber dynamics mixes fast using path coupling results and also by a coupling argument by chaining two glauber Markov Chains tightly. This tightly chained censored glauber dynamics really satisfies all conditions of the exchangeable pairs argument needed. The authors show this carefully by several Lemmas in the appendix. I quite enjoyed reading the paper. b) I have read the proofs in the appendix. Barring some typos that can be fixed, the proof is correct to the best of my knowledge and the technique is the major contribution of the paper. c) The authors apply this to test if samples from a specific distribution on social networks is from an Ising model in the high temperature regime. Both synthetic and real world experiments are provided. Weaknesses: Quite minor weaknesses: a) Page 1 , appendix : Section 2, point 2 Should it not be f(X)- E[f(X)] ?? b) Why is Lemma 1 stated ? It does not help much in understanding Thm 1 quoted. A few lines connecting them could help. c) Definition 1, I think It should be intersection and not union ?? Because proof of lemma 5 has a union of bad events which should be the negation of intersection of good events we want in Defn 1. d) Typo in the last equation in Claim 3. e) Lemma 9 statement. Should it not be (1-C/n)^t ?? f) In proof of Lemma 10, authors want to say that due to fast mixing, total variation distance is bounded by some expression. I think its a bit unclear what the parameter eta is ?? It appears for the first time. It will be great if the authors can explain that step better . g) Regarding the authors experiments with Last.fm - Whats the practical use case of testing if samples came from an Ising model or not ? Actually the theoretical parts of this paper are quite strong. But I just wanted to hear author's thoughts. h) Even for linear functions of uniform distribution on the boolean hypercube, at a different radius there are some anti concentration theorems like Littlewood Offord. Recently they have been generalized to bilinear forms (by Van Vu , terry tao with other co authors). I am wondering if something could be said about anti concentration theorems for Bilinear forms on Ising Models ??